# MemGUI-Bench: Benchmarking Memory of Mobile GUI Agents in Dynamic Environments

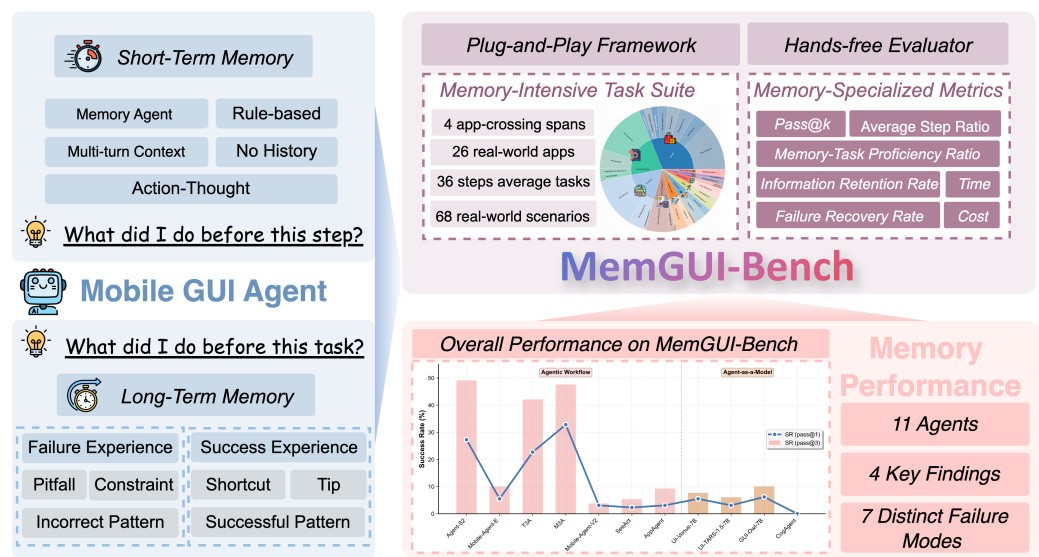

Figure 1: An overview of MemGUI-Bench, first comprehensive benchmark for GUI agent memory evaluation.

## Abstract

Current mobile GUI agent benchmarks systematically fail to assess memory capabilities, with only 5.2-11.8% memory-related tasks and no cross-session learning evaluation. We introduce **MemGUI-Bench**, the the most comprehensive, memory-centric benchmark with pass@k and a staged LLM-as-judge evaluator. Our contributions include: (1) a systematic memory taxonomy with analysis of 11 prominent agents; (2) 128 tasks across 26 applications where 89.8% challenge memory through cross-temporal and cross-spatial information retention; (3) **MemGUI-Eval**, an automated evaluation pipeline with novel *Progressive Scrutiny* and 7 hierarchical metrics for memory fidelity and learning effectiveness; and (4) comprehensive assessment revealing significant memory deficits across all evaluated agents. Our experiments expose 4-10× performance gaps between memory-intensive and standard tasks, demonstrate the potential of explicit long-term memory mechanisms, and identify 7 distinct failure modes through systematic analysis. MemGUI-Bench establishes crucial empirical baselines for developing more capable and human-like GUI agents. Code and results: `https://anonymous.4open.science/r/MemGUI-Bench-Anonymous`.

## 1 Introduction

Large Multimodal Models have enabled autonomous **mobile GUI agents** capable of operating mobile devices (Chen et al., 2024; Rawles et al., 2024). While current agents show promise in basic tasks (Lu et al., 2024; Chai et al., 2025), they struggle with memory-intensive scenarios fundamental to effective mobile usage.

At the heart of this limitation lies **memory**. Human mobile expertise depends on memory mechanisms enabling information retention across temporal boundaries (verification codes, product details) and spatial boundaries (transferring information between applications), serving as the foundation for task success and skill development.

The mobile GUI agent community has recognized this imperative, leading to a proliferation of memory-enhanced architectures (Wang et al., 2025; Agashe et al., 2025; Wang et al., 2024a). However, this growing ecosystem of memory implementations reveals a critical evaluation gap: **the absence of standardized, comprehensive assessment frameworks for memory capabilities**. Existing benchmarks systematically undervalue memory requirements and fail to capture the nuanced cognitive demands of real-world mobile interactions.

Current evaluation platforms suffer from three fundamental limitations: **task design inadequacy** (only 5.2-11.8% memory-related tasks), **evaluation protocol limitations** (no multi-attempt `pass@k` protocols for long-term learning), and **judgment methodology constraints** (scalability and accuracy issues with existing approaches). Detailed analysis is in Appendix A.1.

To address these critical evaluation gaps, we introduce **MemGUI-Bench**, the the most comprehensive, memory-centric benchmark with pass@k and a staged LLM-as-judge evaluator. As illustrated in Figure 1, MemGUI-Bench establishes new standards for memory-centric evaluation through 4 key contributions:

- **Systematic Memory Taxonomy.** We establish a comprehensive taxonomy distinguishing short-term memory (temporary information buffering) and long-term memory (cross-session learning), with analysis of 11 agents identifying 5 distinct architectures (Section 2).

- **Memory-Centric Benchmarking Environment.** We contribute 128 tasks across 26 applications where 89.8% challenge memory through cross-temporal and cross-spatial information retention. Our snapshot-based framework supports `pass@1` and `pass@k` evaluation protocols (Section 3).

- **Automated Evaluation Pipeline.** We introduce `MemGUI-Eval` with novel *Progressive Scrutiny* across 3 stages and 7 hierarchical metrics for memory fidelity, learning effectiveness, and execution efficiency (Section 4).

- **Comprehensive Assessment of 11 Agents.** Our evaluation reveals significant memory deficits across all systems, establishes empirical baselines, and characterizes 7 distinct failure modes (Section 5 and Appendix A.8).

MemGUI-Bench reveals substantial performance gaps (4-10× disparity between memory-intensive and standard tasks) and demonstrates the transformative potential of explicit long-term memory mechanisms. All contributions are publicly available to advance memory-enhanced mobile automation research.

## 2 MEMORY IN MOBILE GUI AGENTS

Inspired by human cognition, we establish a comprehensive taxonomy of memory capabilities for mobile GUI agents. When humans interact with mobile interfaces, they naturally employ sophisticated memory mechanisms that enable intelligent and efficient task completion across diverse scenarios.

**Defining Memory for Mobile GUI Agents.** We define memory for mobile GUI agents as *the ability to retain, process, and utilize both contextual information within tasks and experiential knowledge across tasks to enhance decision-making and task performance over time*. This capability manifests in two fundamental forms, namely short-term (in-session) memory and long-term (cross-session) memory, consistent with the terminology adopted in recent LLM-agent memory research (Wu et al., 2024; Maharana et al., 2024; Zhong et al., 2024).

**Short-term (in-session) memory** refers to the agent's ability to temporarily retain and utilize contextual information during task execution, enabling coherent decision-making across sequential interaction steps. This capability allows agents to maintain awareness of previous actions, intermediate results, and relevant UI state changes throughout a task session. Memory-intensive tasks,

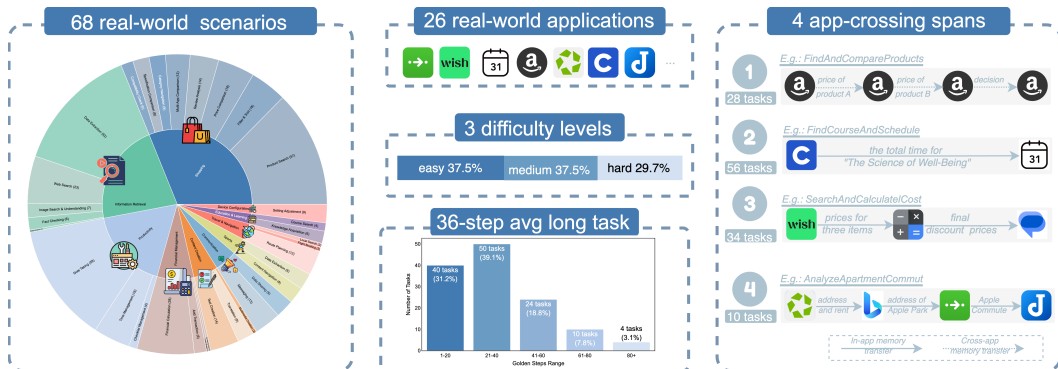

Figure 2: Statistical overview of the MemGUI-Bench task suite.

such as cross-application information transfer or multi-step data collection scenarios, pose significantly greater challenges to short-term memory compared to conventional GUI tasks by requiring agents to extract, retain, and accurately recall specific information units across extended interaction sequences.

**Long-term (cross-session) memory** accumulates experience from each interaction, whether successful or failed, forming reusable skills and knowledge. When agents encounter unfamiliar applications, they may initially make suboptimal decisions, but lessons learned from failures, combined with successful operation patterns, ultimately shape their proficiency with software. This memory is persistent, transferable, and aims to improve long-term efficiency across tasks.

Based on our comprehensive analysis of 11 prominent mobile GUI agents, we identify 5 distinct short-term memory architectures and 2 main categories of long-term memory implementations. Detailed technical implementations and comparative analysis of these memory mechanisms are provided in Appendix A.3.

## 3 MEMORY-CENTRIC BENCHMARKING ENVIRONMENT

Creating a robust benchmark for agent memory requires two key components: a challenging set of tasks that specifically target memory capabilities, and a standardized, efficient environment to execute these tasks. This section details both pillars of our contribution: the memory-centric task suite and the snapshot-based, plug-and-play framework that together form our unified benchmarking environment.

### 3.1 MEMORY-INTENSIVE TASK SUITE DESIGN

MemGUI-Bench comprises 128 carefully designed tasks across 26 real-world applications, spanning 4 different app-crossing complexities to systematically evaluate mobile GUI agents' memory capabilities. Our task suite statistics, illustrated in Figure 2, demonstrate a comprehensive distribution: tasks range from 3 to 160 golden steps (average 36.2), with 78.1% requiring cross-application information transfer, and balanced coverage across three difficulty levels (37.5% easy, 32.8% medium, 29.7% hard). This design reflects realistic user interaction patterns while providing focused evaluation of memory mechanisms in mobile GUI environments.

**Task Design Principles.** We designed 115 memory-intensive tasks alongside 13 standard tasks to systematically evaluate mobile GUI agents' memory capabilities. Our memory-intensive tasks require agents to extract, retain, and accurately recall specific information units across extended interaction sequences, such as retaining product prices for cross-application comparison or maintaining intermediate results across multiple steps. The 13 standard tasks serve as baseline benchmarks for computing the Memory-Task Proficiency Ratio (MTPR) and support long-term memory assessment through our `pass@k` evaluation protocol.

Figure 3: The unified architecture of MemGUI-Bench's snapshot-based plug-and-play framework.

**Cross-Application Information Transfer.** Our tasks implement diverse information transfer patterns ranging from single-app scenarios to complex four-app workflows. For example, *AnalyzeApartmentCommute* requires extracting apartment details from Apartments.com, searching company addresses via Bing, calculating commute times through Citymapper, and recording analysis in Joplin. This hierarchical complexity ensures comprehensive evaluation of memory capabilities across different spatial and temporal scales.

**Long-Term Learning Support.** To enable long-term memory evaluation, the 128 tasks are organized into 64 mirror task pairs with similar application combinations and cognitive demands but distinct specific requirements. This design supports systematic assessment of cross-task learning, where agents can transfer knowledge and strategies from earlier attempts to improve performance on related tasks.

Detailed design specifications, including application selection strategies, task characteristics, and information retention pathways, are provided in Appendix A.4. The complete task suite, presented in Table 8, represents the result of extensive development and validation to ensure the benchmark's reliability for systematic evaluation of mobile GUI agents' memory capabilities.

## 3.2 A Snapshot-Based Plug-and-Play Framework

We developed a comprehensive snapshot-based plug-and-play framework that enables efficient, scalable, and reproducible evaluation of GUI agents while providing robust support for long-term memory assessment through multi-attempt protocols. As illustrated in Figure 3, our framework addresses the critical challenges of environment consistency, agent diversity, and parallel execution that are essential for systematic memory evaluation.

**Evaluation Pipeline.** Our evaluation pipeline follows a systematic five-stage process that ensures reliable assessment across multiple attempts. (1) **Task Dispatch and Unified Scheduling**: Tasks are distributed through a centralized scheduling system that manages experiment queuing and resource allocation. (2) **Agent Task Reception**: GUI agents receive task specifications through our unified interface, which abstracts implementation details and provides consistent task formatting. (3) **Environment Interaction**: Agents interact with Android emulators by reading observational information (screenshots, UI hierarchies) and executing actions (taps, swipes, text input). (4) **Automated Evaluation**: Screenshots and agent decisions are continuously passed to MemGUI-Eval for real-time assessment of task progress and completion. (5) **Multi-Attempt Management**: If a task fails or reaches maximum step limits, the system automatically triggers environment reset and initiates retry attempts up to the configured limit (default $k = 3$ for `pass@k` evaluation), enabling systematic assessment of long-term learning capabilities.

**Key Framework Features.** Our framework provides three distinctive advantages over existing approaches: (1) **Scalable Parallel Execution**: Through sophisticated emulator management and port-based isolation, enabling concurrent evaluation of multiple agents without interference. (2) **Rapid Environment Recovery**: Snapshot-based approach enables instant environment reset, contrasting

with manual reset requirements in existing benchmarks. (3) **Native Long-Term Memory Support**: Built-in `pass@k` protocol and persistent agent state management across multiple attempts, a capability absent in existing benchmarks that focus exclusively on single-attempt evaluation.

Comprehensive technical specifications for the framework architecture, including parallel implementation details, multi-attempt mechanisms, agent integration protocols, and comparative analysis with existing approaches, are provided in Appendix A.5.

## 4 AN AUTOMATED EVALUATION PIPELINE WITH MEMORY-SPECIFIC METRICS

Evaluating memory-intensive tasks poses a significant challenge that demands innovation in both evaluation metrics and the judgment process itself. We address this by proposing a comprehensive, automated evaluation pipeline. This pipeline integrates a novel set of hierarchical metrics designed to quantify memory capabilities with `MemGUI-Eval`, a sophisticated arbiter that ensures accurate and efficient judgment.

### 4.1 MEMORY-SPECIALIZED METRICS WITH HIERARCHICAL ASSESSMENT

To capture the nuances of agent memory capabilities, we introduce a hierarchical framework with 7 specialized metrics across three dimensions: short-term memory fidelity, long-term learning capabilities, and execution efficiency.

**Short-Term Memory Assessment (`pass@1`).** We evaluate agents' memory fidelity through three complementary metrics: (1) *Overall Success Rate (SR)* as baseline performance measurement. (2) *Information Retention Rate (IRR)* as our core memory fidelity metric, quantifying the proportion of required information units that agents correctly recall and utilize. (3) *Memory-Task Proficiency Ratio (MTPR)* isolating memory-specific capabilities by comparing performance on memory-intensive versus standard tasks.

**Long-Term Memory Assessment (`pass@k`).** We quantify cross-session learning capabilities through two metrics: (1) *Multi-Attempt Success Rate (pass@k SR)* measuring agents' ability to succeed within $k$ trials through experience accumulation. (2) *Failure Recovery Rate (FRR)* targeting rapid learning from failure using harmonic decay weighting to reward faster recovery.

**Execution Efficiency Assessment (`pass@1 and pass@k`).** We include three efficiency indicators: (1) *Average Step Ratio* measuring path efficiency compared to golden standards. (2) *Average Time Per Step* quantifying computational overhead. (3) *Average Cost Per Step* evaluating economic efficiency of memory mechanisms.

Comprehensive mathematical definitions, computational procedures, and detailed metric analysis are provided in Appendix A.6.

### 4.2 MEMGUI-EVAL: A PROGRESSIVE SCRUTINY EVALUATOR

To overcome the limitations of existing evaluation methodologies—from rigid rule-based matching to inefficient "LLM-as-Judge" approaches that overwhelm models with complete trajectories—we developed `MemGUI-Eval`, a sophisticated evaluation arbiter designed specifically for memory-intensive tasks. As illustrated in Figure 4, it employs a novel "Progressive Scrutiny" pipeline that mimics efficient human expert verification: starting with minimal, high-efficiency evidence and progressively deepening analysis only when necessary, thereby achieving optimal cost-accuracy balance.

**Stage 1: Cost-Effective Triage.** This stage rapidly processes straightforward successful cases to dramatically reduce evaluation costs. The *Triage Judge* receives minimal evidence: task goal description, raw action logs (e.g., `CLICK`, `TYPE`), and the final three screenshots of the trajectory. Critically, this specialized agent adopts an extremely conservative strategy, concluding "success"

only when the limited evidence irrefutably demonstrates that all task requirements have been satisfied. Any case with ambiguity advances to the next stage, ensuring high precision while maximizing efficiency for clear-cut scenarios (see Figure 18 for a concrete example).

**Stage 2: Full Semantic Analysis.** When initial triage proves inconclusive, the system conducts comprehensive semantic analysis with enriched evidence. The framework first automatically generates detailed textual descriptions (`action_description` and `ui_description`) for every step in the trajectory using the *Step Descriptor*, a specialized agent that analyzes before-and-after action panels to create semantic representations of each interaction. The *Semantic Judge* then synthesizes the complete task goal, this rich step-by-step semantic context, and the same final three screenshots to make an informed judgment. Critically, the system includes explicit warnings about potential incompleteness in automatically generated text descriptions, requiring mandatory verification that all task-critical information is present in either the textual descriptions or visual evidence. For failed memory tasks involving multiple information units, the Semantic Judge ad-

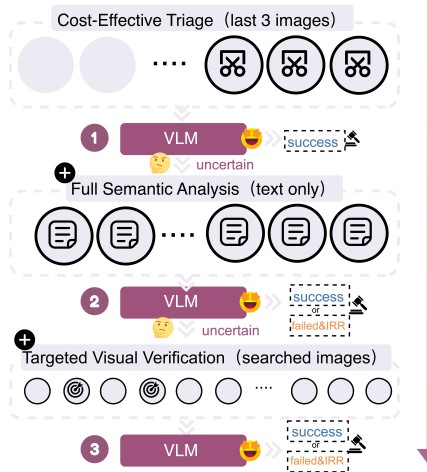

Figure 4: MemGUI-Eval's three-stage progressive scrutiny pipeline.

ditionally triggers the *IRR Analyzer* to compute an Information Retention Rate (IRR) and quantify the degree of memory failure—for instance, distinguishing an agent that correctly recalls 2 out of 3 required news headlines (see Figure 20 for memory failure analysis). When definitive judgment remains elusive despite this enriched context, the Semantic Judge must return a `required_steps` list, explicitly specifying which historical screenshots are essential for final adjudication (see Figure 19 for a successful semantic analysis case).

**Stage 3: Targeted Visual Verification.** This final stage represents our core innovation compared to traditional VLM evaluation methods: rather than overwhelming the model with all historical screenshots, we provide precisely the visual evidence it actively requested. The *Visual Judge* receives all textual evidence from Stage 2 plus a new composite image created by stitching together the specific historical screenshots identified in the `required_steps` list. This targeted approach eliminates information overload while ensuring the Visual Judge has exactly the visual evidence needed for high-fidelity judgment (see Figure 21 for successful visual verification). The system enforces strict verification requirements, mandating that any missing critical information in both textual descriptions and provided screenshots results in task failure, preventing inference or guesswork. The Visual Judge is required to make a definitive binary decision (success or failure) and, for failed memory tasks, triggers the *IRR Analyzer* to compute the final IRR based on all available visual and textual evidence (see Figure 22 for visual verification with failure determination). This progressive scrutiny approach maintains complete automation while ensuring reliable evaluation of complex memory-intensive tasks. Concrete examples illustrating each stage are provided in Appendix A.10.

### 4.3 Validation of the Evaluation Pipeline

To establish the trustworthiness of `MemGUI-Eval`, we conducted comprehensive validation experiments across two datasets: 26 SPA-Bench tasks (78 trajectories) for cross-benchmark comparison and 128 MemGUI-Bench tasks (256 trajectories) for memory-intensive evaluation. We tested three model configurations (`M1`, `M2`, `M3`) against baseline methods with human expert annotations as ground truth.

As detailed in Table 1, `MemGUI-Eval` demonstrates superior accuracy and cost-effectiveness across all configurations. Our `M1` configuration achieves near-perfect performance (99.0% F1-score on SPA-Bench), significantly outperforming baselines. The `M2` configuration provides optimal balance with 95.9% F1-score at reduced cost. Notably, while baseline methods struggle with cross-app

complexity (40-61.5% F1-score), `MemGUI-Eval` maintains exceptional performance (94.1-100% F1-score) across all task types.

**Configuration Selection.** To balance evaluation quality with computational efficiency, we adopt the `M2` configuration (Gemini 2.5 Flash for Step Descriptor, Gemini 2.5 Pro for judgment agents) for all subsequent experiments. This configuration achieves $> 95\%$ F1-score while maintaining cost-effectiveness at \$0.031 per trajectory, compared to \$0.055 for the most accurate `M1` configuration (all Pro) and \$0.018 for the most economical `M3` (all Flash). The validation demonstrates MemGUI-Eval's superiority

Table 1: Validation of MemGUI-Eval performance across different scenarios.

| Evaluator | Config. | Accuracy Metrics (%) | | | Efficiency |
|---|---|---|---|---|---|
| | | F1↑ | Prec.↑ | Recall↑ | Cost (\$)↓ |
| PART A: SPA-BENCH TRAJECTORIES (N=78) | | | | | |
| MemGUI-Eval (Ours) | M1 | **99.0** | **100.0** | **98.0** | 0.064 |
| | M2 | 95.9 | 95.9 | 95.9 | 0.028 |
| | M3 | 93.6 | 97.8 | 89.8 | **0.020** |
| SPA-Bench (Baseline) | G1 | 88.2 | 93.2 | 83.7 | 0.038 |
| | G2 | 81.4 | 94.6 | 71.4 | 0.027 |
| | G3 | 80.9 | 90.0 | 73.5 | 0.103 |
| PART B: MEMGUI-BENCH TRAJECTORIES (N=256) | | | | | |
| MemGUI-Eval (Ours) | M1 | **93.1** | **92.4** | **93.8** | 0.213 |
| | M2 | 81.2 | 82.5 | 80.0 | 0.070 |
| | M3 | 78.4 | 81.7 | 75.4 | **0.060** |

over baseline methods across diverse task complexities: on SPA-Bench trajectories, our `M2` configuration achieves 95.9% F1-score versus 92.5% for the best baseline, with the advantage becoming even more pronounced for cross-app tasks where traditional evaluators struggle (40-61.5% F1-score) while MemGUI-Eval maintains exceptional performance (94.1-100% F1-score). The consistent high performance across both SPA-Bench and MemGUI-Bench datasets (93.1-99.0% F1-score for `M1`, 77.9-95.9% for `M2`) validates the generalizability of our progressive scrutiny approach beyond our specific benchmark domain, establishing confidence in our evaluation methodology for systematic memory assessment of mobile GUI agents.

Comprehensive experimental details, model configurations, human annotation procedures, and detailed performance breakdowns are provided in Appendix A.7.

## 5 BENCHMARKING GUI AGENT BASELINES

In this section, we present a comprehensive evaluation of 11 leading GUI agents on MemGUI-Bench. Our goal is to empirically assess the current state of memory capabilities in SOTA models and validate our evaluation pipeline.

### 5.1 EXPERIMENTAL SETUP

We evaluate 11 prominent GUI agents spanning diverse architectural approaches and memory mechanisms, including 2 agents with explicit long-term memory capabilities and 9 without such mechanisms. Each of the 128 tasks is executed up to a maximum of $k = 3$ times on Android simulators, allowing agents with long-term memory modules to learn from previous attempts. Results are automatically assessed by our evaluation pipeline as described in Section 4. Detailed implementation specifications and deployment configurations for each agent are provided in Appendix A.2.

### 5.2 OVERALL PERFORMANCE ON MEMGUI-BENCH

Table 2 presents the main leaderboard of agent performance on MemGUI-Bench, summarizing success rates across different task difficulties for both short-term (`pass@1`) and long-term (`pass@3`) memory evaluations.

The results reveal striking performance patterns that highlight the current state of memory capabilities in GUI agents. M3A achieves the highest single-attempt success rate (32.8%), while Agent-S2 demonstrates exceptional learning potential with the highest multi-attempt performance (49.2%). Memory-equipped agents consistently outperform those without dedicated memory mechanisms

Table 2: Performance comparison of Mobile GUI agents on MemGUI-Bench.

| Agent | Short-Term Memory (`pass@1`) | | | | Long-Term Memory (`pass@3`) | | | |
|---|---|---|---|---|---|---|---|---|
| | Easy | Med | Hard | Overall | Easy | Med | Hard | Overall |
| AGENTIC WORKFLOW | | | | | | | | |
| Agent-S2 | **41.7** | 19.0 | 18.4 | 27.3 | **64.6** | 42.9 | 36.8 | **49.2** |
| Mobile-Agent-E | 12.5 | 2.4 | 0.0 | 5.5 | 22.9 | 2.4 | 2.6 | 10.2 |
| T3A | 31.2 | 16.7 | 18.4 | 22.7 | 45.8 | 45.2 | 34.2 | 42.2 |
| M3A | 39.6 | **35.7** | **21.1** | **32.8** | 47.9 | **50.0** | 44.7 | 47.7 |
| Mobile-Agent-V2 | 8.3 | 0.0 | 0.0 | 3.1 | 10.4 | 0.0 | 0.0 | 3.9 |
| SeeAct | 6.2 | 0.0 | 0.0 | 2.3 | 12.5 | 2.4 | 0.0 | 5.5 |
| AppAgent | 8.3 | 0.0 | 0.0 | 3.1 | 22.9 | 2.4 | 0.0 | 9.4 |
| AGENT-AS-A-MODEL | | | | | | | | |
| UI-Venus-7B | 14.6 | 0.0 | 0.0 | 5.5 | 20.8 | 0.0 | 0.0 | 7.8 |
| UI-TARS-1.5-7B | 8.3 | 0.0 | 0.0 | 3.1 | 16.7 | 0.0 | 0.0 | 6.2 |
| GUI-Owl-7B | 14.6 | 0.0 | 2.6 | 6.2 | 22.9 | 2.4 | 2.6 | 10.2 |
| CogAgent | 0.0 | 0.0 | 0.0 | 0.0 | 0.0 | 0.0 | 0.0 | 0.0 |

across all difficulty levels. The Agentic Workflow category substantially outperforms Agent-as-a-Model approaches, with framework-based agents achieving 22.7-32.8% single-attempt success rates compared to 0.0-6.2% for end-to-end models. Task difficulty analysis reveals significant scalability challenges, as performance drops dramatically from Easy (0.0-39.6%) to Hard tasks (0.0-21.1%), exposing fundamental limitations in current memory mechanisms for extended information retention requirements. Additionally, cross-application complexity analysis (detailed in Appendix A.9, Table 13) shows that performance degrades substantially as tasks involve more applications, with top agents experiencing 20-50 percentage point drops from single-app to four-app scenarios, confirming that cross-app information transfer poses severe memory challenges for current agent architectures.

### 5.3 SHORT-TERM MEMORY ANALYSIS

We conducted detailed analysis of short-term memory capabilities using single-attempt (`pass@1`) settings, examining Information Retention Rate (IRR), Memory-Task Proficiency Ratio (MTPR), and efficiency metrics across different memory mechanism types. Complete results are presented in Table 11 in Appendix A.9.

**Finding 1: Memory Agent Architectures Excel, Action-Thought Approaches Show Limitations.** Memory Agent frameworks consistently achieve superior memory fidelity compared to other approaches. M3A leads with 32.8% success rate and 39.3% IRR, while Agent-S2 achieves the highest IRR (39.5%) and MTPR (0.45), demonstrating that dedicated memory modules effectively preserve and utilize information across complex multi-step tasks. In contrast, agents employing Action-Thought mechanisms, including AppAgent (3.1% SR, 1.5% IRR) and UI-Venus-7B (5.5% SR, 2.6% IRR), show modest memory capabilities. While these approaches create textual traces of reasoning, they fail to establish robust information retention across extended task sequences, with low MTPR values (0.04-0.07) indicating that explicit reasoning alone is insufficient for complex memory-intensive tasks.

**Finding 2: Memory Tasks Expose Significant Capability Gaps.** The Memory-Task Proficiency Ratio reveals dramatic capability gaps that standard benchmarks fail to capture. While the best-performing agents (Agent-S2: MTPR 0.45, M3A: MTPR 0.41) show reasonable memory-specific performance, most agents exhibit MTPR values below 0.1, indicating fundamental limitations in handling memory-demanding scenarios. This 4-10× performance disparity between memory-intensive and standard tasks suggests that existing benchmarks significantly overestimate agent capabilities by not adequately testing memory requirements.

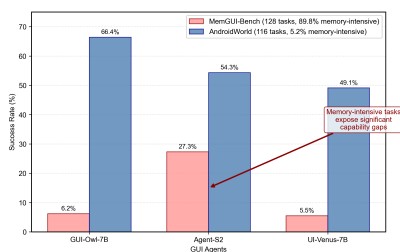

Figure 5: Performance comparison between MemGUI-Bench and Android-World.

Figure 5 compares agent performance between MemGUI-Bench and AndroidWorld, revealing dramatic performance drops—Agent-S2 from 54.3% to 27.3%, GUI-Owl-7B from 66.4% to 6.2%, and UI-Venus-7B from 49.1% to 5.5%—demonstrating that memory-intensive tasks expose fundamental capability limitations that standard benchmarks with minimal memory requirements systematically fail to detect.

**Finding 3: The Untapped Potential of Long-Context Understanding.** We found that Gemini-2.5-Pro's long-context capability can dramatically improve performance. By changing M3A from single-turn to multi-turn conversation (M3A-Multi-Turn), performance improved from 32.8% to 51.6% success rate (Table 3). However, context length limits matter. UI-TARS-1.5-7B uses multi-turn conversations but only keeps the last 5 turns due to context constraints. This leads to poor performance (3.1% SR), showing that truncated context cannot handle memory tasks effectively.

Table 3: Performance comparison between single-turn and multi-turn conversation modes using Gemini-2.5-Pro.

| Agent | pass@1 | pass@2 | pass@3 |
|---|---|---|---|
| M3A (Single-Turn) | 32.8% | 39.8% | 47.7% |
| M3A (Multi-Turn) | **51.6%** | **60.9%** | **68.0%** |
| UI-TARS-1.5-7B (Multi-Turn) | 3.1% | 4.7% | 6.2% |

### 5.4 LONG-TERM MEMORY ANALYSIS

We examined agents' ability to learn and improve across multiple attempts (`pass@3`), with particular focus on the Failure Recovery Rate (FRR) metric that measures how effectively agents learn from previous failures. Complete results are presented in Table 12 in Appendix A.9.

**Finding 4: The Trade-off between Frameworks and End-to-End Models.** Our evaluation reveals a fundamental trade-off between performance capabilities and computational efficiency that defines the current GUI agent ecosystem. Memory-enhanced frameworks excel at high performance costs—Agent-S2 and Mobile-Agent-E achieve superior performance but at substantial computational overhead (27.5 and 38.7 seconds per step, respectively). These frameworks demonstrate exceptional learning capabilities, with Agent-S2 showing 21.9 percentage point improvement (27.3% $\rightarrow$ 49.2%) versus M3A's 14.9 points (32.8% $\rightarrow$ 47.7%), validating that sophisticated memory architectures provide meaningful benefits for complex tasks. Framework-based approaches consistently dominate learning effectiveness, with the Agentic Workflow category substantially outperforming Agent-as-a-Model approaches. While GUI-Owl-7B leads the model-based category (6.2% to 10.2%, +4.0 points), its improvement pales compared to framework-based agents, highlighting the structural advantages of explicit memory modules and strategy adjustment mechanisms for effective cross-session learning.

**Finding 5: Long-Term Memory is Effective but Underutilized.** Explicit long-term memory mechanisms demonstrate remarkable effectiveness that remains largely underutilized in current GUI agent development. Agent-S2 exhibits exceptional learning capabilities with 21.5% FRR and 21.9 percentage point improvement across multiple attempts, while Mobile-Agent-E shows consistent learning patterns (+4.7 points improvement). The efficiency analysis reveals that long-term learning involves computational overhead but often provides favorable cost-benefit ratios, with agents completing tasks in fewer total steps across multiple attempts, partially offsetting their higher per-step computational costs. The FRR metric reveals distinct learning patterns: Agent-S2's exceptional 21.5% FRR indicates rapid failure analysis and strategy adjustment, while most agents without explicit memory systems show minimal FRR (0.8-4.4%), confirming that dedicated memory mechanisms are essential for efficient cross-session learning. Detailed pass@1, pass@2, and pass@3 results for each agent are provided in Appendix A.9.

## 6 CONCLUSION

This work introduces MemGUI-Bench, the the most comprehensive, memory-centric benchmark with pass@k and a staged LLM-as-judge evaluator. Through our evaluation of 11 state-of-the-art agents across 128 memory-intensive tasks, we reveal significant limitations in current systems, with performance gaps of 4-10× between memory-intensive and standard tasks. Our key contributions

include: (1) a systematic memory taxonomy distinguishing short-term and long-term memory mechanisms, (2) a specialized benchmarking environment with 89.8% memory-intensive tasks across 26 real-world applications, (3) MemGUI-Eval, a progressive scrutiny evaluation pipeline achieving 93-99% F1-score accuracy, and (4) comprehensive analysis identifying critical failure modes and architectural trade-offs. Our findings demonstrate that explicit long-term memory mechanisms provide 2-4× greater learning potential, while revealing fundamental inefficiencies where execution timeout accounts for 72.3% of failures. MemGUI-Bench establishes crucial empirical baselines and provides the research community with standardized tools to advance memory-enhanced mobile automation systems toward more capable, robust, and human-like GUI agents.

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

Table 4: Comprehensive comparison of MemGUI-Bench with existing smartphone agent benchmarks across three key dimensions: evaluation environment, evaluation pipeline, and agent support. ✓indicates feature supported; ✗indicates feature not supported.

| Benchmark | Evaluation Environment | | | | | Evaluation Pipeline | | | Agents Tested |
| --- | --- | --- | --- | --- | --- | --- | --- | --- | --- |
| | Memory Tasks | Cross-app Tasks | Total Tasks | 3rd-party Apps | Auto Reset | Long-term Memory | Auto Eval | Memory Metrics | |
| RULE-BASED EVALUATION PIPELINE | | | | | | | | | |
| AndroidArena | 22 | 22 | 221 | ✗ | ✗ | ✗ | ✗ | 1/4 | 1 |
| AndroidWorld | 6 | 6 | 116 | ✓ | ✓ | ✗ | ✗ | 1/1 | 3 |
| AndroidLab | 45 | 0 | 138 | ✓ | ✓ | ✗ | ✗ | 1/4 | 4 |
| LlamaTouch | 0 | 0 | 495 | ✓ | ✗ | ✗ | ✗ | 1/1 | 4 |
| B-MoCA | 0 | 0 | 60 | ✗ | ✗ | ✗ | ✗ | 1/1 | 3 |
| MobileAgentBench | 0 | 0 | 100 | ✗ | ✓ | ✗ | ✗ | 1/6 | 5 |
| LLM-AS-A-JUDGE EVALUATION PIPELINE | | | | | | | | | |
| A3 | 9 | 0 | 201 | ✓ | ✗ | ✗ | ✓ | 1/2 | 6 |
| SPA-Bench | 40 | 40 | 340 | ✓ | ✗ | ✗ | ✗ | 1/7 | 11 |
| **MemGUI-Bench** | **115** | **100** | **128** | ✓ | ✓ | ✓ | ✓ | **4/7** | **12** |

# A APPENDIX

You may include other additional sections here.

## A.1 RELATED WORK

The rapid development of Mobile GUI agents has been accompanied by the emergence of various benchmarks designed to evaluate their performance. These benchmarks can be broadly categorized into two types: static Mobile GUI agent datasets that provide instructions with corresponding operation trajectories (Lu et al., 2024; Li et al., 2024; Chai et al., 2024; Cheng et al., 2024), and dynamic benchmarks that provide task instructions along with corresponding evaluation environments and automated evaluators (Chai et al., 2025; Rawles et al., 2024; Chen et al., 2024). Dynamic Mobile GUI agent benchmarks have achieved consensus for evaluating agent performance in real-world scenarios due to their ability to assess agents in authentic, interactive environments.

However, as shown in Table 4, **none of the current Mobile GUI agent benchmarks systematically and comprehensively evaluate the memory capabilities of Mobile GUI agents**. This limitation stems from two fundamental issues in current benchmark design:

### A.1.1 EVALUATION ENVIRONMENT LIMITATIONS

Current benchmark environments face significant constraints that hinder comprehensive memory evaluation:

**Task Design Inadequacy.** The first issue lies in task design. Current benchmarks severely underrepresent memory-intensive tasks. As shown in Table 4, even the most memory-focused benchmarks like SPA-Bench (Chen et al., 2024) contain only 40 memory tasks out of 340 total tasks (11.8%), while many benchmarks like LlamaTouch (Zhang et al., 2024) and MobileAgentBench (Wang et al., 2024b) contain zero memory tasks. Similarly, cross-app tasks, which are essential for evaluating information retention across application boundaries, are limited or absent in most benchmarks.

This task design fundamentally cannot comprehensively evaluate Mobile GUI agents' memory capabilities. Human-like memory in GUI interaction requires two core abilities: *i)* short-term memory that creates temporary information buffers during complex tasks (e.g., remembering verification codes, product prices for comparison), and *ii)* long-term memory that accumulates experience from each interaction to form reusable skills. Current benchmark tasks are designed to minimize historical dependencies, with key decision information either always present in task instructions or

requiring only vague contextual awareness rather than specific visual information from historical UI observations.

**Environment Scalability Constraints.** The second limitation is evaluation environment scalability. While benchmarks like AndroidWorld (Rawles et al., 2024), AndroidLab (Xu et al., 2024), and MobileAgentBench (Wang et al., 2024b) support rapid environment reset for given tasks, they require manual script writing when adding new tasks, severely limiting scalability for memory-intensive evaluation scenarios.

### A.1.2 EVALUATION PIPELINE LIMITATIONS

Current evaluation pipelines face critical methodological challenges that impede accurate memory assessment:

**Success Rate Detection Issues.** As illustrated in Figure 6, existing approaches for success rate (SR) detection fall into two categories: rule-based methods and LLM-as-a-Judge methods. Rule-based methods include: *i*) state-based approaches that detect device status and execution logs after task completion (Xu et al., 2024; Rawles et al., 2024; Zhang et al., 2024), *ii*) action-based approaches that analyze agent execution actions (Chai et al., 2025), and *iii*) hybrid approaches like MobileAgentBench (Wang et al., 2024b). The common problem with rule-based approaches is that rule formulation requires expert knowledge and has poor scalability.

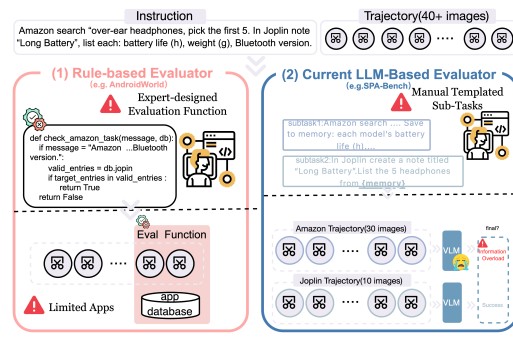

Figure 6: Limitations of existing evaluation approaches for memory-intensive GUI tasks.

LLM-as-a-Judge methods utilize large language models to evaluate agent execution trajectories based on predefined evaluation criteria. However, different approaches handle visual information differently, each with distinct limitations for memory-intensive tasks. SPA-Bench (Chen et al., 2024) provides all screenshots from long trajectories containing dozens of steps to VLMs at once. With such overwhelming visual information, there is no guarantee that VLMs can focus on critical early memory information points, leading to information overload and key detail omission risks. Additionally, cross-application evaluation relies on manual templates and is not fully automated. A3 (Chai et al., 2025) employs a sliding window approach for LLMs to check agent operation trajectories against a critical state pool, which is similarly unsuitable for context-dependent memory-intensive tasks due to the fragmented nature of information processing. The common problem with current LLM-based approaches is their inability to effectively and accurately evaluate memory-intensive tasks.

**Metrics Limitations.** Current evaluation metrics rely solely on SR to determine single-round task completion, lacking comprehensive assessment of short-term and long-term memory capabilities. No existing benchmark supports multi-attempt evaluation protocols (pass@k) necessary for assessing long-term memory and learning capabilities.

As demonstrated in Table 4, MemGUI-Bench systematically addresses these limitations through Memory-Intensive Task Suite Design(Section 3.1), A Snapshot-Based Plug-and-Play Framework(Section 3.2), and Progressive Scrutiny Evaluator(Section 4.2) with Memory-Specific Metrics(Section 4.1).

### A.2 DETAILS OF INTEGRATED AGENTS

We evaluate 11 prominent GUI agents, which can be categorized based on their memory mechanisms. **Agents with Long-Term Memory**: Mobile-Agent-E (Wang et al., 2025), Agent-S2 (Agashe et al., 2025). **Agents without Long-Term Memory**: T3A (Rawles et al., 2024), M3A (Rawles et al., 2024), UI-TARS-1.5-7B (Qin et al., 2025), GUI-Owl-7B (Ye et al., 2025), UI-Venus-7B (Gu et al., 2025), CogAgent (Hong et al., 2024), Mobile-Agent-V2(Wang et al., 2024a), SeeAct(Zheng et al., 2024) and AppAgent (Zhang et al., 2023). All agent workflows use Gemini 2.5 Pro in no-thinking

mode as their backbone model for fair comparison. All agent models are deployed on dual NVIDIA L40S-48G GPUs for experimental evaluation. CogAgent (Hong et al., 2024), deployment utilizes the scripts provided by SPA-Bench (Chen et al., 2024), while other models are deployed through the ms-swift infrastructure (Zhao et al., 2024).

Table 5: Details of integrated GUI agents evaluated in MemGUI-Bench.

| Agent | Agent Type | Core Model | UI Representation | Short-Term Memory Type | LTM |
|---|---|---|---|---|---|
| Agent-S2 (Agashe et al., 2025) | Workflow | Gemini-2.5-Pro | Screenshot | Memory Agent | ✓ |
| Mobile-Agent-E (Wang et al., 2025) | Workflow | Gemini-2.5-Pro | Screenshot | Memory Agent | ✓ |
| T3A | Workflow | Gemini-2.5-Pro | Screenshot+UI Tree | Memory Agent | ✗ |
| M3A (Rawles et al., 2024) | Workflow | Gemini-2.5-Pro | Screenshot+UI Tree | Memory Agent | ✗ |
| Mobile-Agent-V2 (Wang et al., 2024a) | Workflow | Gemini-2.5-Pro | Screenshot | Memory Agent | ✗ |
| SeeAct (Zheng et al., 2024) | Workflow | Gemini-2.5-Pro | UI Tree | Rule-based | ✗ |
| AppAgent (Zhang et al., 2023) | Workflow | Gemini-2.5-Pro | Screenshot+UI Tree | Action-Thought | ✗ |
| UI-Venus-7B (Gu et al., 2025) | Model | Fine-tuned Qwen2.5-VL-7B | Screenshot | Action-Thought | ✗ |
| UI-TARS-1.5-7B (Qin et al., 2025) | Model | Fine-tuned Qwen2.5-VL-7B | Screenshot | Multi-turn Context + Action-Thought | ✗ |
| GUI-Owl-7B | Model | Fine-tuned Qwen2.5-VL-7B | Screenshot | Action-Thought | ✗ |
| CogAgent (Hong et al., 2024) | Model | CogAgent-18B | Screenshot | No History | ✗ |

## A.3 DETAILED MEMORY IMPLEMENTATIONS

This section provides comprehensive technical analysis of memory implementations in mobile GUI agents, categorizing both short-term and long-term memory mechanisms observed across 11 prominent systems. Table 6 provides a concise overview of these memory mechanisms and their representative frameworks.

Table 6: Overview of memory implementations in mobile GUI agents.

| Memory Type & Implementation | Representative Agents |
|---|---|
| SHORT-TERM MEMORY | |
| No History | CogAgent (Hong et al., 2024) |
| Rule-based | SeeAct (Zheng et al., 2024), Autodroid (Wen et al., 2024) |
| Action-Thought | AppAgent (Zhang et al., 2023), UI-Venus (Gu et al., 2025), GUI-Owl (Ye et al., 2025), UI-TARS (Qin et al., 2025) |
| Multi-turn Context | UI-TARS (Qin et al., 2025) |
| Memory Agent | T3A (Rawles et al., 2024), M3A (Rawles et al., 2024), Agent-S2 (Agashe et al., 2025), Mobile-Agent-E (Wang et al., 2025), Mobile-Agent-V2 (Wang et al., 2024a) |
| LONG-TERM MEMORY | |
| Failure Learning | Agent-S2 (Agashe et al., 2025), Mobile-Agent-E (Wang et al., 2025) |
| Success Utilization | Mobile-Agent-E (Wang et al., 2025), Agent-S2 (Agashe et al., 2025) |

### A.3.1 SHORT-TERM MEMORY IMPLEMENTATIONS

**Memory Agent Architecture.** The most sophisticated approach employs dedicated memory modules to maintain structured context throughout task execution. Frameworks like T3A, M3A, Mobile-Agent-E, Agent-S2, and Mobile-Agent-V2 implement specialized memory agents that continuously summarize and update action history. This architecture typically involves a primary action agent for decision-making and a secondary memory agent for contextual management, creating comprehensive textual summaries that serve as memory context for subsequent decisions.

**Action-Thought Pattern.** Many agents implement explicit reasoning chains where each action is accompanied by corresponding thought processes. AppAgent, UI-Venus, and GUI-Owl exemplify this approach by outputting both actions and reasoning, creating structured action histories that capture not only what was done but why it was done. This textual action history serves as memory context for future decision-making steps.

**Multi-turn Context Management.** UI-TARS leverages multi-turn dialogue mechanisms, where each interaction round adds new observational information while maintaining conversation history. This approach treats memory as an evolving dialogue context, though it faces limitations due to context length constraints in practical deployments.

**Rule-based Context Aggregation.** SeeAct and Autodroid implement rule-based decision-making where each step involves selecting UI elements and combining them with corresponding actions. The resulting action sequences are concatenated to form contextual prompts for subsequent decisions, creating a structured but rigid form of memory representation.

**No Historical Context.** CogAgent represents the minimal memory approach, making decisions based solely on current observations and task instructions without maintaining any form of action history or memory context. This approach serves as a baseline for understanding the impact of memory mechanisms.

### A.3.2 LONG-TERM MEMORY IMPLEMENTATIONS

**Success-Based Learning.** Mobile-Agent-E and Agent-S2 implement systematic approaches to extract reusable knowledge from successful task executions. Mobile-Agent-E creates "shortcuts" from successful interaction patterns that can be directly invoked in similar future scenarios, while Agent-S2 distills successful experiences into actionable tips that guide future task execution. These approaches focus on transforming successful patterns into reusable procedural knowledge.

**Failure-Based Learning.** Both Agent-S2 and Mobile-Agent-E incorporate mechanisms to learn from failure experiences. They analyze failed task attempts to extract lessons about common pitfalls, incorrect interaction patterns, and environmental constraints. These failure insights are then used to prompt future task execution, helping agents avoid previously encountered errors and improve decision-making quality.

**Evolution and Trends.** The evolution of these memory mechanisms reflects increasing sophistication in contextual management and cross-session learning capabilities. Short-term memory implementations have progressed from basic action-thought approaches to specialized memory agent frameworks that maintain structured context throughout task execution. Long-term memory remains in early exploration stages, primarily focusing on learning from both successful and failed experiences to improve future task performance. The evolution from basic action-thought patterns to sophisticated memory agent architectures demonstrates the field's growing recognition of memory's critical role in mobile GUI automation. However, long-term memory implementations remain in early exploration stages, with most systems focusing on simple experience aggregation rather than more sophisticated learning mechanisms found in human cognition.

## A.4 DETAILS OF TASK SUITE DESIGN

This section provides comprehensive technical details for the memory-intensive task suite design presented in Section 3.1. The complete task suite specifications are presented in Table 8.

### A.4.1 APPLICATION SELECTION STRATEGY

Our application selection was guided by two complementary approaches to ensure both representativeness and experimental feasibility. First, we curated high-frequency, representative applications from established mobile GUI research (Lu et al., 2024; Chai et al., 2024), encompassing both Android native system applications (Settings, Files, Messages) and popular third-party applications (Amazon, Apartments.com, Citymapper). This selection ensures coverage of diverse interaction paradigms and real-world usage scenarios.

Second, we enforced two critical technical constraints for experimental reliability. *Emulator Compatibility*: Unlike applications such as X (formerly Twitter), Facebook, and Instagram that are incompatible with Android emulators and require physical devices for testing (Chen et al., 2024), our selected applications function reliably in emulated environments, enabling scalable and reproducible experiments. *Login-Free Operation*: To facilitate rapid environment reset through Android snapshots, we prioritized applications whose core functionalities are accessible without user authentication. This design choice eliminates the need for manual cleanup of user-generated data

(favorites, search history, etc.) and enables automated state recovery. Our analysis confirmed that Amazon, Apartments.com, and Citymapper provide comprehensive functionality in guest mode, satisfying our experimental requirements while maintaining task authenticity.

### A.4.2 Task Suite Characteristics

The benchmark provides structured metadata for each task, including *task_description* that captures authentic user intentions, and *golden_steps* determined by human annotators executing tasks in real environments. Based on these golden steps, we categorize tasks into three difficulty levels: Easy (1-20 steps), Medium (21-40 steps), and Hard (41+ steps), ensuring balanced evaluation across different complexity scales.

All task examples were manually annotated by human experts to ensure high quality and alignment with real-world usage patterns. The creation process followed a rigorous protocol:

- **Human Annotation:** Human experts manually crafted the task descriptions and executed the tasks on the target Android emulators to record the *golden_steps*. This ensures that every task is verifiable and executable within the specific app versions and emulator environment.
- **Cross-Validation:** We implemented a three-person cross-validation process. For each task, one expert designed the initial instruction and golden path. A second expert independently verified the task's executability and the optimal nature of the golden steps. A third expert resolved any discrepancies. This rigorous human-in-the-loop validation ensures the rationality, clarity, and correctness of all evaluation examples.

### A.4.3 Memory-Intensive Task Design

Building upon our definition of short-term memory as the agent's ability to temporarily retain and utilize contextual information during task execution (Section 2), we designed 115 memory-intensive tasks alongside 13 standard tasks. Memory-intensive tasks demand agents to create temporary information buffers during complex interactions, such as remembering verification codes for registration, retaining product prices for comparison across applications, or maintaining intermediate results across multiple interaction steps.

To ensure comprehensive evaluation across diverse real-world scenarios, we curated tasks spanning multiple categories. Table 7 presents the detailed hierarchical distribution of task categories, confirming balanced coverage across key domains such as Shopping (31.1%), Information Retrieval (21.9%), Productivity (17.7%), and Financial Management (7.9%).

The 13 standard tasks serve multiple evaluation purposes: they represent the contextual awareness component of short-term memory evaluation, provide baseline performance benchmarks for computing the Memory-Task Proficiency Ratio (MTPR), and support long-term memory assessment through our `pass@k` evaluation protocol. By comparing performance ratios between memory-intensive and standard tasks, we can objectively isolate and quantify agents' memory-specific capabilities.

### A.4.4 Information Retention Pathways

Our memory-intensive tasks implement diverse information transfer patterns across application boundaries. These patterns range from single-app scenarios (e.g., *FindAndCompareProducts*: comparing product ratings and prices within Amazon to identify the best value item) to complex four-app workflows (e.g., *AnalyzeApartmentCommute*: extracting apartment details from Apartments.com, searching company addresses via Bing, calculating commute times through Citymapper, and recording analysis in Joplin). This hierarchical complexity ensures comprehensive evaluation of memory capabilities across different spatial and temporal scales.

### A.4.5 Mirror Task Pairs for Long-Term Learning

To support long-term memory evaluation, the 128 tasks are organized into 64 mirror task pairs with similar application combinations and cognitive demands but distinct specific requirements. This design enables systematic assessment of cross-task learning, where agents can potentially transfer knowledge and strategies from earlier task attempts to improve performance on related tasks.

Table 7: Detailed distribution of task categories in MemGUI-Bench. The suite covers diverse domains including Commerce, Information Retrieval, Productivity, Finance, and Social, reflecting real-world mobile usage patterns. Counts represent category instances, as tasks may involve multiple categories.

| Main Category | Sub Category | Count | % within Main | Global % |
|---|---|---|---|---|
| Communication | Messaging | 13 | 59.1% | 2.9% |
| | Data Sharing | 9 | 40.9% | 2.0% |
| Content Creation | Text Creation | 14 | 58.3% | 3.1% |
| | Translation | 8 | 33.3% | 1.8% |
| | Multimedia Creation | 2 | 8.3% | 0.4% |
| Device Configuration | Setting Adjustment | 9 | 100.0% | 2.0% |
| Education & Learning | Knowledge Acquisition | 6 | 60.0% | 1.3% |
| | Course Search | 4 | 40.0% | 0.9% |
| Financial Management | Financial Calculation | 28 | 77.8% | 6.2% |
| | Add Transaction | 6 | 16.7% | 1.3% |
| | Create Budget | 1 | 2.8% | 0.2% |
| | Set Saving Goal | 1 | 2.8% | 0.2% |
| Information Retrieval | Data Extraction | 62 | 62.6% | 13.7% |
| | Web Search | 23 | 23.2% | 5.1% |
| | Image Search & Understanding | 7 | 7.1% | 1.5% |
| | Fact Checking | 6 | 6.1% | 1.3% |
| | Image Analysis | 1 | 1.0% | 0.2% |
| Productivity | Note Taking | 58 | 72.5% | 12.8% |
| | Time Management | 18 | 22.5% | 4.0% |
| | Checklist Management | 4 | 5.0% | 0.9% |
| Shopping | Product Search | 57 | 40.4% | 12.6% |
| | Filter & Sort | 18 | 12.8% | 4.0% |
| | Price Comparison | 18 | 12.8% | 4.0% |
| | Review Analysis | 14 | 9.9% | 3.1% |
| | Multi-App Comparison | 12 | 8.5% | 2.6% |
| | Category Navigation | 8 | 5.7% | 1.8% |
| | Specification Comparison | 8 | 5.7% | 1.8% |
| | Compatibility Check | 6 | 4.3% | 1.3% |
| Sports | Content Navigation | 8 | 50.0% | 1.8% |
| | Data Extraction | 8 | 50.0% | 1.8% |
| Travel & Navigation | Route Planning | 12 | 75.0% | 2.6% |
| | Local Search | 2 | 12.5% | 0.4% |
| | Flight Booking | 2 | 12.5% | 0.4% |

Table 8 provides the complete task suite with detailed specifications for each task, including task descriptions, applications involved, difficulty levels, and category classifications.

Table 8: Task details for MemGUI-Bench task suite.

| Description | App(s) | #Apps | X-App | Category | RUM | Steps | Diff. |
|---|---|---|---|---|---|---|---|
| Open the Amazon app, search for ""running shoes for men"", then filter for the brand ""ASICS"" and size ""10"". | ['Amazon'] | 1 | N | ['E-commerce: Product Search', 'E-commerce: Filter & Sort'] | N | 11 | 1 |
| Open the Amazon app, search for ""women's handbag"", then filter for the brand ""ALDO"" and color ""Black"". | ['Amazon'] | 1 | N | ['E-commerce: Product Search', 'E-commerce: Filter & Sort'] | N | 11 | 1 |
| Open the audio recorder app. Set the recording format to WAV, 48 kHz, Mono. Record an audio clip for more than 10 seconds, then stop the recording. Save the file with the name ""MyTestAudio"". | ['audio recorder'] | 1 | N | ['Content Creation: Multimedia Creation', 'Device Configuration: Setting Adjustment'] | N | 12 | 1 |
| Open the audio recorder app. Set the recording format to M4a, 8 kHz, 48kbps. Record an audio clip for more than 15 seconds, then stop the recording. Save the file with the name ""M4aTestAudio"". | ['audio recorder'] | 1 | N | ['Device Configuration: Setting Adjustment', 'Content Creation: Multimedia Creation'] | N | 12 | 1 |
| Open the BBC Sports app, find and tap on the Football category, proceed to 'Scores & Fixtures', and then perform a search for 'Real Madrid'. | ['BBC Sports'] | 1 | N | ['Sports: Content Navigation', 'Sports: Data Extraction'] | N | 8 | 1 |
| Open the BBC Sports app, navigate into the Football section, open the 'Scores & Fixtures' view, and then execute a search for 'Bayern Munich' | ['BBC Sports'] | 1 | N | ['Sports: Content Navigation', 'Sports: Data Extraction'] | N | 8 | 1 |

918

| Task | App | | | Category | | | |
|---|---|---|---|---|---|---|---|
| 919 920 Open the bluecoins app, record an expense transaction for an amount of 89.95, named 'New Sneakers', in the 'Clothing' category, and assign the label 'Personal' to it. | ['bluecoins'] | 1 | N | ['Financial Management: Add Transaction'] | N | 10 | 1 |
| 921 Open the bluecoins app, create an expense entry for a 'Summer Dress' with an amount of 65.00, categorized under 'Clothing', and add the 'Personal' label to the transaction. | ['bluecoins'] | 1 | N | ['Financial Management: Add Transaction'] | N | 10 | 1 |
| 922 923 924 925 Open the Clock app. Add two new alarms: one for 7:30 AM tomorrow labeled ""Morning Workout"", and another for 10:15 PM tomorrow labeled ""Read Book"". Then, navigate to the world clock and add ""Tokyo, Japan"" and ""London, UK"". Finally, switch to the stopwatch and let it run for at least 15 seconds before stopping (but not resetting) it. | ['Clock'] | 1 | N | ['Productivity: Time Management'] | N | 25 | 2 |
| 926 927 928 929 Open the Clock app. Navigate to the timer and set three timers simultaneously: one for 15 minutes labeled ""Laundry"", one for 45 minutes labeled ""Baking"", and one for 1 hour 30 minutes labeled ""Study Session"". After starting all three, go to the world clock, delete any existing cities, and add ""Sydney, Australia"". | ['Clock'] | 1 | N | ['Productivity: Time Management', 'Device Configuration: Setting Adjustment'] | Y | 28 | 2 |
| 930 931 932 Open the Clock app and add 'Beijing, China' and 'New-York, USA' to the world clock. By comparing their current times, find a suitable meeting time for tomorrow that falls between 8 AM and 10 PM in both cities. Then, set an alarm for this meeting, using the local time in Beijing as the reference. | ['Clock'] | 1 | N | ['Productivity: Time Management'] | Y | 13 | 1 |
| 933 934 935 Open the Clock app and add 'Beijing, China' and 'Buenos Aires, Argentina' to the world clock. By comparing their current times, find a suitable meeting time for tomorrow that falls between 9 AM and 11 PM in both cities. Then, set an alarm for this meeting, using the local time in Beijing as the reference. | ['Clock'] | 1 | N | ['Productivity: Time Management', 'Device Configuration: Setting Adjustment'] | Y | 15 | 1 |
| 936 937 938 939 In the Coursera app, search for courses offered by ""Stanford University"". For the first six courses in the search results, find and remember each course's star rating, total number of reviews, and number of available languages. Then, for all six courses, calculate a 'popularity score' (star rating * number of reviews * number of languages). Finally, navigate to the course page with the highest calculated popularity score. | ['coursera'] | 1 | N | ['Education & Learning: Course Search', 'Information Retrieval: Data Extraction', 'Education & Learning: Knowledge Acquisition'] | Y | 29 | 2 |
| 940 941 942 943 944 In the Coursera app, search for courses offered by the ""University of Michigan"". For the first six courses in the search results, find and remember each course's star rating, total number of reviews, and number of available languages. Then, for all six courses, calculate a 'popularity score' (star rating * number of reviews * number of languages). Finally, navigate to the course page with the highest calculated popularity score. | ['coursera'] | 1 | N | ['Education & Learning: Course Search', 'Information Retrieval: Data Extraction', 'E-commerce: Review Analysis'] | Y | 29 | 2 |
| 945 946 Open the joplin app, create a new note with the title ""Shopping List"" and the content ""Milk and bread"". | ['joplin'] | 1 | N | ['Productivity: Note Taking', 'Content Creation: Text Creation'] | N | 7 | 1 |
| 947 Open the joplin app, create a new note with the title ""Meeting Minutes"" and the content ""Discuss progress on Project A"". | ['joplin'] | 1 | N | ['Productivity: Note Taking', 'Content Creation: Text Creation'] | N | 7 | 1 |
| 948 949 950 In the Meesho app, find the first search result for three sarees: ""Banarasi Silk"", ""Kanjivaram Silk"", and ""Paithani Silk"". For each, remember its star rating and price. Then, navigate to the product page of the saree with the best value (highest rating-to-price ratio). | ['Meesho'] | 1 | N | ['E-commerce: Product Search', 'E-commerce: Price Comparison', 'Information Retrieval: Data Extraction'] | Y | 23 | 1 |
| 951 952 953 In the Meesho app, find the first search result for three kurtas: ""Chikankari Kurta"", ""Rayon Anarkali Kurta"", and ""Jaipuri Cotton Kurta"". For each, remember its star rating and price. Then, navigate to the product page of the kurta with the best value (highest rating-to-price ratio). | ['Meesho'] | 1 | N | ['E-commerce: Product Search', 'E-commerce: Price Comparison', 'E-commerce: Review Analysis'] | Y | 23 | 1 |
| 954 955 956 957 In the Meesho app, first navigate to the 'Kids & Toys' -¿ 'Toys & Games' category. From the first five results, find the item with the best value (highest rating-to-price ratio) and remember its name. Then, repeat this process for the 'Baby Gears' category. Finally, navigate to the product page of the toy with the better value between the two you identified. | ['Meesho'] | 1 | N | ['E-commerce: Category Navigation', 'E-commerce: Price Comparison', 'E-commerce: Review Analysis', 'Productivity: Note Taking'] | Y | 20 | 1 |
| 958 959 960 961 In the Meesho app, first navigate to the 'Home & Kitchen' -¿ 'Kitchen Tools' category. From the first five results, find the item with the best value (highest rating-to-price ratio) and remember its name. Then, repeat this process for the 'Storage & Organizers' category. Finally, navigate to the product page of the item with the better value between the two you identified. | ['Meesho'] | 1 | N | ['E-commerce: Category Navigation', 'E-commerce: Price Comparison', 'E-commerce: Review Analysis'] | Y | 20 | 1 |
| 962 963 964 965 In the Net-a-Porter app, find the following four items: the first ""black"" handbag, the first ""leather"" belt, the first ""cashmere"" scarf, and the first pair of ""white"" sneakers. For each item, remember its price and its designer or brand name (prioritize designer). Identify the most and least expensive items. If the designer/brand of the most expensive item comes first alphabetically, navigate to its page. Otherwise, navigate to the page of the least expensive item. | ['Net-a-Porter'] | 1 | N | ['E-commerce: Product Search', 'Information Retrieval: Data Extraction', 'E-commerce: Price Comparison'] | Y | 36 | 2 |
| 966 967 968 969 970 In the Net-a-Porter app, find the following four items: the first ""red"" dress, the first pair of ""gold"" sandals, the first ""green"" skirt, and the first ""white"" top. For each item, remember its price and its designer or brand name (prioritize designer). Identify the most and least expensive items. If they are by the same designer/brand, navigate to the page of the most expensive item. Otherwise, navigate to the page of the least expensive item. | ['Net-a-Porter'] | 1 | N | ['E-commerce: Product Search', 'E-commerce: Price Comparison'] | Y | 36 | 2 |
| 971 Open the Setting app and go to the 'Navigation mode' settings. Find the 'Circle to Search' feature and turn its toggle switch to the off position. | ['Setting'] | 1 | N | ['Device Configuration: Setting Adjustment'] | N | 6 | 1 |

| Task | Apps | | | Categories | | | |
|---|---|---|---|---|---|---|---|
| Open the Setting app and go to the 'Navigation mode' settings. Find the 'Circle to Search' feature and turn its toggle switch to the on position. | ['Setting'] | 1 | N | ['Device Configuration: Setting Adjustment'] | N | 3 | 1 |
| Open the Wikipedia app, search for 'English Wikipedia' and find the current number of articles it contains. Remember this number. Then, search for 'German Wikipedia' and find its current number of articles. Go to and stay on the page of the Wikipedia edition that has more articles | ['Wikipedia'] | 1 | N | ['Information Retrieval: Web Search', 'Information Retrieval: Data Extraction', 'Education & Learning: Knowledge Acquisition'] | Y | 13 | 1 |
| Open the Wikipedia app, first search for 'Beijing' and remember its population. Next, search for 'Shanghai' and remember its population. Compare the two, and then go to and remain on the Wikipedia page for the city with the larger population | ['Wikipedia'] | 1 | N | ['Information Retrieval: Web Search', 'Information Retrieval: Data Extraction', 'Education & Learning: Knowledge Acquisition'] | Y | 13 | 1 |
| In the Wish app, identify two items: the cheapest in 'Jewelry & watches' -¿ 'Fashion jewelry' (from top 6), and the cheapest in the 'Watches' sub-category (from top 6). Remember the number of reviews for both. Navigate to the page of whichever of these two items has more reviews. | ['Wish'] | 1 | N | ['E-commerce: Category Navigation', 'E-commerce: Price Comparison', 'E-commerce: Review Analysis'] | Y | 19 | 1 |
| In the Wish app, identify two items: the one with the most reviews in 'Office & tech' -¿ 'Parts & storage' (from top 6), and the one with the most reviews in the 'Hardware' sub-category (from top 6). Remember the price for both. Navigate to the page of whichever of these two items is cheaper. | ['Wish'] | 1 | N | ['E-commerce: Category Navigation', 'E-commerce: Review Analysis', 'E-commerce: Price Comparison'] | Y | 19 | 1 |
| Open the Amazon Kindle app and locate this month's 'Bestsellers' list. From the top 10, identify the three books with the highest number of customer ratings. For each, remember its title, price, and description. Open Joplin app and create a note titled ""Top Rated Books"" listing the title, price, and description for all three books. | ['Amazon Kindle', 'joplin'] | 2 | Y | ['E-commerce: Category Navigation', 'E-commerce: Review Analysis', 'Information Retrieval: Data Extraction', 'Productivity: Note Taking'] | Y | 37 | 2 |
| Open the Amazon Kindle app and navigate to 'Bestsellers'. Find the first three books that support audio narration. For each, remember its title, rating, and description. Open Joplin app and create a note titled ""Audiobooks Found"" listing the title, rating, and description for all three books. | ['Amazon Kindle', 'joplin'] | 2 | Y | ['E-commerce: Category Navigation', 'E-commerce: Filter & Sort', 'Information Retrieval: Data Extraction', 'Productivity: Note Taking'] | Y | 37 | 2 |
| In the Amazon Kindle app, find the title, customer rating, and page count for the first four books of the 'Dune' series. Then, in Joplin, create a note titled 'Dune Series Analysis' listing the books ordered by highest rating, showing all collected data for each. | ['Amazon Kindle', 'joplin'] | 2 | Y | ['E-commerce: Product Search', 'Information Retrieval: Data Extraction', 'E-commerce: Review Analysis', 'Productivity: Note Taking', 'E-commerce: Filter & Sort'] | Y | 29 | 2 |
| In the Amazon Kindle app, find the title, customer rating, and page count for the first four books of the 'A Song of Ice and Fire' series. Then, in Joplin, create a note titled 'A Song of Ice and Fire Series Analysis' listing the books ordered by highest rating, showing all collected data for each. | ['Amazon Kindle', 'joplin'] | 2 | Y | ['E-commerce: Product Search', 'Information Retrieval: Data Extraction', 'E-commerce: Review Analysis', 'Productivity: Note Taking'] | Y | 29 | 2 |
| Open Amazon app and search for these three products: 'Logitech C920', 'Razer Kiyo', 'Elgato Facecam'. For each, find and remember its price, star rating, and total number of reviews. In Joplin app, create a note titled ""Webcam Value Score"". For each camera, calculate a 'value score' using the formula: (star rating * number of reviews) / price. List each camera and its score, then state which has the highest score. | ['Amazon', 'joplin'] | 2 | Y | ['E-commerce: Product Search', 'Information Retrieval: Data Extraction', 'Productivity: Note Taking', 'Financial Management: Financial Calculation', 'E-commerce: Price Comparison'] | Y | 45 | 3 |
| Open Amazon app and search for these three products: 'Kindle Paperwhite', 'Kobo Libra 2', 'reMarkable 2'. For each, find and remember its price, screen size (in inches), and storage capacity (in GB). In Joplin app, create a note titled ""E-reader Value Score"". For each device, calculate a 'value score' using the formula: (screen size * storage capacity) / price. List each device and its score, then state which has the highest score. | ['Amazon', 'joplin'] | 2 | Y | ['E-commerce: Product Search', 'E-commerce: Price Comparison', 'E-commerce: Specification Comparison', 'Information Retrieval: Data Extraction', 'Productivity: Note Taking'] | Y | 45 | 3 |
| Open Amazon app, search for 'Anker 737 Power Bank'. Find its price, capacity (in mAh), and all output port types. Then, search for 'MacBook Air M2' and find its required charging port. Next, search for 'iPhone 15 Pro' and find its charging port. In Joplin app, note all the collected data and answer if the power bank can simultaneously charge both devices. | ['Amazon', 'joplin'] | 2 | Y | ['E-commerce: Product Search', 'E-commerce: Compatibility Check', 'Productivity: Note Taking'] | Y | 24 | 1 |
| Open Amazon app, search for 'Samsung T7 Shield SSD'. Find its price, storage capacity, and read/write speeds. Then search for 'PlayStation 5' and find its USB port specifications. Next, search for 'Xbox Series X' and find its USB port specifications. In Joplin app, note all collected data and answer if the SSD is fully compatible with both consoles' USB standards for external storage. | ['Amazon', 'joplin'] | 2 | Y | ['E-commerce: Product Search', 'E-commerce: Specification Comparison', 'E-commerce: Compatibility Check', 'Productivity: Note Taking'] | Y | 24 | 1 |
| Open the Amazon app. First, search for 'iPhone 15 Pro' and remember its screen size, battery capacity, and storage options. Second, search for 'Samsung Galaxy S24 Ultra' and remember the same three specifications. Third, search for 'Google Pixel 8 Pro' and remember the same three specifications. Finally, open the Joplin app, create a note titled 'Phone Spec Matrix', and list all nine specifications for the three phones. | ['Amazon', 'joplin'] | 2 | Y | ['E-commerce: Product Search', 'E-commerce: Specification Comparison', 'Information Retrieval: Data Extraction', 'Productivity: Note Taking'] | Y | 34 | 2 |

| Task | Apps | | | Categories | | | |
|---|---|---|---|---|---|---|---|
| Open the Amazon app. First, search for 'Sony WH-1000XM5' and remember its weight, battery life, and Bluetooth version. Second, search for 'Bose QuietComfort Ultra Headphones' and remember the same three specifications. Third, search for 'Sennheiser Momentum 4' and remember the same three specifications. Finally, open the Joplin app, create a note titled 'Headphone Spec Matrix', and list all nine specifications for the three headphones. | ['Amazon', 'joplin'] | 2 | Y | ['E-commerce: Product Search', 'E-commerce: Specification Comparison', 'E-commerce: Multi-App Comparison', 'Information Retrieval: Data Extraction', 'Productivity: Note Taking', 'Content Creation: Text Creation'] | Y | 27 | 2 |
| In the Amazon app, search for the 'Instant Pot Duo' and navigate to its customer reviews section. Read and remember the full original text of the top ten reviews listed. Do not use the copy function. Then, open the Joplin app and create a new note titled 'Instant Pot - Full Reviews'. In the note, accurately type out the full original text for all ten reviews you remembered. | ['Amazon', 'joplin'] | 2 | Y | ['E-commerce: Product Search', 'E-commerce: Review Analysis', 'Information Retrieval: Data Extraction', 'Productivity: Note Taking', 'Content Creation: Text Creation'] | Y | 55 | 3 |
| In the Amazon app, search for the 'Bose QuietComfort Ultra Headphones' and navigate to its customer reviews section. Read and remember the full original text of the top ten reviews listed. Do not use the copy function. Then, open the Joplin app and create a new note titled 'Bose QC - Full Reviews'. In the note, accurately type out the full original text for all ten reviews you remembered. | ['Amazon', 'joplin'] | 2 | Y | ['E-commerce: Product Search', 'E-commerce: Review Analysis', 'Information Retrieval: Data Extraction', 'Productivity: Note Taking', 'Content Creation: Text Creation'] | Y | 55 | 3 |
| Open the Amazon app, search for ""32GB DDR5 RAM"", filter for the ""Corsair"" brand, and remember the price and clock speed of the first item in the search results. Then, open the Bing app and search for ""ASUS ROG Strix Z790-E motherboard maximum supported memory speed"". Finally, confirm if the clock speed of that Corsair RAM is less than or equal to the motherboard's maximum supported speed. Directly answer with its price if it is compatible, or ""Not compatible"" if it isn't. | ['Amazon', 'bing'] | 2 | Y | ['E-commerce: Product Search', 'E-commerce: Filter & Sort', 'Information Retrieval: Web Search', 'E-commerce: Compatibility Check', 'Information Retrieval: Data Extraction'] | Y | 14 | 1 |
| Open the Amazon app, search for ""1TB External SSD"", filter for the ""Samsung"" brand, and remember the price and the read/write speed of the first item in the search results. Then, open the Bing app and search for ""PlayStation 5 external SSD speed requirement"". Finally, confirm if the SSD's speed meets or exceeds the PS5's requirement. Directly answer with its price if it is compatible, or ""Not compatible"" if it isn't. | ['Amazon', 'bing'] | 2 | Y | ['E-commerce: Product Search', 'E-commerce: Filter & Sort', 'Information Retrieval: Web Search', 'E-commerce: Compatibility Check', 'E-commerce: Multi-App Comparison', 'Information Retrieval: Data Extraction'] | Y | 17 | 1 |
| In the AP News app, find the three most recent articles from the 'U.S. NEWS' section and the three most recent from the 'World' section. For each of the six articles, open it, read the full text, and remember its title and main content. Then, in the Joplin app, create a note titled ""News Digest"". For each of the six articles, list its title followed by a 50-word summary of its content, grouped by section. | ['AP News', 'joplin'] | 2 | Y | ['Information Retrieval: Data Extraction', 'Education & Learning: Knowledge Acquisition', 'Productivity: Note Taking', 'Content Creation: Text Creation'] | Y | 95 | 3 |
| In the AP News app, find the three most recent articles from the 'Technology' section and the three most recent from the 'Business' section. For each of the six articles, open it, read the full text, and remember its title and main content. Then, in the Joplin app, create a note titled ""Tech & Business Digest"". For each of the six articles, list its title followed by a 50-word summary of its content, grouped by section. | ['AP News', 'joplin'] | 2 | Y | ['Information Retrieval: Data Extraction', 'Productivity: Note Taking', 'Content Creation: Text Creation'] | Y | 95 | 3 |
| In the Apartments.com app, search for listings in 'Austin, TX'. For the first ten results, enter each detail page and remember the address, monthly rent, and square footage. Then, open the Joplin app and create a note titled ""Austin Apartment Data"". In the note, list all ten apartments, ordered from the largest square footage to the smallest, including their address, rent, and square footage for each. | ['Apartments.com Rental Search', 'joplin'] | 2 | Y | ['E-commerce: Product Search', 'Information Retrieval: Data Extraction', 'Productivity: Note Taking'] | Y | 160 | 3 |
| In the Apartments.com app, search for listings in 'Denver, CO'. For the first ten results, enter each detail page and remember the address, monthly rent, and number of bedrooms. Then, open the Joplin app and create a note titled ""Denver Apartment Data"". In the note, list all ten apartments, ordered from the lowest rent to the highest, including their address, rent, and number of bedrooms for each. | ['Apartments.com Rental Search', 'joplin'] | 2 | Y | ['E-commerce: Product Search', 'Information Retrieval: Data Extraction', 'Productivity: Note Taking'] | Y | 160 | 3 |
| Open the Apartments.com app, search for listings in San Francisco, and filter for 'fitness center' and 'pool'. Then, go into the detail pages for the first three results and remember the address and phone number for each. Finally, open the Joplin app, create a new note, and record all collected addresses and phone numbers. | ['Apartments.com Rental Search', 'joplin'] | 2 | Y | ['Travel & Navigation: Local Search', 'E-commerce: Filter & Sort', 'Information Retrieval: Data Extraction', 'Productivity: Note Taking'] | Y | 17 | 1 |
| Open the Apartments.com app, search for listings in New York, NY, and filter for 'cat friendly' and 'in-unit washer'. Then, go into the detail pages for the first three results and remember the address and phone number for each. Finally, open the Joplin app, create a new note, and record all collected addresses and phone numbers. | ['Apartments.com Rental Search', 'joplin'] | 2 | Y | ['E-commerce: Product Search', 'E-commerce: Filter & Sort', 'Information Retrieval: Data Extraction', 'Productivity: Note Taking'] | Y | 19 | 1 |

| Task | Apps | # | Y | Categories | Y | N | N |
|---|---|---|---|---|---|---|---|
| In the AutoUncle app (UK), first search for a 'Ford Focus' and remember the price and mileage of the first non-sponsored result. Then, separately search for a 'Vauxhall Corsa' and remember the price and mileage of its first non-sponsored result. In the Calculator app, determine which car has a lower price-to-mileage ratio (price/mileage). Then, for that better-value car, calculate the loan amount needed for an 80% financing. | ['AutoUncle:Search used cars', 'Calculator'] | 2 | Y | ['E-commerce: Product Search', 'Information Retrieval: Data Extraction', 'E-commerce: Price Comparison', 'Financial Management: Financial Calculation'] | Y | 59 | 3 |
| In the AutoUncle app (Germany), first search for a 'Mercedes-Benz C-Class' and remember the price and mileage of the first non-sponsored result. Then, separately search for a 'BMW 3 Series' and remember the price and mileage of its first non-sponsored result. In the Calculator app, determine which car has a lower price-to-mileage ratio (price/mileage). Then, for that better-value car, calculate the loan amount needed for an 80% financing. | ['AutoUncle:Search used cars', 'Calculator'] | 2 | Y | ['E-commerce: Product Search', 'E-commerce: Multi-App Comparison', 'Information Retrieval: Data Extraction', 'Financial Management: Financial Calculation'] | Y | 59 | 3 |
| In the AutoUncle app (UK), find a 'Honda Civic', a 'Toyota Corolla', and a 'Mazda 3'. For each car, remember its engine size, fuel type, and price. In the Joplin app, create a note titled ""Compact Car Comparison"" and list all three cars with all three of their specs. | ['AutoUncle:Search used cars', 'joplin'] | 2 | Y | ['E-commerce: Product Search', 'E-commerce: Specification Comparison', 'Productivity: Note Taking'] | Y | 51 | 3 |
| In the AutoUncle app (Germany), find a 'BMW 3 Series', an 'Audi A4', and a 'Mercedes-Benz C-Class'. For each car, remember its mileage, transmission type, and price. In the Joplin app, create a note titled ""German Sedan Comparison"" and list all three cars with all three of their specs. | ['AutoUncle:Search used cars', 'joplin'] | 2 | Y | ['E-commerce: Product Search', 'E-commerce: Specification Comparison', 'E-commerce: Multi-App Comparison', 'Productivity: Note Taking'] | Y | 51 | 3 |
| In the AutoUncle app (Germany), search for 'Audi' with filters: after 2021, mileage below 50,000 km, automatic, petrol. From the results, find the most expensive car and the least expensive car. Remember the model and price for both. In the messages app, send ""Most Expensive: [Model] - [Price]. Least Expensive: [Model] - [Price]."" to +8613911112222. | ['AutoUncle:Search used cars', 'messages'] | 2 | Y | ['E-commerce: Product Search', 'E-commerce: Filter & Sort', 'E-commerce: Price Comparison', 'Communication: Messaging', 'Communication: Data Sharing'] | Y | 35 | 2 |
| In the AutoUncle app (UK), search for 'Land Rover' with filters: after 2020, mileage below 80,000 km, diesel. From the results, find the car with the highest mileage and the car with the lowest mileage. Remember the price and mileage for both. In the messages app, send ""Highest Mileage: [Mileage]km - [Price]. Lowest Mileage: [Mileage]km - [Price]."" to +8613933334444. | ['AutoUncle:Search used cars', 'messages'] | 2 | Y | ['E-commerce: Product Search', 'E-commerce: Filter & Sort', 'E-commerce: Specification Comparison', 'Information Retrieval: Data Extraction', 'Communication: Messaging', 'Communication: Data Sharing'] | Y | 35 | 2 |
| In Bing app, find the current exchange rates for USD to EUR and USD to GBP. Remember both rates. In the Calculator app, first calculate how much $1500 USD is in EUR. Then, calculate how much the resulting EUR amount is worth in GBP (this requires a second conversion step using the two initial rates). Directly answer with the final GBP amount. | ['bing', 'Calculator'] | 2 | Y | ['Information Retrieval: Fact Checking', 'Financial Management: Financial Calculation'] | Y | 35 | 2 |
| In Bing app, find the current stock prices for NVIDIA (NVDA) and Apple (AAPL). Remember both prices. In the Calculator app, first calculate the value of 50 NVDA shares. Then, calculate the value of 75 AAPL shares. Finally, calculate the total combined value of both holdings. Directly answer with the final combined value. | ['bing', 'Calculator'] | 2 | Y | ['Information Retrieval: Fact Checking', 'Information Retrieval: Data Extraction', 'Financial Management: Financial Calculation'] | Y | 35 | 2 |
| On the NASA APOD website found via Bing app, locate the three most recent daily pictures. For each of the three pictures, create a separate note in Joplin app. Each note must use the picture's title as its own title, contain a brief visual description, and have the corresponding image directly inserted or attached into the note's body. | ['bing', 'joplin'] | 2 | Y | ['Information Retrieval: Web Search', 'Information Retrieval: Data Extraction', 'Information Retrieval: Image Analysis', 'Productivity: Note Taking', 'Content Creation: Text Creation'] | Y | 51 | 3 |
| On the ""Smithsonian Magazine Photo of the Day"" page found via Bing app, locate the three most recent daily photos. For each of the three photos, create a separate note in Joplin app. Each note must use the photo's title as its own title, contain a brief description, and have the corresponding image directly inserted or attached into the note's body. | ['bing', 'joplin'] | 2 | Y | ['Information Retrieval: Image Search & Understanding', 'Productivity: Note Taking'] | Y | 51 | 3 |
| In Bing app, find the date for next year's Thanksgiving and for next year's Easter. Remember both dates. Open the N calendar app. Create an all-day event on the Thanksgiving date named ""Thanksgiving Dinner"". Then, create a second all-day event on the Easter date named ""Easter Egg Hunt"". | ['bing', 'N calendar'] | 2 | Y | ['Information Retrieval: Fact Checking', 'Productivity: Time Management'] | Y | 35 | 2 |
| In Bing app, find the date for the next leap day and for next year's Halloween. Remember both dates. Open the N calendar app. Create an all-day event on the leap day named ""Leap Day Fun"". Then, create a second all-day event on the Halloween date named ""Halloween Party"". | ['bing', 'N calendar'] | 2 | Y | ['Information Retrieval: Web Search', 'Information Retrieval: Fact Checking', 'Productivity: Time Management'] | Y | 35 | 2 |
| In the Bing app, search for the host cities and years of the next three Summer Olympics. Remember all three cities and years. Open the N Calendar app and create an all-day event on July 1st of the first Olympic year titled ""[City] Olympics"". Then, create two more all-day events on July 1st of the subsequent Olympic years with their appropriate titles. | ['bing', 'N calendar'] | 2 | Y | ['Information Retrieval: Web Search', 'Information Retrieval: Data Extraction', 'Productivity: Time Management'] | Y | 40 | 2 |

| | | | | | | | |
|---|---|---|---|---|---|---|---|
| In the Bing app, search for the host countries and years of the next three FIFA World Cups. Remember all three countries and years. Open the N Calendar app and create an all-day event on June 1st of the first World Cup year titled ""[Country] World Cup"". Then, create two more all-day events on June 1st of the subsequent World Cup years with their appropriate titles. | ['bing', 'N calendar'] | 2 | Y | ['Information Retrieval: Web Search', 'Information Retrieval: Data Extraction', 'Productivity: Time Management'] | Y | 40 | 2 |
| Open the bing app and perform an image search for 'Global Smartphone Shipments Market Share 2021'. From the image results, locate and carefully analyze the chart that specifically displays the data for Q3 2021. Identify the top three brands from this Q3 chart, and remember their names and their exact market share percentages. Finally, open the joplin app, create a new note titled 'Smartphone Market Share 2021 Q3', and list the top three brands with their corresponding percentages. | ['bing', 'joplin'] | 2 | Y | ['Information Retrieval: Image Search & Understanding', 'Information Retrieval: Data Extraction', 'Productivity: Note Taking'] | Y | 17 | 1 |
| Open the bing app and perform an image search for 'Global Vehicle Sales Trend by region November 2023'. Carefully analyze the first clear chart that appears in the search results. From this chart, identify the top three regions with the highest sales growth or volume, and remember the names of these regions and their corresponding data values (e.g., sales numbers or percentage growth). Finally, open the joplin app, create a new note titled 'Vehicle Sales Trend November 2023', and list the top three regions with their data | ['bing', 'joplin'] | 2 | Y | ['Information Retrieval: Image Search & Understanding', 'Information Retrieval: Data Extraction', 'Productivity: Note Taking'] | Y | 17 | 1 |
| In the Citymapper app, plan a route from ""Central Park, NYC"" to ""JFK Airport"". Find the time, cost, and number of transfers for four transport options: Public Transit, Driving, Taxi, and Bikeshare. In the Joplin app, create a note titled ""JFK Transit Analysis"", list the full data for all four options, then summarize which is the fastest, cheapest, and has the fewest transfers. | ['Citymapper', 'joplin'] | 2 | Y | ['Travel & Navigation: Route Planning', 'Information Retrieval: Data Extraction', 'Productivity: Note Taking'] | Y | 25 | 2 |
| In the Citymapper app, plan a route from ""Golden Gate Bridge, SF"" to ""SFO Airport"". Find the time, cost, and number of transfers for four transport options: Public Transit, Driving, Taxi, and Walking. In the Joplin app, create a note titled ""SFO Transit Analysis"", list the full data for all four options, then summarize which is the fastest, cheapest, and has the fewest transfers. | ['Citymapper', 'joplin'] | 2 | Y | ['Travel & Navigation: Route Planning', 'Information Retrieval: Data Extraction', 'Productivity: Note Taking'] | Y | 25 | 2 |
| In the Citymapper app, find the travel times for a two-leg journey in Washington D.C.: 1) The White House to the Lincoln Memorial, and 2) the Lincoln Memorial to the National Air and Space Museum. For each leg, find the times for Public Transit, Walking, and Taxi. Determine the fastest possible total travel time by combining the modes for each leg. Send a message to +8613811118888 stating this fastest route combination and the total time. | ['Citymapper', 'messages'] | 2 | Y | ['Travel & Navigation: Route Planning', 'Communication: Messaging'] | Y | 24 | 1 |
| In the Citymapper app, find the travel times for a two-leg journey in New York City: 1) Statue of Liberty to the Empire State Building, and 2) the Empire State Building to The Metropolitan Museum of Art. For each leg, find the times for Public Transit, Walking, and Taxi. Determine the fastest possible total travel time by combining the modes for each leg. Send a message to +8613822229999 stating this fastest route combination and the total time. | ['Citymapper', 'messages'] | 2 | Y | ['Travel & Navigation: Route Planning', 'Information Retrieval: Data Extraction', 'Communication: Messaging'] | Y | 24 | 1 |
| Open the Citymapper app, plan a public transport route from 'London Eye' to 'The British Museum', and remember the estimated travel time. Then open the N Calendar app and create an event for next Monday at 10 AM titled ""Visit British Museum"", setting the event duration to the travel time you remembered. | ['Citymapper', 'N calendar'] | 2 | Y | ['Travel & Navigation: Route Planning', 'Productivity: Time Management'] | Y | 40 | 2 |
| Open the Citymapper app, plan a walking route from 'Notre-Dame Cathedral' to 'Louvre Museum', and remember the estimated travel time. Then open the N calendar app and create an event for next Tuesday at 3 PM titled ""Trip to the Louvre"", setting the event duration to the travel time you remembered. | ['Citymapper', 'N calendar'] | 2 | Y | ['Travel & Navigation: Route Planning', 'Productivity: Time Management'] | Y | 40 | 2 |
| Open the Coursera app, search for the ""Financial Markets"" course, and find its total time to complete. Then, open the N Calendar app. Create a recurring event titled ""Study Finance"" for all 7 days of next week (Monday to Sunday). Set the duration for each daily event by dividing the course's total completion time equally across the seven days. | ['coursera', 'N calendar'] | 2 | Y | ['Education & Learning: Course Search', 'Information Retrieval: Data Extraction', 'Productivity: Time Management'] | Y | 40 | 2 |
| Open the Coursera app and find the total time required for ""The Science of Well-Being"". Then, open the N Calendar app to create a daily event, ""Study Finance"", from the 6th to the 15th of next month, starting at 2 PM. Calculate the daily duration by dividing the total course time by 10. | ['coursera', 'N calendar'] | 2 | Y | ['Education & Learning: Course Search', 'Information Retrieval: Data Extraction', 'Productivity: Time Management', 'Financial Management: Financial Calculation'] | Y | 40 | 2 |
| In the DeepL Translate app, translate the following paragraph into five languages: Spanish, Japanese, Russian, Arabic, and Portuguese: ""The project's quarterly review meeting is scheduled for next Monday. Key discussion topics will include the budget forecast for Q4, which is currently estimated at $1,250,000, and the initial user feedback analysis from the beta test group."" After remembering all five translations, open the messages app. Important: Do not use the copy-paste function for the translations. Send each translation as a separate message to a different recipient: the Spanish translation to +8613100001111, Japanese to +8613100002222, Russian to +8613100003333, Arabic to +8613100004444, and Portuguese to +8613100005555. | ['DeepL Translate', 'messages'] | 2 | Y | ['Content Creation: Translation', 'Communication: Messaging', 'Communication: Data Sharing'] | Y | 40 | 2 |

| Task | Apps | # | | Categories | | | |
|---|---|---|---|---|---|---|---|
| In the DeepL Translate app, translate the following paragraph into five languages: German, French, Korean, Hindi, and Italian: ""Please be advised that due to a system upgrade, network services will be unavailable from 11:00 PM on Friday until 5:00 AM on Saturday. The expected downtime is approximately 6 hours. We apologize for any inconvenience this may cause."" After remembering all five translations, open the messages app. Important: Do not use the copy-paste function for the translations. Send each translation as a separate message to a different recipient: the German translation to +8613200001111, French to +8613200002222, Korean to +8613200003333, Hindi to +8613200004444, and Italian to +8613200005555. | ['DeepL Translate', 'messages'] | 2 | Y | ['Content Creation: Translation', 'Communication: Messaging', 'Communication: Data Sharing'] | Y | 40 | 2 |
| In Net-a-Porter app, for the designer 'Isabel Marant', find the first ""New In"" item from each of the following four categories: Dresses, Bags, Shoes, and Accessories. For each of the four items, remember its price and primary material/color. Then, in Joplin app, create a single note titled 'Isabel Marant Collection' listing the details for all four items. | ['Net-a-Porter', 'joplin'] | 2 | Y | ['E-commerce: Product Search', 'E-commerce: Category Navigation', 'Productivity: Note Taking'] | Y | 65 | 3 |
| In Net-a-Porter app, for the designer 'Jimmy Choo', find the first item from each of the following four categories: Boots, Heels, Sandals, and Sneakers. For each of the four items, remember its price and primary material/color. Then, in Joplin app, create a single note titled 'Jimmy Choo Collection' listing the details for all four items. | ['Net-a-Porter', 'joplin'] | 2 | Y | ['E-commerce: Product Search', 'E-commerce: Category Navigation', 'Productivity: Note Taking'] | Y | 65 | 3 |
| Open the Setting app and go to 'Special app access'. First, view the apps with 'Wi-Fi control' permission and remember the list of app names. Next, view the apps allowed 'Picture-in-picture' access and remember that list of names. Finally, open the joplin app, create a note titled 'App Access Permissions', and list the names you remembered under two headings: 'Wi-Fi Control' and 'Picture-in-picture'. | ['Setting', 'joplin'] | 2 | Y | ['Information Retrieval: Data Extraction', 'Productivity: Note Taking'] | Y | 13 | 1 |
| Open the Setting app and go to 'Special app access'. First, check how many apps have 'Wi-Fi control' permission and remember the count. Then, check how many apps are allowed to 'Install unknown apps' and remember that count. Finally, open the joplin app and create a note titled 'App Permissions' that records both counts. | ['Setting', 'joplin'] | 2 | Y | ['Device Configuration: Setting Adjustment', 'Information Retrieval: Data Extraction', 'Productivity: Note Taking'] | Y | 15 | 1 |
| In the wikiHow app, sequentially find the main ingredients for the following five dishes: 1. Salad, 2. Spaghetti, 3. Fried Chicken, 4. Chocolate Cake, 5. Mashed Potatoes. After gathering the ingredients for all five dishes, open the Joplin app. Create a single note titled 'Dinner Shopping List'. In this note, create a comprehensive shopping list, grouping all collected ingredients by category (e.g., Produce, Dairy, Pantry, Meat). | ['wikiHow', 'joplin'] | 2 | Y | ['Information Retrieval: Data Extraction', 'Productivity: Note Taking', 'Productivity: Checklist Management'] | Y | 55 | 3 |
| In the wikiHow app, sequentially find the main ingredients for the following five items: 1. Snow Cones, 2. Egg Sandwiches, 3. Green Tea, 4. Feta Cheese (how to make), 5. Chicken Alfredo. After gathering all ingredients, open the Joplin app. Create a single note titled 'Snack & Lunch Plan'. In this note, first create a comprehensive shopping list, grouping all ingredients by category. Then, create a second heading named 'Preparation Order' and list the five items in a logical sequence for preparation. | ['wikiHow', 'joplin'] | 2 | Y | ['Information Retrieval: Data Extraction', 'Productivity: Note Taking', 'Productivity: Checklist Management'] | Y | 55 | 3 |
| In the Yahoo Sports app, find the three most recent news articles in the NBA section and the three most recent in the MLB section. For each of the six articles, read it and remember its title and main content. Then, in Joplin, create a note titled ""Sports News Summary"". For each of the six articles, write down its title followed by a 40-word summary of its content, grouped under 'NBA' and 'MLB' headings. | ['Yahoo Sports', 'joplin'] | 2 | Y | ['Sports: Content Navigation', 'Sports: Data Extraction', 'Productivity: Note Taking', 'Content Creation: Text Creation'] | Y | 64 | 3 |
| In the Yahoo Sports app, find the three most recent news articles in the NFL section and the three most recent in the Men's Tennis section. For each of the six articles, read it and remember its title and main content. Then, in Joplin, create a note titled ""General Sports Briefing"". For each of the six articles, write down its title followed by a 40-word summary of its content, grouped under 'NFL' and 'Men's Tennis' headings. | ['Yahoo Sports', 'joplin'] | 2 | Y | ['Sports: Content Navigation', 'Sports: Data Extraction', 'Productivity: Note Taking', 'Content Creation: Text Creation'] | Y | 64 | 3 |
| Open Yahoo Sports app, go to the "Soccer" section and open the "Premier League" fixture list. Identify the next three scheduled league matches. For each match, remember the date and the two teams involved. Open N Calendar app and create three separate events, each titled | "[Team A] vs [Team B]". | 2 | Y | ['Sports: Content Navigation', 'Sports: Data Extraction', 'Productivity: Time Management'] | Y | 45 | 3 |
| Open Yahoo Sports app, go to the "Basketball" section and open the "NBA" schedule. Identify the next three scheduled league games. For each game, remember the date and the two teams involved. Open N Calendar app and create three separate events, each titled "[Team A] vs [Team B]". | ['Yahoo Sports', 'N calendar'] | 2 | Y | ['Sports: Content Navigation', 'Sports: Data Extraction', 'Productivity: Time Management'] | Y | 45 | 3 |
| Open the Amazon app. Search for and remember the price and star rating for these four components: ""AMD Ryzen 7 7800X3D"", ""NVIDIA GeForce RTX 4070 Super"", ""Corsair Vengeance 32GB DDR5 RAM"", and ""Samsung 990 Pro 2TB SSD"". Then, open the Calculator app and calculate the total cost of all four components. Finally, open the Joplin app, create a note titled ""PC Build Cost & Rating"", and list each component with its price, rating, and the calculated total cost at the end. | ['Amazon', 'Calculator', 'joplin'] | 3 | Y | ['E-commerce: Product Search', 'Financial Management: Financial Calculation', 'Productivity: Note Taking'] | Y | 38 | 2 |

| Task | Apps | | | Categories | | | |
|------|------|---|---|------------|---|---|---|
| Open the Amazon app. Search for and remember the price and star rating for these four components: ""Intel Core i9-14900K"", ""AMD Radeon RX 7900 XTX"", ""G.Skill Trident Z5 32GB DDR5 RAM"", and ""WD Black SN850X 4TB SSD"". Then, open the Calculator app and calculate the total cost of all four components. Finally, open the Joplin app, create a note titled ""High-End PC Parts & Rating"", and list each component with its price, rating, and the calculated total cost at the end. | ['Amazon', 'Calculator', 'joplin'] | 3 | Y | ['E-commerce: Product Search', 'E-commerce: Review Analysis', 'Information Retrieval: Data Extraction', 'Financial Management: Financial Calculation', 'Productivity: Note Taking'] | Y | 38 | 2 |
| In the Amazon app, find the price and star rating for four components: ""AMD Ryzen 7 7800X3D"", ""NVIDIA GeForce RTX 4070 Super"", ""Samsung 990 Pro 2TB SSD"", and ""Corsair Vengeance 32GB DDR5 RAM"". In the Calculator app, calculate the subtotal, then a final total by adding an 8% sales tax. Finally, in the Joplin app, create a note titled ""AMD Build Analysis"" listing each component with its price and rating, plus the subtotal and final total. | ['Amazon', 'Calculator', 'joplin'] | 3 | Y | ['E-commerce: Product Search', 'Financial Management: Financial Calculation', 'Productivity: Note Taking'] | Y | 59 | 3 |
| In the Amazon app, find the price and star rating for four components: ""Intel Core i9-14900K"", ""NVIDIA GeForce RTX 4090"", ""WD Black SN850X 4TB SSD"", and ""G.Skill Trident Z5 64GB DDR5 RAM"". In the Calculator app, calculate the subtotal, then a final total by adding an 8% sales tax. Finally, in the Joplin app, create a note titled ""Intel Build Analysis"" listing each component with its price and rating, plus the subtotal and final total. | ['Amazon', 'Calculator', 'joplin'] | 3 | Y | ['E-commerce: Product Search', 'Information Retrieval: Data Extraction', 'Financial Management: Financial Calculation', 'Productivity: Note Taking'] | Y | 59 | 3 |
| In the Amazon app, search for 'Insta360 Ace Pro' and go to its customer reviews section. Read and remember the full original text and the star rating of the top five reviews. Do not use the copy-paste function. Then, in the DeepL Translate app, translate the full text of all five reviews into French. Finally, in the Joplin app, create a note titled ""Insta360 Review Analysis"". For each of the five reviews, list its original star rating, its original full text, and its French translation. | ['Amazon', 'DeepL Translate', 'joplin'] | 3 | Y | ['E-commerce: Product Search', 'E-commerce: Review Analysis', 'Content Creation: Translation', 'Productivity: Note Taking'] | Y | 70 | 3 |
| In the Amazon app, search for 'DJI Mini 4 Pro' and go to its customer reviews section. Read and remember the full original text and the star rating of the top five reviews. Do not use the copy-paste function. Then, in the DeepL Translate app, translate the full text of all five reviews into Russian. Finally, in the Joplin app, create a note titled ""DJI Review Analysis"". For each of the five reviews, list its original star rating, its original full text, and its Russian translation. | ['Amazon', 'DeepL Translate', 'joplin'] | 3 | Y | ['E-commerce: Product Search', 'E-commerce: Review Analysis', 'Content Creation: Translation', 'Productivity: Note Taking'] | Y | 70 | 3 |
| First, in Amazon, find the price and star rating for the 'GoPro HERO12 Black', then find the founding year and founder of 'GoPro' in Wikipedia. Second, repeat this entire process for the 'Insta360 Ace Pro' camera and 'Insta360' company. Finally, send a single message to +8613412345678 containing all eight pieces of collected data for both products. | ['Amazon', 'Wikipedia', 'messages'] | 3 | Y | ['E-commerce: Product Search', 'E-commerce: Multi-App Comparison', 'Information Retrieval: Data Extraction', 'Communication: Messaging'] | Y | 32 | 2 |
| First, in Amazon, find the price and star rating for the 'DJI Mini 4 Pro', then find the founding year and founder of 'DJI' in Wikipedia. Second, repeat this entire process for the 'Autel EVO Lite+' drone and 'Autel Robotics' company. Finally, send a single message to +8613487654321 containing all eight pieces of collected data for both products. | ['Amazon', 'Wikipedia', 'messages'] | 3 | Y | ['E-commerce: Product Search', 'E-commerce: Price Comparison', 'Information Retrieval: Data Extraction', 'Information Retrieval: Fact Checking', 'Communication: Data Sharing'] | Y | 32 | 2 |
| Search for 'Bose QuietComfort Ultra Headphones' on both the Amazon and Wish apps, remembering the price and currency from each. If the currencies are different, use the bing app to find the exchange rate to compare them. Directly answer with the name of the app, 'Amazon' or 'Wish', where the price is lower. | ['Amazon', 'Wish', 'bing'] | 3 | Y | ['E-commerce: Product Search', 'E-commerce: Price Comparison', 'E-commerce: Multi-App Comparison', 'Information Retrieval: Web Search'] | Y | 14 | 1 |
| Search for 'iPhone 16 Pro Max' on both the Amazon and Wish apps, remembering the price and currency from each. If the currencies are different, use the bing app to find the exchange rate to compare them. Directly answer with the name of the app, 'Amazon' or 'Wish', where the price is lower. | ['Amazon', 'Wish', 'bing'] | 3 | Y | ['E-commerce: Product Search', 'E-commerce: Price Comparison', 'E-commerce: Multi-App Comparison', 'Information Retrieval: Web Search'] | Y | 15 | 1 |
| Open the Amazon app and search for the 'Intel Core i5-13600K' CPU, remembering its price. Open the bing app and search for ""what socket does Intel Core i5-13600K use"". Remember the socket type. Return to the Amazon app, search for a motherboard with that socket type, and remember the price of the first result. Finally, open the joplin app, create a note titled ""CPU/Mobo Combo"", and list the names and prices of the CPU and the compatible motherboard. | ['Amazon', 'bing', 'joplin'] | 3 | Y | ['E-commerce: Product Search', 'E-commerce: Compatibility Check', 'E-commerce: Multi-App Comparison', 'Information Retrieval: Web Search', 'Productivity: Note Taking'] | Y | 25 | 2 |
| Open the Amazon app and search for the 'AMD Ryzen 5 7600X' CPU, remembering its price. Open the bing app and search for ""what socket does AMD Ryzen 5 7600X use"". Remember the socket type. Return to the Amazon app, search for a motherboard with that socket type, and remember the price of the first result. Finally, open the joplin app, create a note titled ""AMD Build Parts"", and list the names and prices of the CPU and the compatible motherboard. | ['Amazon', 'bing', 'joplin'] | 3 | Y | ['E-commerce: Product Search', 'E-commerce: Compatibility Check', 'E-commerce: Multi-App Comparison', 'Information Retrieval: Web Search', 'Productivity: Note Taking'] | Y | 25 | 2 |
| In the AP News app, find the top three stories from the 'World' section and the top three from the 'U.S. News' section. For each of the six stories, summarize its first paragraph (30 words) and translate its headline into German in DeepL Translate app. Then, in the Joplin app, create a note titled ""Global News Report"" listing the original headline, its summary, and its German translation for all six stories, grouped by section. | ['AP News', 'DeepL Translate', 'joplin'] | 3 | Y | ['Information Retrieval: Data Extraction', 'Content Creation: Text Creation', 'Content Creation: Translation', 'Productivity: Note Taking'] | Y | 45 | 3 |

| Task | Apps | | | Categories | | | |
|---|---|---|---|---|---|---|---|
| In the AP News app, find the top three stories from the 'Technology' section and the top three from the 'Sports' section. For each of the six stories, summarize its first paragraph (30 words) and translate its headline into Spanish in DeepL Translate app. Then, in the Joplin app, create a note titled ""Tech & Sports Report"" listing the original headline, its summary, and its Spanish translation for all six stories, grouped by section. | ['AP News', 'DeepL Translate', 'joplin'] | 3 | Y | ['Sports: Content Navigation', 'Information Retrieval: Data Extraction', 'Content Creation: Text Creation', 'Content Creation: Translation', 'Productivity: Note Taking'] | Y | 45 | 3 |
| In the Apartments.com app, search for 'Chicago, IL' apartments with '2 beds' and 'in-unit laundry'. For the first three results, remember the monthly rent and square footage of each. In the Calculator app, calculate the rent per square foot for all three. Finally, open the messages app and send a message to +8613355556666 identifying the apartment with the best value (lowest rent per sq ft) and stating its calculated value. | ['Apartments.com Rental Search', 'Calculator', 'messages'] | 3 | Y | ['E-commerce: Product Search', 'E-commerce: Filter & Sort', 'Information Retrieval: Data Extraction', 'Financial Management: Financial Calculation', 'Communication: Messaging', 'Communication: Data Sharing'] | Y | 39 | 2 |
| In the Apartments.com app, search for 'Miami, FL' apartments with '2 beds' and 'in-unit laundry'. For the first three results, remember the monthly rent and square footage of each. In the Calculator app, calculate the rent per square foot for all three. Finally, open the messages app and send a message to +8613377778888 identifying the apartment with the best value (lowest rent per sq ft) and stating its calculated value. | ['Apartments.com Rental Search', 'Calculator', 'messages'] | 3 | Y | ['E-commerce: Product Search', 'E-commerce: Filter & Sort', 'Information Retrieval: Data Extraction', 'Financial Management: Financial Calculation', 'Communication: Messaging', 'Communication: Data Sharing', 'Travel & Navigation: Local Search'] | Y | 39 | 2 |
| Open Apartments.com app, search 'Austin, TX', and apply three filters: '2 Beds', 'Dog Friendly', and a max price of '$3000'. For the top two results, go to the details page, find the list of amenities, and remember three specific ones. Also, find the address. Then, use Citymapper app to find the commute time from each address to 'University of Texas at Austin'. In Joplin app, note which apartment has more of the desired amenities and a shorter commute. | ['Apartments.com Rental Search', 'Citymapper', 'joplin'] | 3 | Y | ['E-commerce: Product Search', 'E-commerce: Filter & Sort', 'Information Retrieval: Data Extraction', 'Travel & Navigation: Route Planning', 'E-commerce: Multi-App Comparison', 'Productivity: Note Taking'] | Y | 70 | 3 |
| In the Apartments.com app, search for 'Seattle, WA', and apply three filters: '1 Bed', 'Cat Friendly', and a max price of '$2500'. For the top two results, go to the details page, find the list of amenities, and remember if they have three specific ones: 'In-unit Washer', 'Balcony', and 'Fitness Center'. Also, find the address. Then, in the Citymapper app, find the commute time from each address to 'University of Washington'. In the Joplin app, note which apartment has more of the desired amenities and a shorter commute. | ['Apartments.com Rental Search', 'Citymapper', 'joplin'] | 3 | Y | ['E-commerce: Product Search', 'E-commerce: Filter & Sort', 'E-commerce: Specification Comparison', 'Information Retrieval: Data Extraction', 'Travel & Navigation: Route Planning', 'Productivity: Note Taking'] | Y | 70 | 3 |
| In the BBC Sports app, go to the 'Formula 1 -¿ 'Drivers" standings. Remember the names and points of the top three drivers. Then, for each driver, search in the Bing app for their age. In the Joplin app, create a note titled ""F1 Top 3 Ages"" listing each driver with their points and age. | ['BBC Sports', 'bing', 'joplin'] | 3 | Y | ['Sports: Content Navigation', 'Sports: Data Extraction', 'Information Retrieval: Web Search', 'Productivity: Note Taking'] | Y | 24 | 1 |
| In BBC Sports app, go to the 'Formula 1 -¿ 'Constructors" standings. Remember the names and points of the top three teams. Then, for each team, search in Bing app for their ""team principal"" name. In Joplin app, create a note titled ""F1 Top 3 Principals"" listing each team with their points and team principal. | ['BBC Sports', 'bing', 'joplin'] | 3 | Y | ['Sports: Data Extraction', 'Information Retrieval: Web Search', 'Productivity: Note Taking'] | Y | 24 | 1 |
| In Bing app, find the current USD prices for both Bitcoin (BTC) and Ethereum (ETH). In the Calculator app, calculate the total value of a portfolio with 1.5 BTC and 25 ETH. Then, in Bluecoins app, create a new asset account under the 'Investments' group named ""Crypto Portfolio"", with the calculated total as its initial value. | ['bing', 'Calculator', 'bluecoins'] | 3 | Y | ['Information Retrieval: Web Search', 'Financial Management: Financial Calculation', 'Financial Management: Add Transaction'] | Y | 32 | 2 |
| In Bing app, find the current USD prices for one ounce of gold and silver. In the Calculator app, calculate the total value of a portfolio with 10 ounces of gold and 500 ounces of silver. Then, in Bluecoins app, create a new asset account under the 'Investments' group named ""Precious Metals"", with the calculated total as its initial value. | ['bing', 'Calculator', 'bluecoins'] | 3 | Y | ['Information Retrieval: Web Search', 'Financial Management: Financial Calculation', 'Financial Management: Add Transaction'] | Y | 31 | 2 |
| In the Bing app, find the cheapest flight from London to Rome within the next seven days. Remember its airline, departure date and time, and price. In the Citymapper app, find the journey duration from ""Trafalgar Square"" to ""Heathrow Airport"". Finally, in the N calendar app, create an event on the cheapest flight's date and time, title it ""[Airline] Flight - [Price]"", and set a reminder for [journey duration + 2 hours] before departure. | ['bing', 'N calendar', 'Citymapper'] | 3 | Y | ['Travel & Navigation: Flight Booking', 'Travel & Navigation: Route Planning', 'Productivity: Time Management', 'Information Retrieval: Data Extraction'] | Y | 75 | 3 |
| In the Bing app, find the cheapest flight from Tokyo to Seoul within the next seven days. Remember its airline, arrival date and time, and price. In the Citymapper app, find the journey duration from ""Incheon International Airport"" to ""Myeong-dong"". Finally, in the N calendar app, create an arrival event on the cheapest flight's date and time, titled ""[Airline] Arrival - [Price]"", then create a second event immediately following it for the transit to Myeong-dong, using the remembered duration. | ['bing', 'N calendar', 'Citymapper'] | 3 | Y | ['Travel & Navigation: Flight Booking', 'Travel & Navigation: Route Planning', 'Productivity: Time Management', 'Information Retrieval: Data Extraction'] | Y | 75 | 3 |

| Task | Apps | | | Capabilities | | | |
|---|---|---|---|---|---|---|---|
| Open the bing app, search for 'NVIDIA stock price', and remember the current price. Then open the AP News app, find the latest news about NVIDIA earnings report, and remember the reported revenue growth rate. Finally, open the Calculator app and calculate: stock price * (1 + revenue growth rate). | ['bing', 'AP News', 'Calculator'] | 3 | Y | ['Information Retrieval: Web Search', 'Information Retrieval: Data Extraction', 'Financial Management: Financial Calculation'] | Y | 17 | 1 |
| Open the bing app, search for 'Apple stock price', and remember the current price. Then open the AP News app, find the latest news about Apple earnings report, and remember the reported revenue growth rate. Finally, open the Calculator app and calculate: stock price * (1 + revenue growth rate). | ['bing', 'AP News', 'Calculator'] | 3 | Y | ['Information Retrieval: Data Extraction', 'Financial Management: Financial Calculation'] | Y | 18 | 1 |
| Open the bing app and search for 'minimalist black and white wallpaper' images. From the results, find the first image that has a portrait (vertical) orientation suitable for a phone wallpaper. Long-press to download this image. Then, open the Setting app, navigate to wallpaper settings, and change your home screen wallpaper to the image you just downloaded. | ['bing', 'Files', 'Setting'] | 3 | Y | ['Information Retrieval: Image Search & Understanding', 'Device Configuration: Setting Adjustment'] | Y | 15 | 1 |
| Open the bing app and search for 'abstract blue ocean wallpaper' images. From the results, find the first image that is in a portrait (vertical) format. Long-press and download this image. Afterwards, open the Setting app, navigate to the wallpaper settings, and change your lock screen wallpaper to the downloaded image. | ['bing', 'Files', 'Setting'] | 3 | Y | ['Information Retrieval: Image Search & Understanding', 'Device Configuration: Setting Adjustment'] | Y | 16 | 1 |
| Open Cars.co.za app, search for 'Toyota' cars. Apply filters: price between R150,000-R200,000, year after 2020. From the results, find the top three cars. For each, remember its price and mileage. Open the Calculator app and calculate the price-to-mileage ratio (price/mileage) for each car. In Joplin app, list the three cars and their ratios, and state which has the best (lowest) ratio. | ['Cars.co.za', 'Calculator', 'joplin'] | 3 | Y | ['E-commerce: Product Search', 'E-commerce: Filter & Sort', 'Information Retrieval: Data Extraction', 'Financial Management: Financial Calculation', 'Productivity: Note Taking', 'E-commerce: Price Comparison'] | Y | 60 | 3 |
| Open Cars.co.za app, search for 'BMW' cars. Apply filters: price between R250,000-R300,000, year after 2019. From the results, find the top three cars. For each, remember its price and engine size. Open the Calculator app and calculate the average engine size. In Joplin app, list the three cars with their prices and state the calculated average engine size. | ['Cars.co.za', 'Calculator', 'joplin'] | 3 | Y | ['E-commerce: Product Search', 'E-commerce: Filter & Sort', 'Information Retrieval: Data Extraction', 'Financial Management: Financial Calculation', 'Productivity: Note Taking'] | Y | 60 | 3 |
| Open the wikiHow app and search for ""how to bake chocolate chip cookies"". Create a checklist in the joplin app named ""Cookie Ingredients"" with the first four ingredients listed. Then, open the Calculator app and calculate the total cost, assuming each of the four ingredients costs $3.50. | ['wikiHow', 'joplin', 'Calculator'] | 3 | Y | ['Information Retrieval: Web Search', 'Education & Learning: Knowledge Acquisition', 'Productivity: Checklist Management', 'Financial Management: Financial Calculation'] | Y | 15 | 1 |
| Open the wikiHow app and search for ""how to make a pizza"". Create a checklist in the joplin app named ""Pizza Ingredients"" with the first four ingredients listed. Then, open the Calculator app and calculate the total cost, assuming each of the four ingredients costs $4.25. | ['wikiHow', 'joplin', 'Calculator'] | 3 | Y | ['Education & Learning: Knowledge Acquisition', 'Productivity: Checklist Management', 'Information Retrieval: Data Extraction', 'Financial Management: Financial Calculation'] | Y | 12 | 1 |
| In the Wish app, find and remember the prices for three items: an 'RGB Mechanical Keyboard', a 'Wireless Gaming Mouse', and a 'Large Gaming Mousepad'. In the Calculator app, first sum the prices of all three items to get a subtotal. Then, calculate a 15% discount on this subtotal. Finally, add a 7% tax to the discounted price to get the final total. Open the messages app and send a message to +8613211112222 with the content: ""Subtotal: [sum], After 15% discount: [discounted price], Final Total (incl. 7% tax): [final price]"". | ['Wish', 'Calculator', 'messages'] | 3 | Y | ['E-commerce: Product Search', 'Financial Management: Financial Calculation', 'Communication: Messaging', 'Communication: Data Sharing'] | Y | 60 | 3 |
| In the Wish app, find and remember the prices for three items: 'Bluetooth Earbuds', a 'Portable Power Bank', and a 'Phone Stand for Desk'. In the Calculator app, first sum the prices of all three items. Then, calculate the final grand total by adding a fixed shipping fee of 3 units per item and a 4% import duty on the item subtotal. Open the messages app and send a message to +8613233334444 with the content: ""Subtotal: [sum], Shipping: 9.00, Duty (4%): [duty amount], Grand Total: [final price]"". | ['Wish', 'Calculator', 'messages'] | 3 | Y | ['E-commerce: Product Search', 'Financial Management: Financial Calculation', 'Communication: Messaging', 'Communication: Data Sharing'] | Y | 60 | 3 |
| In the Amazon app, find the USD prices for two items: ""Sony WH-1000XM5 headphones"" and a ""Logitech MX Master 3S mouse"". In the Bing app, find the ""USD to EUR"" exchange rate. In the Calculator app, sum the two item prices, convert the subtotal to EUR, then add a fixed EUR25 shipping fee to get the grand total. Finally, in the bluecoins app, add a single expense transaction for the grand total, named ""EU Tech Import"", and add a note: ""Items: [USD subtotal], Shipping: EUR25"". | ['Amazon', 'bing', 'Calculator', 'bluecoins'] | 4 | Y | ['E-commerce: Product Search', 'Information Retrieval: Fact Checking', 'Financial Management: Financial Calculation', 'Financial Management: Add Transaction'] | Y | 51 | 3 |
| In the Amazon app, find the USD prices for two items: ""Apple Watch Series 9"" and ""Apple AirPods Pro 2"". In the Bing app, find the ""USD to GBP"" exchange rate. In the Calculator app, sum the two item prices, convert the subtotal to GBP, then add a fixed GBP20 shipping fee to get the grand total. Finally, in the bluecoins app, add a single expense transaction for the grand total, named ""UK Apple Import"", and add a note: ""Items: [USD subtotal], Shipping: GBP20"". | ['Amazon', 'bing', 'Calculator', 'bluecoins'] | 4 | Y | ['E-commerce: Product Search', 'Information Retrieval: Web Search', 'Financial Management: Financial Calculation', 'Financial Management: Add Transaction'] | Y | 51 | 3 |

| Task | Apps | | | Categories | | | |
|---|---|---|---|---|---|---|---|
| Open the Apartments.com app, search in 'Mountain View, CA', and remember the address and rent of the first result. Open the bing app and search for the address of the ""Googleplex"". Open the Citymapper app and find the public transit commute time between the apartment address and the Googleplex. Finally, open the joplin app and create a note titled ""Google Commute"" with the apartment's rent and the calculated commute time. | ['Apartments.com Rental Search', 'bing', 'Citymapper', 'joplin'] | 4 | Y | ['E-commerce: Product Search', 'Information Retrieval: Web Search', 'Travel & Navigation: Route Planning', 'Productivity: Note Taking', 'Information Retrieval: Data Extraction'] | Y | 19 | 1 |
| Open the Apartments.com app, search in 'Cupertino, CA', and remember the address and rent of the first result. Open the bing app and search for the address of ""Apple Park"". Open the Citymapper app and find the driving time between the apartment address and Apple Park. Finally, open the joplin app and create a note titled ""Apple Commute"" with the apartment's rent and the calculated driving time. | ['Apartments.com Rental Search', 'bing', 'Citymapper', 'joplin'] | 4 | Y | ['E-commerce: Product Search', 'Information Retrieval: Web Search', 'Information Retrieval: Data Extraction', 'Travel & Navigation: Route Planning', 'Productivity: Note Taking'] | Y | 25 | 2 |
| Open Apartments.com, search for 'Los Angeles, CA', filter for '2 beds', and remember the monthly rent of the first result. Open the Calculator app and calculate the annual rent. Open the bluecoins app and create a new monthly budget for 'Rent' with the remembered monthly amount. Finally, open the N calendar app and set a reminder for the 1st of next month titled ""Pay Rent"". | ['Apartments.com Rental Search', 'Calculator', 'bluecoins', 'N calendar'] | 4 | Y | ['E-commerce: Product Search', 'E-commerce: Filter & Sort', 'Financial Management: Financial Calculation', 'Financial Management: Create Budget', 'Productivity: Time Management'] | Y | 25 | 2 |
| Open Apartments.com, search for 'Chicago, IL', filter for 'in-unit laundry', and remember the monthly rent of the first result. Open the Calculator app and calculate a 1.5x security deposit based on the rent. Open the bluecoins app and create a savings goal named ""Apartment Deposit"" for this calculated amount. Finally, open the N calendar app and set a reminder for tomorrow titled ""Follow up on Chicago apartment"". | ['Apartments.com Rental Search', 'Calculator', 'bluecoins', 'N calendar'] | 4 | Y | ['E-commerce: Product Search', 'E-commerce: Filter & Sort', 'Information Retrieval: Data Extraction', 'Financial Management: Financial Calculation', 'Financial Management: Set Saving Goal', 'Productivity: Time Management'] | Y | 17 | 1 |
| In the Bing app, search for and observe four paintings: ""The Scream"" by Munch, ""Guernica"" by Picasso, ""The Third of May 1808"" by Goya, and ""Saturn Devouring His Son"" by Goya. In the Joplin app, write a note comparing how these four artworks depict human suffering and fear (60 words total). In the DeepL Translate app, translate your entire comparison into Spanish. Finally, open the messages app and send the Spanish translation to +8613511112222. | ['bing', 'joplin', 'DeepL Translate', 'messages'] | 4 | Y | ['Information Retrieval: Image Search & Understanding', 'Content Creation: Text Creation', 'Content Creation: Translation', 'Communication: Messaging'] | Y | 40 | 2 |
| In the Bing app, search for and observe four paintings: ""Mona Lisa"" by da Vinci, ""The Starry Night"" by van Gogh, ""Les Demoiselles d'Avignon"" by Picasso, and ""Composition VII"" by Kandinsky. In the Joplin app, write a note comparing the use of realism, color, and form across these four distinct art movements (60 words total). In the DeepL Translate app, translate your entire comparison into Italian. Finally, open the messages app and send the Italian translation to +8613633334444. | ['bing', 'joplin', 'DeepL Translate', 'messages'] | 4 | Y | ['Information Retrieval: Image Search & Understanding', 'Content Creation: Text Creation', 'Productivity: Note Taking', 'Content Creation: Translation', 'Communication: Messaging'] | Y | 40 | 2 |
| Open Cars.co.za app, search for a 'Volkswagen Polo', and remember the price of the first result. Then search for a 'Hyundai i20' and remember the price of the first result. Open the Calculator app and find the price difference. Open the joplin app to note which car is more expensive and by how much. Finally, open the bing app and search for ""Volkswagen Polo vs Hyundai i20 safety rating"". Stay on the search results page. | ['Cars.co.za', 'bing', 'joplin', 'Calculator'] | 4 | Y | ['E-commerce: Product Search', 'E-commerce: Price Comparison', 'E-commerce: Multi-App Comparison', 'Financial Management: Financial Calculation', 'Productivity: Note Taking', 'Information Retrieval: Web Search'] | Y | 12 | 1 |
| Open Cars.co.za app, search for a 'Ford Ranger', and remember the price of the first result. Then search for a 'Toyota Hilux' and remember the price of the first result. Open the Calculator app and find the price difference. Open the joplin app to note which car is more expensive and by how much. Finally, open the bing app and search for ""Ford Ranger vs Toyota Hilux reliability"". Stay on the search results page. | ['Cars.co.za', 'Calculator', 'joplin', 'bing'] | 4 | Y | ['E-commerce: Product Search', 'E-commerce: Price Comparison', 'E-commerce: Multi-App Comparison', 'Financial Management: Financial Calculation', 'Productivity: Note Taking', 'Information Retrieval: Web Search'] | Y | 19 | 1 |

## A.5 DETAILS OF FRAMEWORK ARCHITECTURE

This section provides comprehensive technical specifications for the snapshot-based plug-and-play framework presented in Section 3.2.

### A.5.1 PARALLEL EXPERIMENT IMPLEMENTATION

Our framework achieves scalable parallel execution through a sophisticated emulator management system. We pre-configured MemGUI-AVD (Android Virtual Device), a customized emulator image that includes all required applications with pre-established permissions (file access, location services, etc.) and optimized settings for GUI automation. Each experimental instance creates an

independent emulator from this base image, ensuring identical starting conditions across all parallel executions.

The system implements port-based isolation using Android Debug Bridge (ADB) connections, where each emulator instance is assigned a unique port number (e.g., 5554, 5556, 5558) to enable simultaneous agent-environment communication without interference. This architecture supports concurrent execution of multiple agents on the same hardware while maintaining strict experimental isolation. The mirror task design ensures sequential execution within each parallel stream, preserving the integrity of long-term learning assessment where task order may influence learning outcomes.

### A.5.2 Long-Term Memory Support Through Multi-Attempt Mechanism

Our framework implements long-term memory evaluation through the `pass@k` protocol, where agents are allowed up to $k$ attempts per task (default $k = 3$). Between attempts, agents with long-term memory capabilities can analyze failure patterns, update their knowledge bases, and adjust strategies for subsequent tries. The framework maintains persistent agent state across attempts while ensuring environment consistency through snapshot-based resets, enabling fair assessment of cross-session learning capabilities.

### A.5.3 Comprehensive Agent Integration

The framework supports twelve prominent GUI agents across diverse architectural paradigms through a unified interface that accommodates both agentic workflows and end-to-end models. Table 5 provides detailed specifications for each integrated agent, including their memory mechanisms, backbone models, and deployment configurations. All agents utilize standardized action spaces and observation formats while preserving their unique architectural characteristics.

### A.5.4 Advantages Over Existing Approaches

Our framework provides significant improvements over existing benchmarking environments in three key areas:

**Environment Scalability and Convenience.** Unlike AndroidWorld (Rawles et al., 2024) and AndroidLab (Xu et al., 2024), which rely on pre-written expert scripts for environment recovery and setup, our approach offers superior extensibility without requiring specialized knowledge for script development. While expert scripts facilitate environment reset for pre-configured applications, they are fundamentally limited by application constraints—mainstream software like Amazon cannot be easily manipulated through script injection or state reading mechanisms. Additionally, the scalability is severely constrained by the expert knowledge required for script development.

**Rapid Environment Recovery.** In contrast to SPA-Bench (Chen et al., 2024) and A3 (Chai et al., 2025), which include mainstream applications but require manual environment reset and partially depend on physical devices, our snapshot-based approach enables instant environment recovery. This advantage stems from our strategic application selection constraints: emulator compatibility ensures reliable operation in virtualized environments, while login-free operation eliminates the need for manual cleanup of user-generated data (favorites, search history, etc.). As demonstrated in our application selection strategy, Amazon, Apartments.com, and Citymapper provide comprehensive functionality in guest mode, enabling automated state recovery while maintaining task authenticity.

**Native Long-Term Memory Support.** Our framework uniquely provides built-in support for long-term memory evaluation through the `pass@k` protocol and persistent agent state management across multiple attempts. This capability is absent in existing benchmarks, which focus exclusively on single-attempt evaluation and cannot assess agents' ability to learn from experience and improve performance over time.

### A.6 Details of Memory-Specialized Metrics

This section provides comprehensive mathematical definitions and computational procedures for the 7 specialized metrics introduced in Section 4.1.

### A.6.1 SHORT-TERM MEMORY ASSESSMENT METRICS

**Overall Success Rate (SR)** serves as our baseline metric, measuring the fundamental ability to complete tasks and providing essential context for interpreting memory-specific performance. This metric provides a foundation for understanding overall agent capabilities before analyzing memory-specific performance patterns.

**Information Retention Rate (IRR)** constitutes our core memory fidelity metric, quantifying the proportion of required information units that agents correctly recall and utilize during task execution. Unlike binary success indicators, IRR provides fine-grained insights into partial memory failures—for instance, distinguishing an agent that correctly processes 7 out of 9 required information pieces from one that fails entirely. This metric specifically targets the temporary information buffering capability that characterizes human-like short-term memory in GUI interactions.

Mathematical Definition:

$$\text{IRR}_i = \frac{\text{Correctly recalled and used information units in task } i}{\text{Total required information units in task } i} \times 100\%$$

The average IRR across all memory-intensive tasks is computed as:

$$\text{Avg. IRR} = \frac{1}{N_{memory}} \sum_{i \in memory\_tasks} \text{IRR}_i$$

where $N_{memory}$ represents the number of memory-intensive tasks (115 in MemGUI-Bench).

**Memory-Task Proficiency Ratio (MTPR)** isolates memory-specific capabilities by comparing performance on our 115 memory-intensive tasks against 13 standard tasks, enabling researchers to distinguish memory limitations from general task execution deficits.

Mathematical Definition:

$$\text{MTPR} = \frac{\text{SR}_{memory\_tasks}}{\text{SR}_{standard\_tasks}}$$

where $\text{SR}_{memory\_tasks}$ and $\text{SR}_{standard\_tasks}$ represent success rates on memory-intensive and standard tasks, respectively.

### A.6.2 LONG-TERM MEMORY ASSESSMENT METRICS

**Multi-Attempt Success Rate (pass@k SR)** serves as our primary long-term learning indicator, measuring agents' ability to leverage knowledge from previous attempts to eventually succeed within $k$ trials. This metric directly reflects the cumulative benefit of long-term memory mechanisms in helping agents overcome initial failures through experience accumulation.

Mathematical Definition:

$$\text{pass@k SR} = \frac{\text{Number of tasks succeeded within } k \text{ attempts}}{\text{Total number of tasks}} \times 100\%$$

**Failure Recovery Rate (FRR)** specifically targets the speed and effectiveness of learning from failure, employing a harmonic decay weighting model that rewards agents capable of rapid recovery from initial failures. This metric recognizes that superior long-term memory should enable faster learning rather than merely eventual success.

Mathematical Definition:

$$\text{FRR} = \frac{\sum_{i=2}^{k} w_i \times R_i}{N_{failed}} \times 100\%$$

where: - $N_{failed}$ = number of tasks that failed on the first attempt - $R_i$ = number of tasks that succeeded for the first time on attempt $i$ - $w_i = \frac{1}{i-1}$ = harmonic decay weight for attempt $i \geq 2$

This weighting scheme ensures that earlier recoveries contribute more significantly to the overall score, reflecting the principle that effective long-term memory should enable rapid learning from experience.

### A.6.3 EXECUTION EFFICIENCY ASSESSMENT METRICS

**Average Step Ratio** measures path efficiency by comparing agent execution paths against golden standards exclusively for successfully completed tasks, revealing whether sophisticated memory systems enable more direct task completion when they do succeed.

Mathematical Definition:

$$\text{Step Ratio}_i = \frac{\text{Agent steps in task } i}{\text{Golden steps in task } i}$$

$$\text{Average Step Ratio} = \frac{1}{N_{success}} \sum_{i \in successful\_tasks} \text{Step Ratio}_i$$

**Average Time Per Step** quantifies the computational overhead of memory-enhanced decision-making across all task attempts, providing insights into the speed-accuracy trade-offs inherent in different memory architectures.

Mathematical Definition:

$$\text{Time Per Step}_i = \frac{\text{Total execution time for task } i}{\text{Agent steps in task } i}$$

$$\text{Average Time Per Step} = \frac{1}{N_{total}} \sum_{i=1}^{N_{total}} \text{Time Per Step}_i$$

**Average Cost Per Step** evaluates the economic efficiency of memory mechanisms across all executions, particularly relevant for comparing framework-based agents with dedicated memory modules against end-to-end model approaches.

Mathematical Definition:

$$\text{Cost Per Step}_i = \frac{\text{Total API cost for task } i}{\text{Agent steps in task } i}$$

$$\text{Average Cost Per Step} = \frac{1}{N_{total}} \sum_{i=1}^{N_{total}} \text{Cost Per Step}_i$$

### A.6.4 COMPUTATIONAL CONSIDERATIONS

For tasks where agents achieve perfect success (SR = 100%), the IRR is automatically set to 100%. For failed tasks, IRR is computed based on the actual proportion of correctly recalled and utilized information units. In cases of early failure where no information units are processed, IRR = 0%.

The MTPR provides insights into memory-specific capabilities: MTPR > 1 indicates superior performance on memory tasks, MTPR = 1 suggests equivalent performance across task types, and MTPR < 1 reveals memory-specific deficits.

For pass@k evaluation, tasks are considered successful if they achieve success in any of the k attempts. The FRR metric specifically focuses on the subset of initially failed tasks to quantify learning effectiveness from failure experiences.

### A.7 DETAILS OF EVALUATION PIPELINE VALIDATION

This section provides comprehensive technical details for the evaluation pipeline validation experiments presented in Section 4.3.

### A.7.1 EXPERIMENTAL SETUP DETAILS

**Task Selection Strategy.** Our validation employs two complementary evaluation datasets to comprehensively assess pipeline reliability. First, we selected 26 tasks from SPA-Bench (Chen et al., 2024)—18 single-app and 8 cross-app tasks—executing each three times with M3A (Rawles et al., 2024) to generate 78 trajectories (54 single-app, 24 cross-app) for direct comparison with SPA-Bench's evaluator. This selection ensures cross-benchmark transferability assessment while maintaining fair comparison conditions. Second, we utilized all 128 MemGUI-Bench tasks executed by both M3A and T3A under `pass@1` settings, yielding 256 trajectories that represent the full spectrum of our memory-intensive evaluation scenarios.

**Model Configuration Design.** To systematically assess evaluator robustness and cost-performance trade-offs, we designed comprehensive model configurations for both `MemGUI-Eval` and baseline methods. For `MemGUI-Eval`, we tested three strategic configurations: M1 (Gemini 2.5 Pro + Pro) where all specialized agents—*Triage Judge*, *Step Descriptor*, *Semantic Judge*, *Visual Judge*, and *IRR Analyzer*—use Gemini 2.5 Pro for maximum accuracy; M2 (Gemini 2.5 Flash + Pro) where the *Step Descriptor* uses Gemini 2.5 Flash for cost efficiency while judgment agents use Pro for accuracy; and M3 (Gemini 2.5 Flash + Flash) where all agents use Flash for maximum cost reduction. For SPA-Bench baseline comparisons, we evaluated G1 (Gemini 2.5 Pro), G2 (Gemini 2.5 Flash), and G3 (GPT-4o) configurations. This design enables systematic analysis of evaluator robustness across different cost-accuracy configurations while ensuring fair comparison with existing evaluation methodologies.

**Human Annotation Process.** To establish ground truth labels, each trajectory was independently annotated by three human experts for success/failure determination. Annotators achieved consensus through structured discussion, resolving any disagreements to produce final labels that serve as the gold standard for evaluator performance assessment. The annotation process followed strict guidelines to ensure consistency and reliability across all evaluation scenarios.

### A.7.2 DETAILED RESULTS ANALYSIS

**Cross-Benchmark Performance.** The cost metric represents the average API expense per trajectory evaluation, encompassing all model calls made by the evaluator during the progressive scrutiny process. On SPA-Bench trajectories, our M1 configuration achieves near-perfect performance (99.0% F1-score), significantly outperforming the best baseline (G1: 92.5% F1-score). The M2 configuration provides an optimal balance with 95.9% F1-score at substantially reduced cost ($0.031 vs $0.055), while even our most economical M3 configuration (93.7% F1-score) maintains competitive accuracy with dramatic cost reduction.

**Memory-Intensive Task Performance.** For MemGUI-Bench trajectories, our evaluation maintains consistent high performance across diverse memory-intensive scenarios. The M1 configuration achieves 93.1% F1-score, demonstrating robustness across different task complexities and memory requirements. Notably, the performance gap between single-app and cross-app tasks reveals the sophistication of our progressive scrutiny approach: while baseline methods struggle with cross-app complexity (achieving only 40-61.5% F1-score), `MemGUI-Eval` maintains exceptional performance (94.1-100% F1-score) across all task types.

**Cost-Effectiveness Analysis.** The progressive scrutiny approach demonstrates superior cost-effectiveness compared to traditional evaluation methods. The M2 configuration achieves the optimal balance between evaluation quality and computational efficiency, providing robust assessment capabilities while maintaining economic feasibility for large-scale evaluation scenarios.

### A.7.3 DETAILED PERFORMANCE BREAKDOWN

This section provides a more granular breakdown of the evaluator validation experiments with comprehensive performance analysis across different task complexities and agent types.

Table 9 presents the comprehensive evaluation performance breakdown on SPA-Bench task subsets. The results clearly demonstrate `MemGUI-Eval`'s superiority over baseline methods across different model configurations. For single-app tasks, our method consistently outperforms SPA-Bench's evaluator across all accuracy metrics, with the M1 configuration achieving near-perfect performance (98.8% F1-score vs. 92.5% for the best baseline). The advantage becomes even more pronounced

Table 9: Detailed evaluation performance breakdown on SPA-Bench task subsets.

| Task Subset | Evaluator | Model Config. | Accuracy Metrics (%) | | | Efficiency |
|---|---|---|---|---|---|---|
| | | | F1 ↑ | Prec. ↑ | Recall ↑ | Cost ($) ↓ |
| SINGLE-APP TASKS (N=54) | | | | | | |
| Single-App (N=54) | MemGUI-Eval (Ours) | Gemini 2.5 Pro + Pro | **98.8** | **100.0** | **97.6** | 0.059 |
| | | Gemini 2.5 Flash + Pro | 96.3 | 97.5 | 95.1 | 0.027 |
| | | Gemini 2.5 Flash + Flash | 93.7 | 97.4 | 90.2 | **0.018** |
| | SPA-Bench (Baseline) | Gemini 2.5 Pro | 92.5 | 94.9 | 90.2 | 0.040 |
| | | Gemini 2.5 Flash | 86.8 | 94.3 | 80.5 | 0.037 |
| | | GPT-4o | 84.2 | 91.4 | 78.0 | 0.099 |
| CROSS-APP TASKS (N=24) | | | | | | |
| Cross-App (N=24) | MemGUI-Eval (Ours) | Gemini 2.5 Pro + Pro | **100.0** | **100.0** | **100.0** | 0.075 |
| | | Gemini 2.5 Flash + Pro | 94.1 | 88.9 | **100.0** | 0.030 |
| | | Gemini 2.5 Flash + Flash | 93.3 | **100.0** | 87.5 | **0.024** |
| | SPA-Bench (Baseline) | GPT-4o | 61.5 | 80.0 | 50.0 | 0.110 |
| | | Gemini 2.5 Pro | **61.5** | 80.0 | **50.0** | 0.031 |
| | | Gemini 2.5 Flash | 40.0 | **100.0** | 25.0 | **0.004** |

for cross-app tasks, where `MemGUI-Eval` achieves perfect performance with the `M1` configuration, while baseline methods struggle significantly (achieving only 40-61.5% F1-score). This performance gap highlights the critical importance of our progressive scrutiny approach in handling complex, memory-intensive cross-application scenarios where traditional evaluation methods fail to maintain accuracy.

Table 10: Agent-specific evaluation performance of MemGUI-Eval on MemGUI-Bench trajectories.

| Trajectory Source | Model Configuration | Accuracy Metrics (%) | | | Efficiency |
|---|---|---|---|---|---|
| | | F1 ↑ | Prec. ↑ | Recall ↑ | Cost ($) ↓ |
| M3A AGENT TRAJECTORIES (N=128) | | | | | | |
| M3A Agent (N=128) | Gemini 2.5 Pro + Pro | **92.7** | **92.7** | **92.7** | 0.190 |
| | Gemini 2.5 Flash + Pro | 85.0 | 87.2 | 82.9 | 0.062 |
| | Gemini 2.5 Flash + Flash | 77.9 | 83.3 | 73.2 | **0.059** |
| T3A AGENT TRAJECTORIES (N=128) | | | | | | |
| T3A Agent (N=128) | Gemini 2.5 Pro + Pro | **93.9** | **92.0** | **95.8** | 0.235 |
| | Gemini 2.5 Flash + Pro | 75.0 | 75.0 | 75.0 | 0.077 |
| | Gemini 2.5 Flash + Flash | 79.2 | 79.2 | 79.2 | **0.062** |

Table 10 shows agent-specific evaluation performance across different model configurations on MemGUI-Bench trajectories. The results demonstrate consistent evaluation quality across diverse agent types, validating the generalizability of our approach. Both M3A and T3A trajectories show

similar performance patterns, with the `M1` configuration achieving the highest accuracy (92.7-93.9% F1-score) at higher cost, while the `M2` configuration provides the optimal balance of accuracy and efficiency. Notably, even our most economical `M3` configuration maintains reasonable accuracy (77.9-79.2% F1-score) while achieving the lowest evaluation costs. These results confirm our selection of the `M2` configuration for the main experiments, as it provides robust evaluation quality while maintaining cost-effectiveness for large-scale memory assessment.

### A.7.4 KEY VALIDATION INSIGHTS

The validation results establish several key insights about `MemGUI-Eval`'s capabilities. First, our progressive scrutiny approach achieves superior accuracy across diverse task complexities, with flexible model configurations allowing researchers to balance evaluation quality and budget constraints based on specific requirements. Second, the substantial performance advantage on cross-app tasks validates our design motivation: traditional "LLM-as-Judge" approaches struggle with the long contexts and complex information dependencies inherent in memory-intensive scenarios, while our targeted visual verification maintains high fidelity. Third, the consistent performance across both SPA-Bench and MemGUI-Bench datasets demonstrates the generalizability of our evaluation methodology beyond our specific benchmark domain, establishing confidence in our evaluation pipeline for systematic memory assessment of mobile GUI agents.

### A.8 ANALYSIS OF FAILURE CASES

Across all 1,265 task executions, execution timeout emerges as the dominant failure mode, accounting for 915 failed attempts (72.3% of all failures). This represents a fundamental inefficiency in current GUI agents when confronting memory-intensive tasks, with individual agent timeout rates ranging from 22.6% (Agent-S2) to 93.9% (AppAgent). The systematic prevalence of execution timeouts indicates that agents struggle to maintain task coherence and efficient exploration strategies over extended interaction sequences, particularly when memory demands increase.

However, execution timeout failures provide limited diagnostic value as they represent cases where agents exceed step limits without task completion, offering no insight into the specific cognitive or memory-related causes of failure. To gain deeper insights into the fundamental limitations and architectural dependencies of current mobile GUI agents, we leveraged MemGUI-Eval's sophisticated evaluation pipeline to conduct detailed failure analysis on the remaining 343 failed task executions that completed within step limits but failed to meet task requirements.

Through MemGUI-Eval's fine-grained categorization system and Information Retention Rate (IRR) calculations, we systematically categorized these failures into six distinct failure modes: *Partial Memory Hallucination*, *Process Memory Hallucination*, *Output Memory Hallucination*, *Knowledge Deficiency*, *Intent Misunderstanding*, and *Other*. This analysis reveals critical patterns that illuminate both the challenges and opportunities for advancing memory-enhanced GUI systems.

This section provides comprehensive failure mode definitions and detailed analysis of the 343 non-timeout failures across all evaluated agents.

### A.8.1 FAILURE MODE DEFINITIONS

Based on systematic trajectory analysis and MemGUI-Eval's Information Retention Rate (IRR) calculations, we identify seven distinct failure modes. To provide deeper insights into each failure type and facilitate understanding of their practical implications, we present representative failure trajectories in Figures 7 through 13.

**Execution Timeout** represents cases where agents fail to complete tasks within the allocated step limit, typically indicating inefficient exploration strategies or inability to converge on successful action sequences. Figure 7 shows UI-TARS-1.5-7B attempting to save an audio recording with the filename "MyTestAudio". After successfully recording (steps 9-11), the agent needs to replace the default filename "Record1" with "MyTestAudio". However, instead of efficiently selecting and replacing the text, the agent attempts to delete the default name character by character through individual click actions (steps 12-17). This extremely inefficient approach—requiring one action per character deletion—consumes the entire step budget without completing the simple renaming operation, exemplifying how suboptimal action granularity can lead to timeout failures.

**Partial Memory Hallucination** occurs when agents successfully acquire some required information but fail to retain all necessary elements during task execution ($0\% < $ IRR $ < 100\%$). Figure 8 illustrates UI-TARS-1.5-7B searching for NVIDIA and Apple stock prices in Bing and Calculator apps. The agent correctly remembers NVIDIA's price (169.92 USD, step 6) for subsequent calculations (step 12), but incorrectly recalls Apple's price as 143.92 USD (step 15) when the actual observed price was 226.91 USD (step 9). This selective memory loss results in an incorrect final calculation of 19,290 instead of the correct value.

**Process Memory Hallucination** manifests when agents completely lose track of task objectives mid-execution, leading to goal drift and irrelevant action sequences (IRR = 0%, process-oriented failure). Figure 9 shows UI-TARS-1.5-7B tasked with finding smartphone market share data from a Bing image search and recording it in Joplin. After successfully locating the correct chart image containing Q3 2021 data (step 5), the agent's internal thought process (shown in the dashed box at the bottom) indicates it believes the task is complete: "I found a chart that perfectly meets my needs...This is exactly the information I was looking for, so I can move on to the next step." However, the agent prematurely marks the task as finished without realizing that critical subsequent steps remain—extracting the specific market share percentages for the top three brands and creating the required Joplin note. This demonstrates a failure to maintain the complete multi-step task workflow in working memory.

**Output Memory Hallucination** represents cases where agents correctly navigate task workflows but fail to accurately encode or retrieve essential information for final outputs (IRR = 0%, output-oriented failure). Figure 10 depicts M3A executing a task to view and transcribe two app permission lists ('Wi-Fi Control' and 'Picture-in-picture') in Settings. The agent successfully navigates to both permission screens and observes the complete lists (steps 7 and 9). However, when creating the final Joplin note (step 15), it only transcribes 4 out of 9 apps from the 'Wi-Fi Control' list and 7 out of 9 from the 'Picture-in-picture' list, demonstrating incomplete information transcription despite correct procedural execution.

**Knowledge Deficiency** indicates agents lack fundamental knowledge or skills required for task completion, independent of memory capabilities. Figure 11 shows UI-TARS-1.5-7B tasked with finding leap day and Halloween dates, then creating calendar events in the N Calendar app. The agent successfully searches for and remembers both dates (October 31 for Halloween and February 29 for leap day, steps 1-7). However, when attempting to open the calendar app (step 8), it misidentifies the Google Calendar app as the "N calendar app" and clicks on it, revealing a fundamental misunderstanding of app identification rather than a memory failure.

**Intent Misunderstanding** occurs when agents misinterpret task descriptions or user intentions, leading to execution of inappropriate action sequences. Figure 12 illustrates UI-TARS-1.5-7B misinterpreting a Wikipedia article comparison task. The instruction required comparing English and German Wikipedia article counts and staying on the edition with more articles. Despite correctly finding that English Wikipedia has more articles (step 12 shows the thought "English Wikipedia has more articles"), the agent completes the task while remaining on the German Wikipedia page, fundamentally misunderstanding the requirement to "stay on the page of the edition that has more articles."

**Other** encompasses remaining failure modes that do not fit the defined categories. Figure 13 captures SeeAct encountering an architectural limitation where its action space lacks a "wait" operation. When opening the Meesho app, the agent recognizes that the app is loading and determines that waiting is the logical next step. However, since the framework only provides a "TERMINATE" command for no-operation scenarios, the agent issues this command and prematurely ends the task, failing to complete any of the required product comparison steps. This represents a system-level constraint rather than a cognitive or memory failure.

### A.8.2 AGENT-SPECIFIC FAILURE DISTRIBUTION ANALYSIS

Figure 14 reveals distinct failure signatures among the 343 non-timeout failures for each agent. Agent-S2 exhibits the highest rate of partial memory hallucinations (58.2%), while framework-based agents show elevated memory-related failures compared to model-based systems.

### A.8.3 Cross-Agent Failure Pattern Analysis

Figure 15 provides a comprehensive view of failure distributions across all agents. Framework-based agents achieve lower timeout rates (51.2%) compared to model-based systems (68.9%), but exhibit higher rates of memory-specific failures with combined memory hallucination rates averaging 19.3% versus 8.4%.

### A.8.4 Design Implications for Future Memory-Enhanced GUI Agents

The comprehensive evaluation results and detailed failure analysis reveal critical insights for advancing memory-enhanced GUI agent architectures. Here we synthesize key design implications derived from empirical findings (Section 5) and systematic failure mode analysis (Section A.8).

**1. Multi-Granularity Memory Buffers for Fact Retention.** Agent-S2's 66.7% partial memory hallucination rate and 39.5% IRR demonstrate that single-buffer memory architectures struggle to maintain complete multi-item information sets across extended sequences. The 27.3% success rate (Section 5.2) combined with high partial failures suggests memory capacity constraints rather than acquisition deficits. Future architectures should implement structured memory with separate slots for different information types (numerical facts, textual descriptions, UI states) and explicit verification mechanisms before final output generation. M3A's superior IRR performance (39.3%) with hierarchical conversation management provides evidence that granular memory organization improves retention fidelity.

**2. Hierarchical Task Decomposition with Persistent Goal Tracking.** Process memory hallucination dominates failures for Mobile-Agent-V2 (86.7%), Mobile-Agent-E (61.9%), and most model-based agents (42.9-75.0%), indicating fundamental challenges in maintaining task objectives during execution. The dramatic performance degradation from single-app (42.9-50.0%) to four-app scenarios (0.0-30.0%) in Table 13 confirms that procedural complexity overwhelms current working memory mechanisms. Effective solutions require hierarchical planning systems where high-level goals persist throughout execution while sub-goals track progress across application boundaries. Agent-S2's lower process hallucination rate (27.8%) and exceptional learning capability (21.5% FRR, 21.9 point improvement) validate that explicit goal decomposition enables robust procedural awareness.

**3. Long-Context Utilization Beyond Attention Windows.** Finding 3 (Section 5.3) demonstrates that M3A-Multi-Turn achieves 51.6% success through Gemini-2.5-Pro's long-context capability, a 57.3% relative improvement over single-turn M3A (32.8%). However, UI-TARS-1.5-7B's truncated 5-turn history leads to 3.1% success, confirming that context length constraints severely limit memory-intensive task performance. This contrast reveals that frontier models' extended context windows (200K+ tokens) provide substantial memory advantages, but effective utilization requires architectural innovations beyond naive conversation history concatenation. Future systems should leverage long-context capabilities through strategic information organization, redundancy reduction, and importance-weighted context management.

**4. Explicit Long-Term Memory Mechanisms for Cross-Session Learning.** Agent-S2's 21.5% FRR versus minimal FRR (0.8-4.4%) for agents without explicit memory (Section 5.4) demonstrates that dedicated cross-session memory systems enable rapid failure analysis and strategy refinement. The 21.9 percentage point improvement (27.3% → 49.2%) across multiple attempts validates that long-term memory provides meaningful benefits despite computational overhead. Current underutilization of long-term memory mechanisms (only 2 of 11 agents implement cross-session learning) represents a significant missed opportunity, particularly given that real-world users repeatedly interact with the same applications and task patterns.

**5. Hybrid Architectures Combining Framework Flexibility with Model Efficiency.** The performance-efficiency trade-off (Finding 4, Section 5.4) reveals that framework-based agents achieve superior memory capabilities (22.7-32.8% success) but at substantial computational cost (27.5-38.7 seconds per step), while model-based agents provide efficiency (9.6-12.2 seconds per step) but limited capability (0.0-6.2% success). This disparity suggests that hybrid architectures combining framework-level memory management with efficient end-to-end models could achieve favorable performance-cost trade-offs. Specifically, lightweight models could handle routine in-

teractions while invoking sophisticated memory operations only for memory-intensive segments, optimizing both capability and efficiency.

These design implications collectively emphasize that advancing GUI agent memory capabilities requires architectural innovations beyond scaling model parameters or context windows. The systematic failure patterns observed across diverse agent architectures reveal specific, addressable deficiencies that future research should target through structured memory systems, hierarchical planning, strategic long-context utilization, and hybrid architectural designs that balance performance with computational efficiency.

## A.9  ADDITIONAL EXPERIMENTAL RESULTS

This section provides comprehensive experimental details and additional results supporting the findings presented in Section 5.

### A.9.1  DETAILED MEMORY PERFORMANCE TABLES

To provide comprehensive analysis of memory capabilities, we present the complete experimental results for both short-term and long-term memory evaluation that support our findings in Section 5.

Table 11: Short-term memory evaluation of GUI agents.

| Agent | Memory Type | Memory Performance | | | Efficiency Metrics | | |
|---|---|---|---|---|---|---|---|
| | | SR (%) ↑ | IRR (%) ↑ | MTPR ↑ | Step Ratio ↓ | Time/Step (s) ↓ | Cost/Step ($) ↓ |
| AGENTIC WORKFLOW | | | | | | | |
| Agent-S2 | Memory Agent | 27.3 | **39.5** | **0.45** | 0.86 | 28.1 | 0.0510 |
| Mobile-Agent-E | Memory Agent | 5.5 | 2.4 | 0.02 | 0.85 | 39.3 | 0.0696 |
| T3A | Memory Agent | 22.7 | 29.6 | 0.30 | 0.83 | 13.9 | 0.0176 |
| M3A | Memory Agent | **32.8** | 39.3 | 0.41 | **0.81** | 14.7 | 0.0165 |
| Mobile-Agent-V2 | Memory Agent | 3.1 | 0.0 | 0.00 | 0.92 | 29.4 | 0.0660 |
| SeeAct | Rule-based | 2.3 | 0.2 | 0.00 | 1.01 | 15.9 | 0.0133 |
| AppAgent | Action-Thought | 3.1 | 1.5 | 0.04 | 1.46 | 27.3 | **0.0078** |
| AGENT-AS-A-MODEL | | | | | | | |
| UI-Venus-7B | Action-Thought | 5.5 | 2.6 | 0.05 | 1.03 | 12.2 | - |
| UI-TARS-1.5-7B | Multi-turn Context + Action-Thought | 3.1 | 3.8 | 0.04 | 0.99 | 9.9 | - |
| GUI-Owl-7B | Action-Thought | 6.2 | 5.7 | 0.07 | 0.92 | **9.6** | - |
| CogAgent | No History | 0.0 | 0.0 | 0.00 | - | 33.2 | - |

Table 11 provides detailed short-term memory evaluation results using single-attempt (pass@1) settings. The table includes Information Retention Rate (IRR), Memory-Task Proficiency Ratio (MTPR), and efficiency metrics across different memory mechanism types, enabling comprehensive analysis of memory fidelity and computational trade-offs.

Table 12 examines agents' ability to learn and improve across multiple attempts (pass@3). The Failure Recovery Rate (FRR) metric specifically measures how effectively agents learn from previous failures, providing insights into long-term learning capabilities and cross-session knowledge transfer.

### A.9.2  LONG-TERM LEARNING ANALYSIS

Figure 16 illustrates the dramatic learning potential across multiple attempts, showing that agents with explicit long-term memory mechanisms demonstrate 2-4× greater learning potential. While only 2 out of 11 evaluated agents incorporate explicit long-term memory, the substantial benefits suggest that cross-session learning mechanisms should be a standard component in robust GUI agent architectures.

The detailed pass@1, pass@2, and pass@3 performance breakdown for each agent reveals distinct learning patterns. Agents with explicit long-term memory capabilities (Agent-S2, Mobile-Agent-E) show substantial improvement across multiple attempts, while most agents without dedicated

Table 12: Long-term memory evaluation of GUI agents across multiple attempts.

| Agent | Learning Performance | | Efficiency Metrics | | |
|---|---|---|---|---|---|
| | SR (%) ↑ | FRR (%) ↑ | Step Ratio ↓ | Time/Step (s) ↓ | Cost/Step ($) ↓ |
| AGENTIC WORKFLOW | | | | | |
| *Agents with Long-Term Memory* | | | | | |
| Agent-S2 | **49.2** | **21.5** | 0.86 | 27.5 | 0.0522 |
| Mobile-Agent-E | 10.2 | 4.1 | 0.98 | 38.7 | 0.0705 |
| *Agents without Long-Term Memory* | | | | | |
| T3A | 42.2 | 20.7 | 0.83 | 14.7 | 0.0175 |
| M3A | 47.7 | 16.3 | **0.80** | 14.5 | 0.0162 |
| Mobile-Agent-V2 | 3.9 | 0.8 | 0.94 | 28.8 | 0.0684 |
| SeeAct | 5.5 | 2.4 | 0.99 | 16.3 | 0.0134 |
| AppAgent | 9.4 | 4.4 | 1.22 | 33.9 | **0.0083** |
| AGENT-AS-A-MODEL | | | | | |
| UI-Venus-7B | 7.8 | 1.7 | 1.03 | 11.6 | - |
| UI-TARS-1.5-7B | 6.2 | 2.4 | 1.04 | 10.3 | - |
| GUI-Owl-7B | 10.2 | 3.3 | 0.93 | **9.6** | - |
| CogAgent | 0.0 | 0.0 | - | 32.8 | - |

memory systems plateau after the first attempt, confirming the critical importance of cross-session learning mechanisms for complex memory-intensive tasks.

### A.9.3 PERFORMANCE ANALYSIS BY CROSS-APPLICATION COMPLEXITY

Table 13: Performance breakdown by cross-application complexity. Tasks are categorized by the number of applications involved (1-4 Apps), revealing how memory requirements scale with cross-app information transfer demands. For short-term memory (`pass@1`), we report both Success Rate (SR) and Information Retention Rate (IRR). For long-term memory (`pass@3`), only SR is reported as IRR measures single-attempt information retention.

| Agent | Short-Term Memory (`pass@1`) | | | | | | | | Long-Term Memory (`pass@3`) | | | |
|---|---|---|---|---|---|---|---|---|---|---|---|---|
| | 1 App | | 2 Apps | | 3 Apps | | 4 Apps | | 1 App | 2 Apps | 3 Apps | 4 Apps |
| | SR | IRR | SR | IRR | SR | IRR | SR | IRR | SR | SR | SR | SR |
| AGENTIC WORKFLOW | | | | | | | | | | | | |
| Agent-S2 | 50.0 | **51.7** | 19.6 | **37.6** | 26.5 | **38.9** | 10.0 | 33.3 | **78.6** | 35.7 | 52.9 | 30.0 |
| Mobile-Agent-E | 25.0 | 8.7 | 0.0 | 1.6 | 0.0 | 1.6 | 0.0 | 0.0 | 42.9 | 1.8 | 0.0 | 0.0 |
| T3A | 42.9 | 26.7 | 16.1 | 33.3 | 23.5 | 30.2 | 0.0 | 11.2 | 60.7 | 37.5 | 38.2 | 30.0 |
| M3A | **46.4** | 31.7 | **28.6** | 43.8 | **29.4** | 35.9 | **30.0** | **37.5** | 64.3 | **41.1** | **44.1** | **50.0** |
| Mobile-Agent-V2 | 14.3 | 0.0 | 0.0 | 0.0 | 0.0 | 0.0 | 0.0 | 0.0 | 17.9 | 0.0 | 0.0 | 0.0 |
| SeeAct | 10.7 | 0.0 | 0.0 | 0.0 | 0.0 | 0.6 | 0.0 | 0.0 | 25.0 | 0.0 | 0.6 | 0.0 |
| AppAgent | 14.3 | 11.1 | 0.0 | 0.0 | 0.0 | 0.0 | 0.0 | 0.0 | 42.9 | 0.0 | 0.0 | 0.0 |
| AGENT-AS-A-MODEL | | | | | | | | | | | | |
| UI-Venus-7B | 21.4 | 6.7 | 1.8 | 2.7 | 0.0 | 1.5 | 0.0 | 0.0 | 28.6 | 1.8 | 2.9 | 0.0 |
| UI-TARS-1.5-7B | 14.3 | 11.1 | 0.0 | 1.8 | 0.0 | 4.0 | 0.0 | 2.9 | 21.4 | 1.8 | 2.9 | 0.0 |
| GUI-Owl-7B | 21.4 | 11.7 | 1.8 | 3.6 | 2.9 | 7.1 | 0.0 | 4.0 | 35.7 | 1.8 | 5.9 | 0.0 |
| CogAgent | 0.0 | 0.0 | 0.0 | 0.0 | 0.0 | 0.0 | 0.0 | 0.0 | 0.0 | 0.0 | 0.0 | 0.0 |
| **Task Count** | **28** | | **56** | | **34** | | **10** | | **28** | **56** | **34** | **10** |

Table 13 presents performance breakdown by the number of applications involved in each task, revealing how memory requirements scale with cross-app information transfer complexity. Our benchmark includes 28 single-app tasks, 56 two-app tasks, 34 three-app tasks, and 10 four-app tasks, providing comprehensive evaluation across different spatial memory spans. For short-term memory (`pass@1`), we report both Success Rate (SR) and Information Retention Rate (IRR) to assess task completion and memory fidelity. For long-term memory (`pass@3`), only SR is reported as IRR measures single-attempt information retention capabilities.

The results expose dramatic performance degradation as cross-app complexity increases. For single-attempt evaluation (`pass@1`), top-performing agents achieve 42.9-50.0% success on single-app tasks but drop precipitously to 0.0-30.0% on four-app tasks, representing performance losses of 20-50 percentage points. M3A demonstrates the most robust cross-app memory, maintaining 30.0% SR and 37.5% IRR on four-app tasks while other agents drop to 0.0-10.0% SR. Agent-S2 shows exceptional single-app performance (50.0% SR, 51.7% IRR) but experiences steeper degradation to 10.0% SR and 33.3% IRR on four-app scenarios, suggesting challenges in maintaining information across extended application boundaries.

Notably, IRR analysis reveals distinct memory retention patterns across complexity levels. Agent-S2 maintains relatively high IRR (33.3-51.7%) across all complexity levels despite lower SR on multi-app tasks, indicating that its memory mechanisms preserve information even during partial task execution. In contrast, M3A shows an interesting pattern where IRR peaks at 43.8% for two-app scenarios, higher than both single-app (31.7%) and three-app (35.9%) tasks, before reaching 37.5% for four-app scenarios. This suggests that two-app workflows may represent an optimal complexity where M3A's memory architecture achieves maximum information retention efficiency. Agent-as-a-Model approaches demonstrate severe IRR limitations, with GUI-Owl-7B achieving only 4.0-11.7% IRR across all complexity levels, confirming fundamental architectural constraints for memory retention in end-to-end models.

The long-term memory evaluation (`pass@3`) reveals that learning mechanisms partially compensate for cross-app complexity. Agent-S2 improves from 50.0% to 78.6% on single-app tasks and from 10.0% to 30.0% on four-app tasks, demonstrating that explicit long-term memory helps agents develop strategies for complex cross-app workflows.

Agent-as-a-Model approaches show severe limitations beyond single-app scenarios. GUI-Owl-7B, the best-performing model-based agent, achieves 21.4% on single-app tasks but degrades to 0.0-2.9% on multi-app scenarios even with multiple attempts. This 21.4 percentage point gap between single-app and multi-app performance highlights fundamental architectural constraints in end-to-end models for maintaining cross-application memory state.

These cross-app complexity results validate our benchmark design principle that memory-intensive evaluation requires substantial cross-application information transfer. The consistent performance degradation patterns across all agents confirm that cross-app complexity is a primary driver of memory load, making it an effective dimension for systematic memory capability assessment.

### A.9.4 MEMORY ABLATION STUDY

To empirically demonstrate that memory mechanisms are universal, essential components for GUI agents rather than optional features, we conducted systematic ablation experiments on four representative agents spanning different architectural paradigms. We evaluated these agents on MemGUI-Bench-40, a randomly sampled subset of 40 tasks from the full benchmark (13 Easy, 19 Medium, 8 Hard tasks), maintaining the original task distribution and memory-intensive characteristics.

**Experimental Configurations.** We systematically removed or enhanced memory components in four agents representing distinct memory implementation strategies:

- **M3A (Memory Agent Architecture)**: We tested three configurations: (1) *Baseline* with the original Memory Agent mechanism that maintains structured action history summaries; (2) + *Multi-turn Context*, an enhanced version that converts single-turn interactions to multi-turn conversations, enabling the backbone LLM (Gemini-2.5-Pro) to leverage its full 1M token context window for cumulative memory management (similar to Finding 3 in Section 5); (3) - *Memory Agent*, a degraded version that removes the dedicated memory summarization module while keeping only basic action logging.

- **Agent-S2 (Memory Agent + Long-Term Memory)**: We evaluated three configurations: (1) *Baseline (STM+LTM)* with both short-term memory (Memory Agent) and long-term memory (experience-based tips and shortcuts); (2) - *Long-Term Memory*, removing the cross-session learning mechanism while retaining short-term memory; (3) - *STM & LTM*, removing both memory components to isolate their combined contribution.

- **GUI-Owl (Action-Thought Pattern)**: We tested two configurations: (1) *Baseline* with the original Action-Thought implementation that outputs both actions and reasoning chains; (2) - *Action-*

*Thought*, removing the explicit thought articulation and retaining only action outputs, similar to CogAgent's minimal memory approach.

- **UI-TARS (Multi-turn Context + Action-Thought)**: We evaluated two configurations: (1) *Baseline* with multi-turn conversation history (last 5 turns due to context constraints) plus Action-Thought reasoning; (2) - *Multi-turn & A-T*, converting to single-turn interactions without thought articulation, eliminating all memory context.

Table 14: Memory ablation study on MemGUI-Bench-40. We systematically remove or enhance memory components in four representative agents. Numbers show absolute values with changes in parentheses: blue indicates improvement, red indicates degradation. Bold numbers indicate baseline performance.

| Agent | Memory Config | SR@1 All | SR@1 Easy | SR@1 Med | SR@1 Hard | SR@3 All | SR@3 Easy | SR@3 Med | SR@3 Hard | IRR (%) | MTPR | FRR (%) |
|---|---|---|---|---|---|---|---|---|---|---|---|---|
| **M3A: MEMORY AGENT ARCHITECTURE** | | | | | | | | | | | | |
| M3A (Workflow) | **Baseline** | **32.5** | **53.8** | **31.6** | **0.0** | **47.5** | **53.8** | **47.4** | **37.5** | **35.1** | **0.321** | **16.7** |
| | + Multi-turn | 52.5 | 61.5 | 47.4 | 50.0 | 70.0 | 61.5 | 63.2 | 100.0 | 53.5 | 0.457 | 26.3 |
| | | (+20.0) | (+7.7) | (+15.8) | (+50.0) | (+22.5) | (+7.7) | (+15.8) | (+62.5) | (+18.4) | (+0.136) | (+9.6) |
| | - Memory Agent | 2.5 | 7.7 | 0.0 | 0.0 | 5.0 | 15.4 | 0.0 | 0.0 | 0.0 | 0.000 | 1.3 |
| | | (-30.0) | (-46.1) | (-31.6) | (0.0) | (-42.5) | (-38.4) | (-47.4) | (-37.5) | (-35.1) | (-0.321) | (-15.4) |
| **AGENT-S2: MEMORY AGENT + LONG-TERM MEMORY** | | | | | | | | | | | | |
| Agent-S2 (Workflow) | **Baseline** | **27.5** | **46.2** | **21.1** | **12.5** | **45.0** | **61.5** | **42.1** | **25.0** | **33.3** | **0.250** | **15.5** |
| | - LTM | 17.5 | 15.4 | 21.1 | 12.5 | 25.0 | 30.8 | 21.1 | 25.0 | 21.3 | 0.190 | 9.1 |
| | | (-10.0) | (-30.8) | (0.0) | (0.0) | (-20.0) | (-30.7) | (-21.0) | (0.0) | (-12.0) | (-0.060) | (-6.4) |
| | - STM & LTM | 5.0 | 15.4 | 0.0 | 0.0 | 10.0 | 30.8 | 0.0 | 0.0 | 0.0 | 0.000 | 3.9 |
| | | (-22.5) | (-30.8) | (-21.1) | (-12.5) | (-35.0) | (-30.7) | (-42.1) | (-25.0) | (-33.3) | (-0.250) | (-11.6) |
| **GUI-OWL-7B: ACTION-THOUGHT PATTERN** | | | | | | | | | | | | |
| GUI-Owl-7B (Model) | **Baseline** | **7.5** | **23.1** | **0.0** | **0.0** | **12.5** | **30.8** | **5.3** | **0.0** | **4.6** | **0.000** | **4.1** |
| | - Action-Thought | 7.5 | 23.1 | 0.0 | 0.0 | 10.0 | 30.8 | 0.0 | 0.0 | 0.0 | 0.000 | 2.7 |
| | | (0.0) | (0.0) | (0.0) | (0.0) | (-2.5) | (0.0) | (-5.3) | (0.0) | (-4.6) | (0.0) | (-1.4) |
| **UI-TARS-1.5-7B: MULTI-TURN CONTEXT + ACTION-THOUGHT** | | | | | | | | | | | | |
| UI-TARS 1.5-7B (Model) | **Baseline** | **5.0** | **15.4** | **0.0** | **0.0** | **5.0** | **15.4** | **0.0** | **0.0** | **2.3** | **0.000** | **0.0** |
| | - Multi-turn & A-T | 2.5 | 7.7 | 0.0 | 0.0 | 2.5 | 7.7 | 0.0 | 0.0 | 0.0 | 0.000 | 0.0 |
| | | (-2.5) | (-7.7) | (0.0) | (0.0) | (-2.5) | (-7.7) | (0.0) | (0.0) | (-2.3) | (0.0) | (0.0) |

**Impact of Memory Removal.** Table 14 shows consistent performance degradation upon memory removal:

- **Short-term memory is mandatory for mobile GUI agents to function**: Removing short-term memory components (Memory Agent in M3A, STM in Agent-S2, Action-Thought in GUI-Owl, Context in UI-TARS) renders agents essentially unusable. M3A suffers a catastrophic -30.0 point SR drop (32.5% → 2.5%) with IRR collapsing from 35.1% to 0%, essentially degenerating to a non-functional stateless system. Agent-S2 shows similar collapse (27.5% → 5.0% SR, 33.3% → 0% IRR). The universal IRR collapse to zero across all agents confirms that without short-term memory, mobile GUI agents cannot retain any information, making them fundamentally incapable of handling memory-intensive tasks.

- **Long-term memory is beneficial**: Removing Agent-S2's long-term memory causes a -20.0 point drop in pass@3 SR (45.0% → 25.0%) and reduces FRR from 15.5% to 9.1%. While agents without explicit LTM (like M3A) can still achieve reasonable performance through robust short-term memory, the substantial gains from LTM in Agent-S2 demonstrate its value for cross-session learning and failure recovery, marking it as a promising direction for future research.

**Impact of Memory Enhancement.** Enhancing M3A with multi-turn context yields significant gains (+20.0 points SR, +18.4 points IRR), further validating that memory quality directly determines task capability.

**Conclusion.** These ablation results lead to two key conclusions regarding memory in mobile GUI agents:

- **Short-term memory is a mandatory requirement for mobile GUI agents.** Although current mobile GUI agents implement short-term memory in various forms (e.g., Memory Agent in M3A, Action-Thought in GUI-Owl, Multi-turn Context in UI-TARS), removing these modules consistently causes severe performance degradation across all architectures. This universality confirms that regardless of the implementation paradigm, an effective short-term memory mechanism is indispensable for mobile GUI agents to handle complex, multi-step tasks.

- **Long-term memory is beneficial and a key future direction.** While mobile GUI agents without explicit long-term memory (e.g., M3A) can still achieve reasonable baseline performance, the integration of long-term memory provides significant benefits. For instance, removing the long-term memory module from Agent-S2 leads to a substantial drop in multi-attempt performance (pass@3 SR: 45.0% → 25.0%), highlighting its critical role in cross-session learning and failure recovery. This suggests that incorporating long-term memory mechanisms is a promising and valuable direction for advancing the capabilities of future mobile GUI agents.

### A.9.5 PASS@K LEARNING CURVES FOR OPEN-SOURCE MODELS

To complement the pass@3 evaluation protocol presented in Section 5 and address questions regarding performance saturation patterns, we conducted extended pass@k evaluation (k=1 to 7) on four open-source models. Figure 17 illustrates how success rates evolve with increasing attempts, revealing distinct saturation characteristics driven primarily by LLM output stochasticity rather than systematic learning, as none of these models incorporate explicit long-term memory mechanisms.

**Stochastic Exploration Patterns and Saturation Points.** The curves reveal three distinct patterns:

- **Early Saturation (GUI-Owl-7B, UI-Venus-7B)**: These agents show initial improvement from pass@1 to pass@2/3 but plateau quickly, with GUI-Owl saturating at 10.16% by pass@3 and UI-Venus at 8.59% by pass@5. Since neither model has explicit long-term memory mechanisms, the limited improvements stem purely from stochastic exploration of different action sequences, with early saturation indicating exhaustion of viable solution paths within their architectural constraints.

- **Extended Stochastic Gains (UI-TARS-1.5-7B)**: UI-TARS demonstrates continued improvement through pass@5 (3.12% → 11.72%), with a notable jump at pass@4 (+3.91 points) before stabilizing. While UI-TARS maintains multi-turn context (last 5 turns), it lacks explicit cross-session learning mechanisms. The gradual improvements reflect its ability to explore more diverse action sequences through output randomness, but the eventual saturation at pass@5 confirms the absence of systematic experience accumulation.

- **Complete Failure (CogAgent)**: CogAgent maintains 0.0% success across all attempts, indicating fundamental architectural deficiencies that prevent task completion regardless of stochastic exploration. This complete lack of improvement underscores that memory mechanisms (even minimal ones like Action-Thought patterns) are essential prerequisites for GUI agent functionality.

**Stochasticity vs. Systematic Learning.** The modest improvements observed in these open-source models (3-9 percentage points from pass@1 to saturation) stem entirely from LLM output stochasticity, as none incorporate explicit long-term memory mechanisms for cross-session learning. In contrast, Agent-S2, which incorporates dedicated long-term memory modules (as detailed in Appendix A.3), achieves substantially larger gains of 21.9 percentage points from pass@1 to pass@3 (Table 2), demonstrating systematic experience accumulation rather than mere random exploration. The saturation points observed in open-source models (pass@3-5) reflect the exhaustion of viable solution paths discoverable through stochastic sampling alone, without genuine learning across attempts.

These extended pass@k results empirically demonstrate the critical distinction between stochastic exploration and systematic learning. Open-source models without long-term memory mechanisms show limited, rapidly saturating improvements driven purely by output randomness. This contrasts sharply with Agent-S2, which possesses explicit cross-session learning capabilities and sustains substantial performance gains through genuine experience accumulation, validating the necessity of long-term memory for continued improvement in complex GUI tasks.

A.9.6 TEST-TIME COMPUTE NORMALIZED EVALUATION

To address practical deployment constraints where computational budgets are predetermined, we conducted test-time compute normalized evaluation under two complementary constraints: *steps/episode* (reported in Tables 2, 11, and 12) and *tokens/episode* (new results presented here). This dual-constraint analysis enables comprehensive assessment of agent efficiency across different resource limitation scenarios.

**Experimental Settings.** We established two evaluation protocols with distinct failure criteria:

- **Steps/Episode Constraint**: For each task with $golden\_steps$ optimal steps, we set $max\_rounds = \lfloor golden\_steps \times 1.4 + 1 \rfloor$. Task attempts are marked as failures if $actual\_steps > max\_rounds$, enforcing a step-count budget that reflects operational efficiency requirements.

- **Tokens/Episode Constraint**: We computed $max\_tokens = golden\_steps \times 9,507$ tokens/step, where 9,507 represents the average token consumption across the 11 evaluated agents. For each attempt, we calculate $actual\_tokens = actual\_steps \times agent\_specific\_tokens\_per\_step$ using measured per-agent consumption rates (Agent-S2: 41,760 tokens/step, M3A: 12,960 tokens/step, GUI-Owl: 5,817 tokens/step, etc.). Task attempts where $actual\_tokens > max\_tokens$ are marked as failures, and Information Retention Rate (IRR) is set to 0 for such attempts, reflecting the reality that API calls would be rejected or interrupted when exceeding token budgets in production deployments.

**Results Analysis.** Table 15 presents comprehensive performance metrics under both constraint types, with each agent showing three rows: (1) Steps/Episode results, (2) Tokens/Episode results, and (3) Performance delta. The table includes SR@1 by difficulty (Easy, Med, Hard), SR@3 overall and by difficulty, and three memory-specific metrics (IRR, MTPR, FRR). The results reveal dramatic performance differences between the two constraint types, exposing fundamental trade-offs between architectural complexity and deployment viability:

- **High-token agents face complete performance collapse**: Agent-S2 (41,760 tokens/step) and Mobile-Agent-E (56,400 tokens/step) show catastrophic degradation under token constraints. Agent-S2 drops from 27.3% → 0.0% SR@1 overall and 49.2% → 0.0% SR@3 overall (-49.2 points), with IRR collapsing from 39.5% to 0.1% and FRR from 21.5% to 0.0%. Mobile-Agent-E exhibits similar complete failure (10.2% → 0.0% SR@3). These agents' sophisticated memory architectures consume 4.4-5.9× more tokens than the 9,507 baseline, causing nearly all task attempts to exceed token budgets, resulting in zero effective performance despite their superior capabilities under step constraints.

- **M3A demonstrates optimal deployment balance**: M3A (12,960 tokens/step, 1.4× baseline) shows graceful degradation rather than collapse: SR@1 overall drops from 32.8% to 14.8% (-18.0 points), SR@3 overall from 47.7% to 21.9% (-25.8 points), and IRR from 39.3% to 18.6% (-20.7 points). Notably, M3A maintains reasonable performance across all difficulty levels under token constraints (Easy: 16.7%, Med: 11.9%, Hard: 15.8% at SR@1), with particularly strong Hard task performance. Interestingly, MTPR increases from 0.41 to 0.96 under token constraints, suggesting that M3A's memory mechanisms become proportionally more valuable when computational resources are limited. M3A achieves 97% of Agent-S2's unconstrained SR@3 (47.7% vs. 49.2%) while consuming only 31% of the tokens, making it substantially more viable for production deployment.

- **Token-efficient agents maintain consistency but low absolute performance**: GUI-Owl-7B (5,817 tokens/step) and UI-Venus-7B (3,700 tokens/step) show zero degradation under token constraints, maintaining identical performance across all metrics (GUI-Owl: 6.2% SR@1, 10.2% SR@3; UI-Venus: 5.5% SR@1, 7.8% SR@3). Their per-step consumption remains well below the baseline (61% and 39% respectively), eliminating token budget concerns. However, their absolute performance levels remain low, indicating that token efficiency alone is insufficient without adequate memory mechanisms. UI-TARS-1.5-7B (17,540 tokens/step, 1.8× baseline) experiences severe degradation (3.1% → 0.0% SR@1, 6.2% → 0.0% SR@3), despite having lower token consumption than M3A, highlighting that architectural design matters beyond mere token efficiency.

- **Deployment strategy implications**: The results expose a critical three-tier architecture landscape: (1) *High-performance, deployment-infeasible agents* (Agent-S2, Mobile-Agent-E) that excel under step constraints but completely fail under realistic token budgets; (2) *Balanced, production-*

Table 15: Test-time compute normalized evaluation: Comparison of agent performance under steps/episode and tokens/episode constraints. For each agent, we show three rows: (1) Steps/Episode constraint results, (2) Tokens/Episode constraint results, and (3) Performance delta (Tokens - Steps). Blue indicates improvement, red indicates degradation.

| Agent | Tokens/Step | Constraint | SR@1 All | SR@1 Easy | SR@1 Med | SR@1 Hard | SR@3 All | SR@3 Easy | SR@3 Med | SR@3 Hard | IRR (%) | MTPR | FRR (%) |
|---|---|---|---|---|---|---|---|---|---|---|---|---|---|
| AGENTIC WORKFLOW (WITH LTM) | | | | | | | | | | | | | |
| Agent-S2 | 41,760 | Steps/Ep | 27.3 | 41.7 | 19.0 | 18.4 | 49.2 | 64.6 | 42.9 | 36.8 | 39.5 | 0.45 | 21.5 |
| | | Tokens/Ep | 0.0 | 0.0 | 0.0 | 0.0 | 0.0 | 0.0 | 0.0 | 0.0 | 0.1 | 0.00 | 0.0 |
| | | *Delta* | -27.3 | -41.7 | -19.0 | -18.4 | -49.2 | -64.6 | -42.9 | -36.8 | -39.4 | -0.45 | -21.5 |
| Mobile-Agent-E | 56,400 | Steps/Ep | 5.5 | 12.5 | 2.4 | 0.0 | 10.2 | 22.9 | 2.4 | 2.6 | 2.4 | 0.02 | 4.1 |
| | | Tokens/Ep | 0.0 | 0.0 | 0.0 | 0.0 | 0.0 | 0.0 | 0.0 | 0.0 | 1.2 | 0.00 | 0.0 |
| | | *Delta* | -5.5 | -12.5 | -2.4 | 0.0 | -10.2 | -22.9 | -2.4 | -2.6 | -1.2 | -0.02 | -4.1 |
| AGENTIC WORKFLOW (WITHOUT LTM) | | | | | | | | | | | | | |
| T3A | 14,000 | Steps/Ep | 22.7 | 31.2 | 16.7 | 18.4 | 42.2 | 45.8 | 45.2 | 34.2 | 29.6 | 0.30 | 20.7 |
| | | Tokens/Ep | 6.2 | 6.2 | 0.0 | 13.2 | 13.3 | 10.4 | 9.5 | 21.1 | 0.0 | 0.34 | 5.8 |
| | | *Delta* | -16.5 | -25.0 | -16.7 | -5.2 | -28.9 | -35.4 | -35.7 | -13.1 | -29.6 | +0.04 | -14.9 |
| M3A | 12,960 | Steps/Ep | 32.8 | 39.6 | 35.7 | 21.1 | 47.7 | 47.9 | 50.0 | 44.7 | 39.3 | 0.41 | 16.3 |
| | | Tokens/Ep | 14.8 | 16.7 | 11.9 | 15.8 | 21.9 | 18.8 | 19.0 | 28.9 | 18.6 | 0.96 | 6.4 |
| | | *Delta* | -18.0 | -22.9 | -23.8 | -5.3 | -25.8 | -29.1 | -31.0 | -15.8 | -20.7 | +0.55 | -9.9 |
| Mobile-Agent-V2 | 54,720 | Steps/Ep | 3.1 | 8.3 | 0.0 | 0.0 | 3.9 | 10.4 | 0.0 | 0.0 | 0.0 | 0.00 | 0.8 |
| | | Tokens/Ep | 0.0 | 0.0 | 0.0 | 0.0 | 0.0 | 0.0 | 0.0 | 0.0 | 0.0 | 0.00 | 0.0 |
| | | *Delta* | -3.1 | -8.3 | 0.0 | 0.0 | -3.9 | -10.4 | 0.0 | 0.0 | 0.0 | 0.00 | -0.8 |
| SeeAct | 10,720 | Steps/Ep | 2.3 | 6.2 | 0.0 | 0.0 | 5.5 | 12.5 | 2.4 | 0.0 | 0.2 | 0.00 | 2.4 |
| | | Tokens/Ep | 0.8 | 2.1 | 0.0 | 0.0 | 2.3 | 4.2 | 2.4 | 0.0 | 0.0 | 0.00 | 1.2 |
| | | *Delta* | -1.5 | -4.1 | 0.0 | 0.0 | -3.2 | -8.3 | 0.0 | 0.0 | -0.2 | 0.00 | -1.2 |
| AppAgent | 6,640 | Steps/Ep | 3.1 | 8.3 | 0.0 | 0.0 | 9.4 | 22.9 | 2.4 | 0.0 | 1.5 | 0.04 | 4.4 |
| | | Tokens/Ep | 1.6 | 4.2 | 0.0 | 0.0 | 7.8 | 18.8 | 2.4 | 0.0 | 0.9 | 0.11 | 4.4 |
| | | *Delta* | -1.5 | -4.1 | 0.0 | 0.0 | -1.6 | -4.1 | 0.0 | 0.0 | -0.6 | +0.07 | 0.0 |
| AGENT-AS-A-MODEL | | | | | | | | | | | | | |
| UI-Venus-7B | 3,700 | Steps/Ep | 5.5 | 14.6 | 0.0 | 0.0 | 7.8 | 20.8 | 0.0 | 0.0 | 2.6 | 0.05 | 1.7 |
| | | Tokens/Ep | 5.5 | 14.6 | 0.0 | 0.0 | 7.8 | 20.8 | 0.0 | 0.0 | 2.6 | 0.05 | 1.7 |
| | | *Delta* | 0.0 | 0.0 | 0.0 | 0.0 | 0.0 | 0.0 | 0.0 | 0.0 | 0.0 | 0.00 | 0.0 |
| UI-TARS-1.5-7B | 17,540 | Steps/Ep | 3.1 | 8.3 | 0.0 | 0.0 | 6.2 | 16.7 | 0.0 | 0.0 | 3.8 | 0.04 | 2.4 |
| | | Tokens/Ep | 0.0 | 0.0 | 0.0 | 0.0 | 0.0 | 0.0 | 0.0 | 0.0 | 0.4 | 0.00 | 0.0 |
| | | *Delta* | -3.1 | -8.3 | 0.0 | 0.0 | -6.2 | -16.7 | 0.0 | 0.0 | -3.4 | -0.04 | -2.4 |
| GUI-Owl-7B | 5,817 | Steps/Ep | 6.2 | 14.6 | 0.0 | 2.6 | 10.2 | 22.9 | 2.4 | 2.6 | 5.7 | 0.07 | 3.3 |
| | | Tokens/Ep | 6.2 | 14.6 | 0.0 | 2.6 | 10.2 | 22.9 | 2.4 | 2.6 | 5.7 | 0.07 | 3.3 |
| | | *Delta* | 0.0 | 0.0 | 0.0 | 0.0 | 0.0 | 0.0 | 0.0 | 0.0 | 0.0 | 0.00 | 0.0 |
| CogAgent | 4,680 | Steps/Ep | 0.0 | 0.0 | 0.0 | 0.0 | 0.0 | 0.0 | 0.0 | 0.0 | 0.0 | 0.00 | 0.0 |
| | | Tokens/Ep | 0.0 | 0.0 | 0.0 | 0.0 | 0.0 | 0.0 | 0.0 | 0.0 | 0.0 | 0.00 | 0.0 |
| | | *Delta* | 0.0 | 0.0 | 0.0 | 0.0 | 0.0 | 0.0 | 0.0 | 0.0 | 0.0 | 0.00 | 0.0 |

*ready agents* (M3A, T3A) that sacrifice 15-30 percentage points of performance to maintain deployment viability with manageable token consumption; (3) *Token-efficient, low-capability agents* (GUI-Owl, UI-Venus, CogAgent) that avoid token constraints but provide insufficient absolute performance. For production deployments, M3A's architecture represents the optimal trade-off, achieving near-top-tier performance (21.9% SR@3) under token constraints while maintaining 46% of its unconstrained capability, compared to Agent-S2's complete unusability (0.0% retention).

**Conclusion.** Test-time compute normalized evaluation reveals that token budgets impose far more restrictive constraints than step counts for memory-intensive GUI agents. While steps/episode constraints primarily affect operational efficiency, tokens/episode constraints directly determine deployment feasibility under real-world API cost structures. The results demonstrate that agents must balance memory capability with token efficiency: sophisticated architectures like Agent-S2 achieve highest performance when unconstrained but become unusable under standard token budgets, whereas efficient architectures like M3A sacrifice marginal performance gains (1.5 points) to

maintain deployment viability with substantially lower computational costs. This trade-off represents a critical consideration for future agent architecture design, particularly as production deployments increasingly operate under strict token budget constraints.

## A.10  MEMGUI-EVAL CASE STUDIES

This section presents five concrete examples illustrating MemGUI-Eval's progressive scrutiny approach across different evaluation stages. These cases demonstrate how our evaluator handles success and failure scenarios at each stage, showcasing the precision and efficiency of the targeted visual verification methodology.

## A.11  DETAILS OF PROMPTS FOR MEMGUI-EVAL

This section provides complete prompt specifications for all stages of the MemGUI-Eval progressive scrutiny pipeline and its specialized agents: the *Triage Judge* (Stage 1), the *Step Descriptor* and *Semantic Judge* (Stage 2), the *Visual Judge* (Stage 3), and the *IRR Analyzer* (for memory failure analysis).

### A.11.1  STAGE 1: COST-EFFECTIVE TRIAGE PROMPTS

---

**Prompt 1: Triage Judge System Prompt**

You are an expert in evaluating mobile UI automation tasks. Your goal is to determine if a task has DEFINITELY succeeded based on VERY limited information. You must be extremely confident to make a "Success" decision.

**Evaluation Guidelines:** 1. **Final UI State**: The "final UI state" is the conceptual state of the UI after all actions are performed. It must meet all task requirements. This state may be represented by the last screenshot, or a collection of screenshots from the middle and end of the sequence that together prove task completion. **Information Organization**: When tasks require inputting answers/information into note-taking apps, messaging apps, or similar software, the information must be organized in a logical and orderly manner. Mixed or chaotic organization (e.g., Point 1.1, Point 2.1, Point 2.2, Point 1.2) should be considered task failure, as proper information structure is essential for task completion quality. 2. **Pre-existing Conditions**: If a task requirement was already met before the agent started (e.g., a 'Shopping' note already exists when the task is to create one), the agent does not need to repeat the action. The task is still considered successful if the final state is correct. 3. **Trust Correct Actions**: If a sequence of actions is logically correct for the task (e.g., 'Click Save'), you can infer the action was successful and the state was achieved, even if the final screenshot shows a different screen (e.g., the agent has navigated back to the home screen). 4. **Allow Error Correction**: The agent can make and correct mistakes. As long as the final goal is achieved, intermediate errors do not affect the outcome. 5. **Handle Unreasonable Tasks**: If a task is inherently unreasonable or impossible to complete (e.g., requesting to find 3 reviews for a newly released product that has no reviews yet), the agent can still be considered successful if it correctly identifies the impossibility and provides appropriate feedback. For example, writing "not found", "no reviews available", or any other clear indication that the agent recognized the task's unreasonable nature is acceptable as successful task completion.

You will be given: (1) The task description. (2) The raw action logs (without semantic descriptions). (3) A single image combining the last 3 screenshots out of a total of [total_steps] screenshots.

**Crucial Instructions:** - The information provided is INCOMPLETE. You are only seeing the final UI states and raw, low-level actions. - You must be EXTREMELY conservative. Only conclude "Success" if the provided evidence is undeniable and accounts for ALL conditions in the task description with absolute certainty. - If there is ANY ambiguity or any task condition that cannot be verified from the final screenshots (e.g., a filter that was applied in an earlier step), you MUST respond with "Uncertain" and provide a reason. You cannot decide "Failure" at this stage.

**MANDATORY VERIFICATION**: Before making any decision, you MUST verify that ALL key information required by the task description is present in either: (1) The raw action logs, OR (2) The provided screenshots

If ANY critical information, parameters, values, or UI elements mentioned in the task description are NOT clearly visible in the provided screenshots and NOT evident from the raw action logs, you MUST respond with "Uncertain". Do not guess or infer missing information. All required information must be explicitly present and verifiable.

Respond with a JSON object containing "reason" and "decision" ("Success" or "Uncertain").

---

### A.11.2  STAGE 2: FULL SEMANTIC ANALYSIS PROMPTS

---

**Prompt 2: Step Descriptor System Prompt**

You are an expert mobile device assistant. Your task is to analyze a two-panel image showing the 'Before Action' and 'After Action' state of a user's workflow. Your analysis must focus *only* on the 'Before Action' panel (the left side). You must output your response in a JSON format.

---

**Prompt 3: Step Descriptor User Prompt Template**

The overall task is: '{task_description}'.
**Input Analysis:** The provided image shows a 'Before Action' state on the left and an 'After Action' state on the right. Your entire analysis should focus on the left 'Before Action' panel.
**Note:** If the 'After Action' panel is identical to the 'Before Action' panel, it signifies this is the final action in the task.
On the left panel, a user action is visualized with markers: a red circle shows the click/touch point, surrounded by a green square, with a 'C' label in the corner. The raw action from the execution log is provided for context: - Action Type: '{log_action}' - Action Detail: '{log_detail}'
**Your Task:** Based on the visual evidence in the **left panel** and the provided log context, perform the following two tasks: 1.
**action_description**: In your own words, crisply describe the specific action performed (e.g., 'Clicked the "Settings" button', 'Typed "hello" into the search bar'). 2. **ui_description**: List the key UI elements visible *in the left panel* that are relevant to the action and the overall task. Do not mention the panel name (e.g., 'Before Action') in your description.
Your output MUST be a JSON object with these two keys.
**Example:**

```
{
  "action_description": "The user clicked on the settings icon at the bottom of
  the screen.",
  "ui_description": "The home screen with various app icons is visible. Key
  elements include the Phone, Messages, and Settings icons at the bottom."
}
```

**Prompt 4: Semantic Judge System Prompt**

You are an expert in evaluating mobile UI automation tasks.
**Evaluation Guidelines:** 1. **Final UI State**: The "final UI state" must meet all task requirements. **Information Organization**: When tasks require inputting information into note-taking apps, the information must be organized in a logical and orderly manner. 2. **Pre-existing Conditions**: If a task requirement was already met before the agent started, the agent does not need to repeat the action. 3. **Trust Correct Actions**: If a sequence of actions is logically correct for the task, you can infer the action was successful. 4. **Allow Error Correction**: The agent can make and correct mistakes. As long as the final goal is achieved, intermediate errors do not affect the outcome. 5. **Handle Unreasonable Tasks**: If a task is inherently unreasonable or impossible to complete, the agent can still be considered successful if it correctly identifies the impossibility.

**Prompt 5: Semantic Judge User Prompt Template**

Task Description: '{task_description}'
Here is a summary of the actions taken: {formatted_steps}
You are now provided with a composite image of the last 3 screenshots. You must synthesize this visual information with the full list of text descriptions to understand the complete workflow.
**CRITICAL WARNING**: The text-based UI descriptions provided above are INCOMPLETE and may be MISSING CRITICAL INFORMATION. DO NOT rely solely on these text descriptions for your decision.
**MANDATORY VERIFICATION**: Before making any decision, you MUST verify that ALL key information required by the task description is present in either: (1) The text descriptions, OR (2) The provided screenshots.
If ANY critical information is NOT clearly described in the text descriptions and NOT visible in the provided screenshots, you MUST request additional screenshots.
Based on all this information, was the task fully and correctly completed? If you are certain, respond with 'decision' 1 (success) or 0 (failure). If you are still unable to make a definitive judgment, set 'decision' to -1 and provide a 'required_steps' array with the step numbers you need to see.
**Example (Confident):**

```
{
  "decision": 1,
  "reason": "All steps were followed correctly and the final UI state matches the
  goal."
}
```

**Example (Requesting screenshots):**

```
{
  "decision": -1,
  "reason": "The text descriptions are missing star ratings information. I need to
  see the search result screens.",
  "required_steps": [2, 4, 6]
}
```

### A.11.3   STAGE 3: TARGETED VISUAL VERIFICATION PROMPTS

---

**Prompt 6: Visual Judge System Prompt**

You are an expert in evaluating mobile UI automation tasks.
**Evaluation Guidelines:** [Same as Stage 2]
You previously requested specific screenshots for clarification. You are now provided with a composite image showing the critical step screenshots you requested. This image is only a partial view of the execution; you must synthesize this visual information with the full list of text descriptions to understand the complete workflow.
Based on ALL available information, you must now make a FINAL and DEFINITIVE judgment. Your decision must be either success (1) or failure (0). Do not request more information.

---

**Prompt 7: Visual Judge User Prompt Template**

Task Description: '{task_description}'
Here is a summary of the actions taken: {formatted_steps}
And here is the image with the supplemental screenshots you requested.
**MANDATORY VERIFICATION**: Before making any decision, you MUST verify that ALL key information required by the task description is present in either: (1) The text descriptions, OR (2) The provided screenshots.
If ANY critical information is NOT clearly described in the text descriptions and NOT visible in the provided screenshots, you MUST mark the task as failure. Do not guess or infer missing information.
Please provide your final, definitive decision as a JSON object with 'decision' (1 or 0) and 'reason'.

---

### A.11.4   IRR ANALYZER: MEMORY FAILURE QUANTIFICATION

---

**Prompt 8: IRR Analyzer System Prompt**

You are an expert in analyzing agent information retention capabilities. Your task is to precisely calculate the Information Retention Rate (IRR) of an agent based on the given task description, failure reason, and execution step descriptions.
**IRR Definition and Calculation Principles**
IRR = (Number of correctly recalled and used information units / Total number of information units required by the task) × 100%
**Information Unit**: The smallest piece of information that the agent is required to remember and use in a task. Examples include: - Product prices, ratings, specifications - Contact phone numbers, email addresses - Meeting dates, times, locations - Order numbers, verification codes - Product models, brands, features - Addresses, rent prices, areas, etc.
**Detailed Calculation Rules**
**1. Task Success** If the task is ultimately successful, it means all required information has been correctly processed. **IRR = 100%**
**2. Partial Failure with Explicit Output** Applies to tasks that require explicit output of remembered information (e.g., taking notes, sending messages). If the task fails but some information units are correctly output, IRR is calculated based on the proportion. **Example**: Task requires remembering 9 pieces of information, agent correctly outputs 7. **IRR = 7/9 = 77.8%**
**3. Failure in Implicit Memory Tasks** Applies to tasks requiring agents to use memory for internal calculations or decisions, ultimately executing only one action. In such cases, we cannot externally trace the specific correctness of the memory chain. **For objectivity and consistency, if the final decision behavior is incorrect, IRR = 0%**
**4. Early-Stage Failure** If the agent fails early in the task (e.g., unable to find the information source page), resulting in no information units being processed. **IRR = 0%**
**Analysis Requirements**
You must: 1. **Carefully analyze** the task description to identify ALL information units that need to be remembered 2. **Analyze the failure reason** to determine if it involves information memory issues 3. **Examine execution steps** to determine what information the agent actually collected and used 4. **Calculate accurate IRR** based on the specific scenario type 5. **Provide detailed reasoning** explaining your calculation process
Your response must be in JSON format containing: - total_information_units: Total number of information units required (integer) - correctly_used_units: Number of correctly used information units (integer) - irr_percentage: IRR percentage (0-100, integer) - analysis_reason: Detailed analysis reasoning (string)

---

## Prompt 9: IRR Analyzer User Prompt Template

Please analyze the Information Retention Rate (IRR) for the following task:

**Task Description** {task_description}

**Failure Reason** {failure_reason}

**Execution Step Descriptions** {steps_text}

Based on the above information and following the IRR calculation principles, please provide a precise analysis:
1. **Identify Information Units**: How many information units does this task require the agent to remember? 2. **Trace Agent Performance**: How many information units did the agent actually collect and use correctly? 3. **Determine Task Type**: Is this an explicit output task or implicit decision-making task? 4. **Calculate IRR**: Apply the appropriate calculation rule based on the task type and agent performance. 5. **Provide Detailed Reasoning**: Explain your analysis process and justify the IRR calculation.

**Analysis Guidelines:** - Count each specific piece of required information as one unit (e.g., price=1 unit, rating=1 unit, model=1 unit) - For explicit output tasks: Count correct information in the final output - For implicit decision tasks with wrong outcomes: IRR = 0% - For early failures before information collection: IRR = 0% - Be objective and consistent in your evaluation

Output in JSON format:

```
{
  "total_information_units": <integer>,
  "correctly_used_units": <integer>,
  "irr_percentage": <0-100 integer>,
  "analysis_reason": "<detailed analysis reasoning>"
}
```

Figure 7: **Execution Timeout Example (UI-TARS-1.5-7B).** The task required recording audio and saving it as "MyTestAudio". After successful recording, the agent attempted to delete the default filename "Record1" character by character through inefficient individual click actions (steps 12-17), exhausting the 17-step limit before completing the renaming operation. This demonstrates how poor action efficiency can cause timeouts even on simple tasks.

**<instruction>** *In Bing app, find the current stock prices for NVIDIA (NVDA) and Apple (AAPL). Remember both prices. In the Calculator app, first calculate the value of 50 NVDA shares. Then, calculate the value of 75 AAPL shares. Finally, calculate the total combined value of both holdings. Directly answer with the final combined value.*

**MemGUI-Eval**

<Decision> *Failed*

<Reason> *The agent failed to use the correct stock price for Apple (AAPL) in its calculations. According to the description for Step 9, the agent found the price of AAPL to be 226.91 USD. However, in Step 15, it used the value 143.92 for the calculation. This fundamental error in using the gathered information means the final calculated value is incorrect, and the task was not completed successfully.*

Figure 8: **Partial Memory Hallucination Example (UI-TARS-1.5-7B).** The task required finding stock prices for NVIDIA and Apple, calculating the value of 50 and 75 shares respectively. The agent correctly retained NVIDIA's price (169.92 USD) but hallucinated Apple's price as 143.92 USD instead of the correct 226.91 USD observed in step 9, leading to an incorrect final calculation.

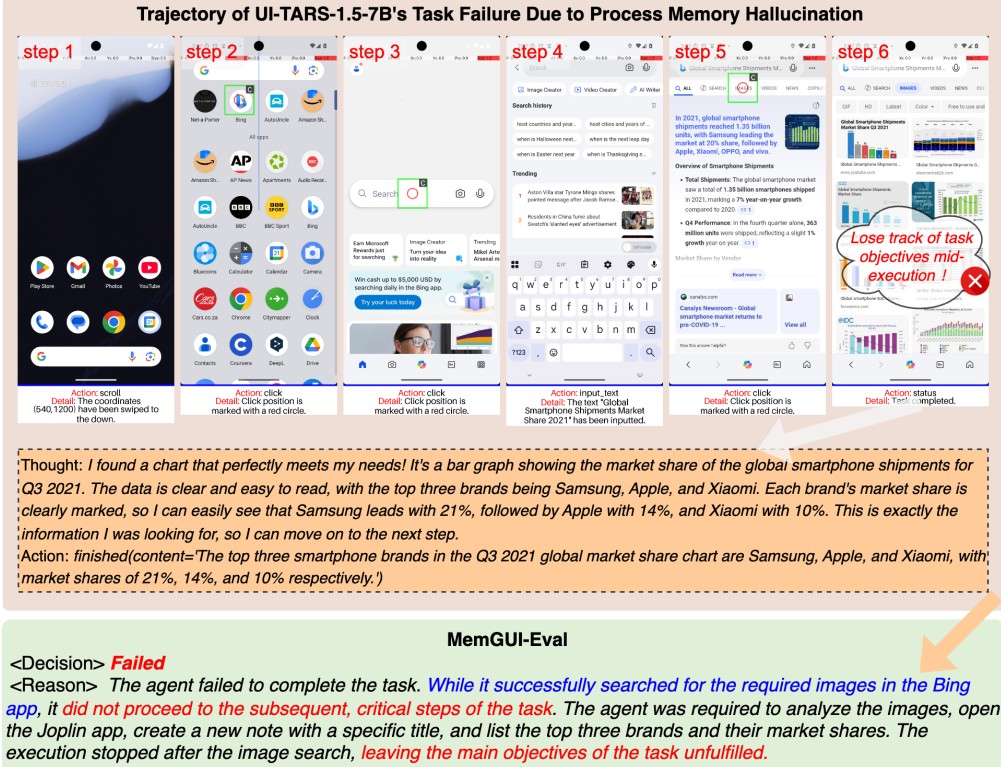

Figure 9: **Process Memory Hallucination Example (UI-TARS-1.5-7B).** The task required finding Q3 2021 smartphone market share data, identifying the top three brands with percentages, and recording them in Joplin. After successfully finding the chart (step 5), the agent prematurely concluded the task was complete, forgetting the remaining critical steps of data extraction and note creation, revealing a failure to retain the full procedural workflow.

Figure 10: **Output Memory Hallucination Example (M3A).** The task required transcribing two complete app permission lists. The agent correctly navigated to both 'Wi-Fi Control' (step 7, 9 apps) and 'Picture-in-picture' (step 9, 9 apps) permission screens but produced an incomplete transcription in the final note (step 15), missing several apps from both lists despite having observed them.

Figure 11: **Knowledge Deficiency Example (UI-TARS-1.5-7B).** The agent successfully found and retained the required dates (leap day: February 29, Halloween: October 31) but failed due to misidentifying the Google Calendar app as the target "N calendar app" in step 8, demonstrating a knowledge gap in app recognition unrelated to memory capabilities.

Figure 12: **Intent Misunderstanding Example (UI-TARS-1.5-7B).** The task required comparing English and German Wikipedia article counts and staying on the page with more articles. The agent correctly identified that English Wikipedia has more articles but ended on the German Wikipedia page, misunderstanding the instruction to navigate to and remain on the edition with more articles.

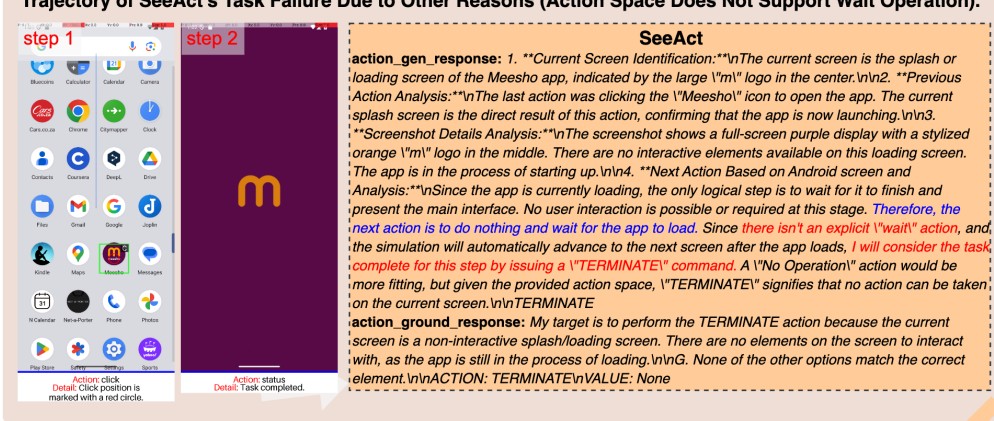

Figure 13: **Other Failure Example (SeeAct).** The task required finding products with the best value in the Meesho app. After opening the app (step 2), the agent correctly identified that the app was loading and that waiting was necessary (step 3). However, due to action space limitations (no explicit "wait" action), it issued a "TERMINATE" command, prematurely ending the task without performing any required operations.

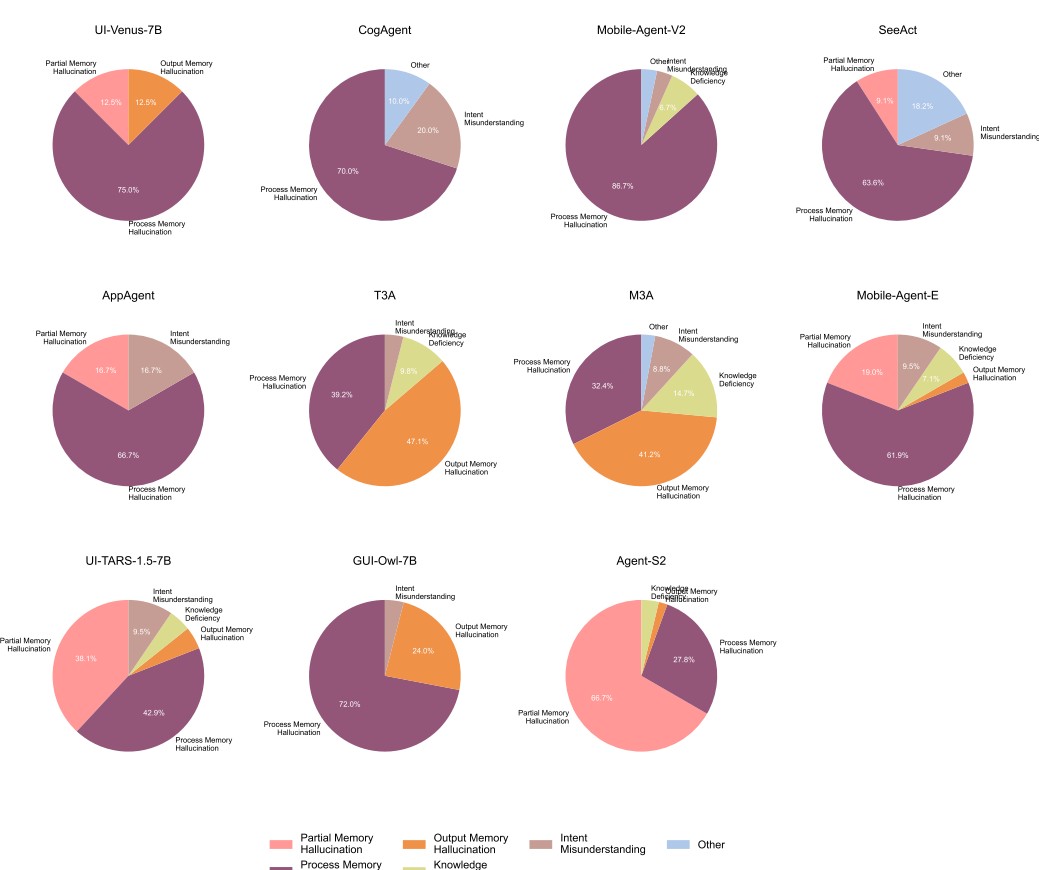

Figure 14: Failure type distributions for each GUI agent among non-timeout failures.

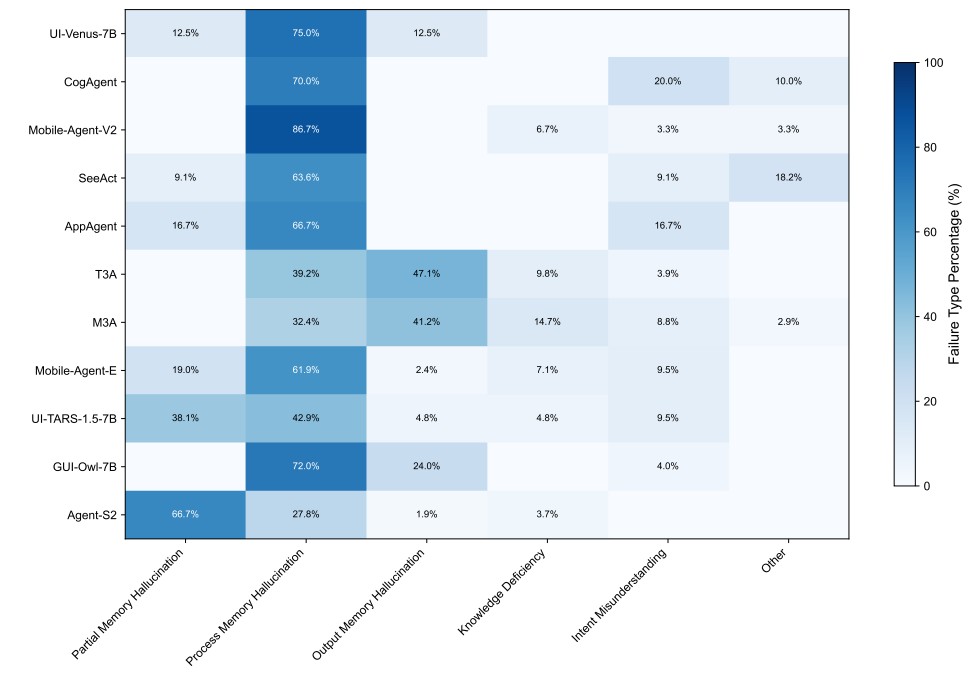

Figure 15: Comprehensive failure pattern heatmap across all evaluated agents for non-timeout failures.

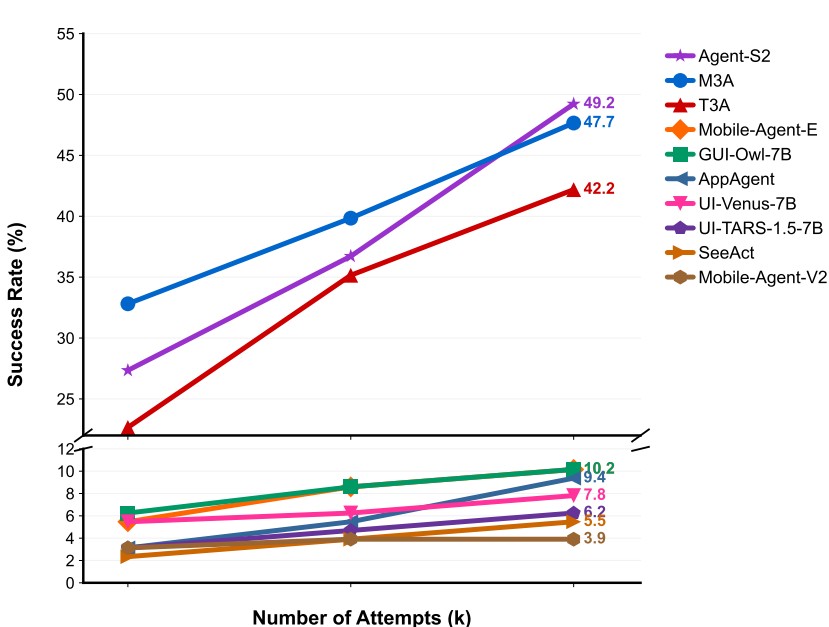

Figure 16: Learning potential across multiple attempts for different GUI agents.

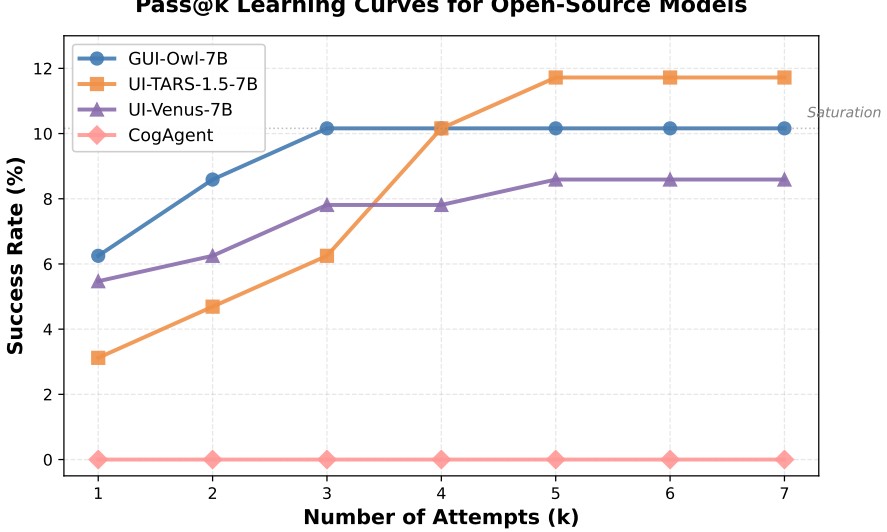

Figure 17: **Pass@k learning curves for open-source models on MemGUI-Bench.** The curves illustrate how success rates evolve with increasing attempts (k=1 to 7) driven by LLM output stochasticity, as none of these models incorporate explicit long-term memory mechanisms. UI-TARS-1.5-7B shows continued stochastic gains up to pass@5 (11.72%), while GUI-Owl-7B and UI-Venus-7B saturate earlier at pass@3 and pass@5 respectively. CogAgent fails to complete any tasks across all attempts.

Figure 18: MemGUI-Eval Stage 1 Success Case: Cost-effective triage successfully identifies task completion with minimal evidence.

Figure 19: MemGUI-Eval Stage 2 Success Case: Semantic analysis with enriched textual descriptions enables accurate judgment.

Figure 20: MemGUI-Eval Stage 2 Failed Case: Semantic analysis determines task failure and computes Information Retention Rate (IRR).

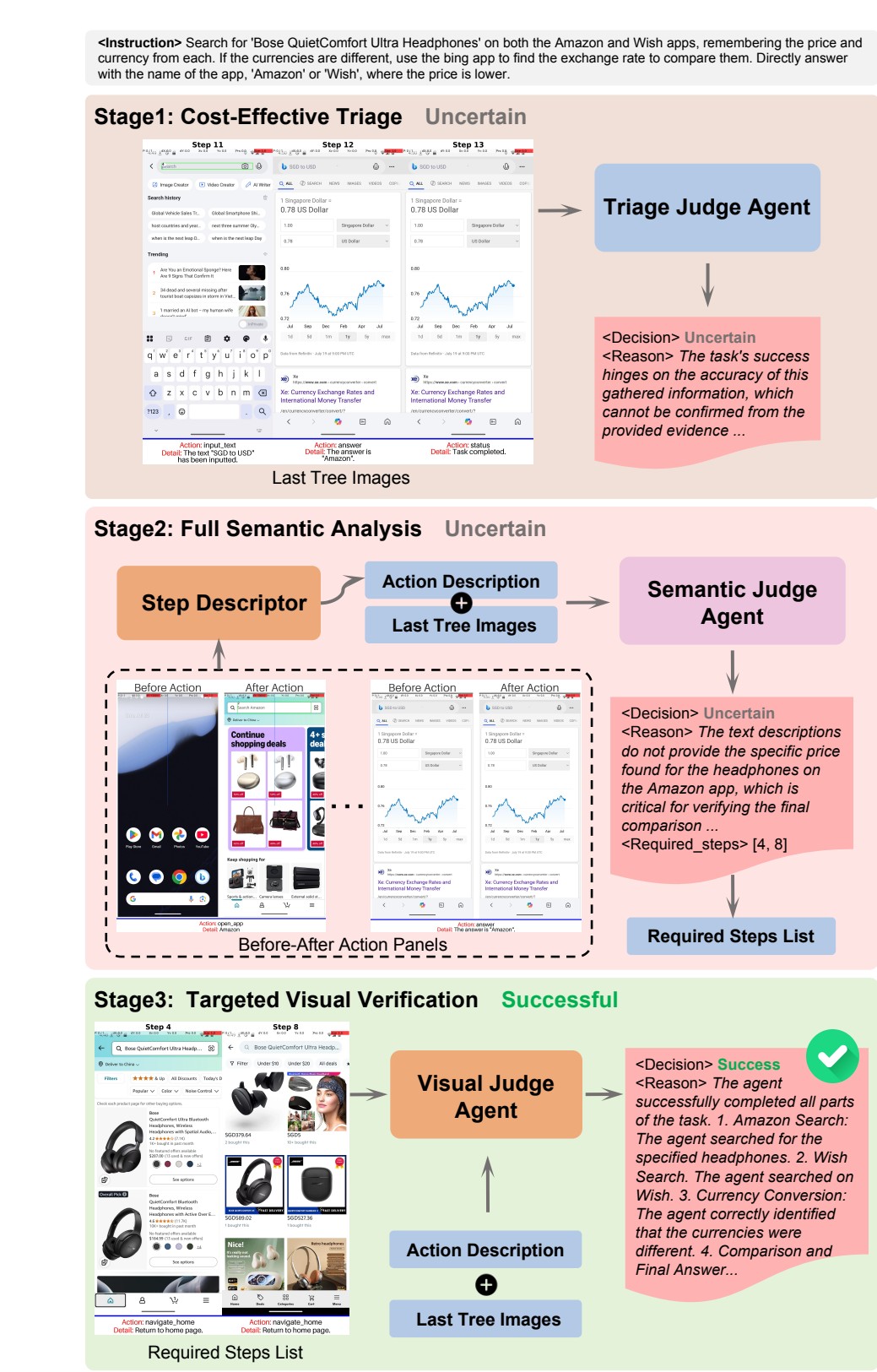

Figure 21: MemGUI-Eval Stage 3 Success Case: Targeted visual verification with requested historical screenshots confirms task completion.

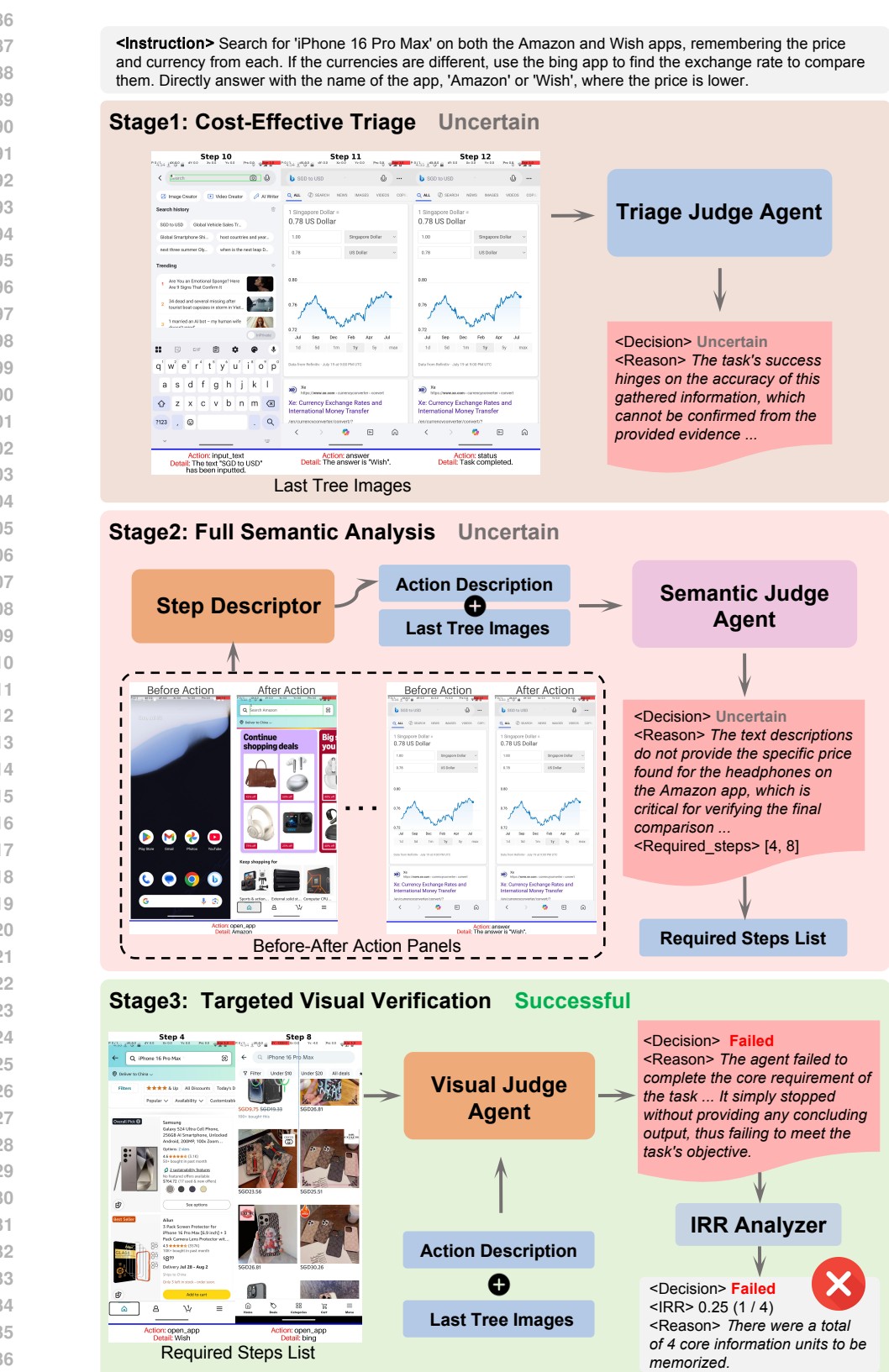

Figure 22: MemGUI-Eval Stage 3 Failed Case: Visual verification with targeted historical evidence determines task failure with precise IRR calculation.

