# OpenReview forum: "MemGUI-Bench: Benchmarking Memory of Mobile GUI Agents in Dynamic Environments"
_ICLR.cc/2026/Conference — ICLR 2026 Conference Withdrawn Submission_

### Official Review · Reviewer_DFxX · 2025-10-26

**Soundness:** 3
**Presentation:** 2
**Contribution:** 2
**Rating:** 2
**Confidence:** 4

**Summary:**

This paper proposed a benchmark to evaluate short- and long-term memory usage among mobile GUI agents. This paper aggregates tasks from varied applications and compares 11 agent frameworks, revealing their relative performance.

**Strengths:**

1. **Large-Scale Effort.**
> This paper analyzes 11 agents and aggregates tens of applications, covering major works in the domain of mobile manipulation.

2. **Comprehensive agent support and evaluation protocol.**
> As introduced in section sec 3.2, this work supports a unified pipeline to ensure robust agent evaluation. Also, the metrics proposed in sec 4 enable memory-targeting evaluations, with human-annotated references.

**Weaknesses:**

1. **Limited practical utility among agent developments.**
> First, memory seems like a useful component designed in *some* works, yet not a universal feature that needs to be incorporated by agents. Therefore, evaluating memory is an interval, intermediate self-check for some agents, rather than a universal correctness metric such as task success rates.

> Second, this work structure agent memory possibly inspired by how human memory works (this point is less justified as well), yet this may not generalize across all agent designs or be useful for the most performant agent at all. For example, this work separates short-term and long-term memory, yet agents may become effective in long context modeling thus no need to split memory by time. It may be more reasonable to frame this work as an empirical analysis on *some* agent frameworks that share similar memory structure.

> Lastly, it is unclear whether the established memory taxonomy generally applies to domains (web browsing, computer use) and tasks (personal assistant, work) beyond mobile manipulation.

2. **Inaccurate definition of memory.**
> From the description in section 2, it may be more accurate to frame the two categories as "in-session" and "cross-session" memory, as "short-term" and "long-term" are somewhat vague therefore cause confusion.

3. **Uncertain quality of evaluation examples.**
> It is unclear (based on the description in section 3.1) how the applications are selected, how the examples are created (synthesized by LM, manually annotated by human, recorded from real-human activities, etc.?), and what principles are integrated into the examples during this process.

4. **Shallow analysis findings.**
> The findings in section 5 lack practical implications, beyond that agent A is better in short/long-term memory. Deeper analysis, especially supported by clearer, more fine-grained memory aspects, such as procedural workflow/fact retention, cross-app retention, should be much more informative for future agent development works. This limitation in analysis may be somewhat incurred by the lack of clarity in task design (weakness pt3).

5. **Presentation needs to be improved.**
> There are multiple places where the tables and figures are not placed properly, e.g., Table 1. Meanwhile, improving the writing to elaborate the motivation and detailed procedures clearer, could be helpful for readers to understand this work.

**Questions:**

N/A

---

> ### Author Response · Authors · 2025-11-24
> **Response to Reviewer DFxX (1/4)**
>
> We thank Reviewer DFxX for recognizing our **"large-scale effort"** analyzing 11 agents across tens of applications covering major works in mobile manipulation, and the **"comprehensive agent support and evaluation protocol"** with a unified pipeline (Section 3.2) and memory-targeting metrics (Section 4) backed by human-annotated references. We appreciate the detailed feedback and address each concern below.
>
> ---
>
> ## Response to W1: Limited practical utility among agent developments
>
> Thank you for this concern. We conducted **systematic memory ablation experiments (`Appendix A.9.4.`, Table 14)** on 4 agents across different architectures using MemGUI-Bench-40 to empirically demonstrate that memory is a universal necessity, not an optional feature. Results show: (1) **removing short-term memory renders mobile GUI agents essentially unusable** (M3A: -30.0% SR@1 → 2.5%, -35.1% IRR → 0%; Agent-S2: -22.5% SR@1 → 5.0%, -33.3% IRR → 0%), and (2) **enhancing memory provides substantial gains** (M3A + Multi-turn: +20.0% SR@1, +18.4% IRR), establishing memory as a core determinant of agent capability.
>
> ### Ablation Results (New: `Appendix A.9.4.`, `Table 14`)
>
> **Experimental Design:**
>
> - **M3A**: Baseline → + Multi-turn context → - Memory Agent
> - **Agent-S2**: Baseline (STM+LTM) → - LTM only → - Both STM & LTM
> - **GUI-Owl-7B**: Baseline (Action-Thought) → - Action-Thought
> - **UI-TARS-1.5-7B**: Baseline (Multi-turn+A-T) → - Both removed
>
> **Key Results:**
>
> | Agent                | Configuration    | SR@1 (Δ)       | SR@3 (Δ)       | IRR (Δ)        | MTPR (Δ)       |
> | :------------------- | :--------------- | :-------------- | :-------------- | :-------------- | :-------------- |
> | **M3A**        | Baseline         | **32.5%** | **47.5%** | **35.1%** | **0.321** |
> |                      | + Multi-turn     | 52.5% (+20.0)   | 70.0% (+22.5)   | 53.5% (+18.4)   | 0.457 (+0.14)   |
> |                      | - Memory Agent   | 2.5% (−30.0)   | 5.0% (−42.5)   | 0.0% (−35.1)   | 0.00 (−0.32)   |
> | **Agent-S2**   | Baseline         | **27.5%** | **45.0%** | **33.3%** | **0.250** |
> |                      | - LTM only       | 17.5% (−10.0)  | 25.0% (−20.0)  | 21.3% (−12.0)  | 0.19 (−0.06)   |
> |                      | - STM & LTM      | 5.0% (−22.5)   | 10.0% (−35.0)  | 0.0% (−33.3)   | 0.00 (−0.25)   |
> | **GUI-Owl-7B** | Baseline         | **7.5%**  | **12.5%** | **4.6%**  | **0.000** |
> |                      | - Action-Thought | 7.5% (0.0)      | 10.0% (−2.5)   | 0.0% (−4.6)    | 0.00 (0.0)      |
> | **UI-TARS**    | Baseline         | **5.0%**  | **5.0%**  | **2.3%**  | **0.000** |
> |                      | - Multi-turn+A-T | 2.5% (−2.5)    | 2.5% (−2.5)    | 0.0% (−2.3)    | 0.00 (0.0)      |
>
> **Conclusions:**
>
> 1. **Short-term memory is mandatory for mobile GUI agents to function**: Removing STM renders agents essentially unusable (M3A: 32.5% → 2.5% SR@1, 35.1% → 0% IRR; Agent-S2: 27.5% → 5.0% SR@1, 33.3% → 0% IRR), with IRR collapsing to zero across all agents, confirming that STM is indispensable regardless of implementation paradigm.
> 2. **Long-term memory is beneficial**: Agent-S2 with LTM achieves +20.0% SR@3 gain versus baseline without LTM, demonstrating LTM's critical value for cross-session learning and establishing it as a key direction for future research.

---

> ### Author Response · Authors · 2025-11-24
> **Response to Reviewer DFxX (2/4)**
>
> ## Response to W1 (continued): Long Context vs. Memory Structure
>
> Thank you for this insightful question. We conducted token consumption analysis comparing **Current Memory Structures** (summarization/action-history) versus **Full Context** (retaining all historical infomation) for GUI-Owl and M3A to empirically demonstrate that **memory structuring remains essential** even with long-context VLMs, due to: (1) mainstream 128k/200k models saturate in 27-40 steps under full context, insufficient for single complex tasks, (2) even 2M context is finite for lifelong learning (saturates after a few days), and (3) processing 2M tokens per action incurs prohibitive latency/cost.
>
> ### Context Saturation Analysis
>
> | Agent Configuration                         | GPT-4o / Qwen2.5 (128k) | Claude 3.7 Sonnet (200k) | Gemini 2.5 Pro (2M) |
> | :------------------------------------------ | :---------------------- | :----------------------- | :------------------ |
> | **GUI-Owl** (Current: Action-Thought) | >800 steps              | >1,000 steps             | >10k steps          |
> | **GUI-Owl** (Ablation: Full Context)  | ~40 steps               | ~65 steps                | ~660 steps          |
> | **M3A** (Current: Summarization)      | ~400 steps              | ~600 steps               | ~6,000 steps        |
> | **M3A** (Ablation: Full Context)      | ~27 steps               | ~43 steps                | ~440 steps          |
>
> *"Saturation" = single-step inference context exceeds model's window limit.*
>
> ### Key Findings
>
> 1. **Mainstream models require memory structuring**: 128k/200k VLMs saturate in 27-40 steps under full context, insufficient for complex MemGUI-Bench tasks (some exceed 50 steps). Memory structuring extends this to 400-1,000+ steps, making current deployment viable.
> 2. **2M context is finite for lifelong learning**: Full context mode saturates after 440-660 steps (~few days of operation). Lifelong agents accumulating months of experience (tens of thousands of steps) require retrieval-based LTM for unbounded memory capacity.
> 3. **Efficiency matters**: Processing 2M tokens per action incurs prohibitive latency/cost. Structured memory maintains lightweight inference context (STM) with efficient retrieval access to vast knowledge (LTM).
>
> **Conclusion**: Long-context VLMs extend the buffer but do not eliminate the need for explicit memory structuring in production mobile GUI agents requiring efficiency and unbounded lifelong learning capabilities.
>
> ---
>
> ## Response to W2: Terminology clarification for "short-term" vs. "long-term" memory
>
> Thank you for this suggestion. In the revised draft, we first note that prior LLM agent works on memory (LongMemEval [1], LoCoMo [2], MemoryBank [3]) consistently adopt the "short-term vs. long-term" distinction, and we follow this established convention. To address potential ambiguity, we have **explicitly clarified the correspondence in revised `Section 2.`**: *short-term memory* = **in-session** contextual buffers (interaction history during task execution), while *long-term memory* = **cross-session** experiential memory (reusable skills across attempts).
>
> **References:**
>
> [1] Wu, Di, et al. "Longmemeval: Benchmarking chat assistants on long-term interactive memory." *ICLR*, 2025.
>
> [2] Maharana, Adyasha, et al. "Evaluating Very Long-Term Conversational Memory of LLM Agents." *ACL*, 2024.
>
> [3] Zhong, Wanjun, et al. "Memorybank: Enhancing large language models with long-term memory." *AAAI*, 2024.

---

> ### Author Response · Authors · 2025-11-24
> **Response to Reviewer DFxX (3/4)**
>
> ## Response to W3: Uncertain quality of evaluation examples
>
> Thank you for this question. We have enhanced `Appendix A.4.` to comprehensively document the curation pipeline:
>
> **Application Selection** (already in original `Section 3.1.`, `Appendix A.4.`): We selected high-frequency, login-free apps compatible with emulators to ensure reproducibility. Tasks prioritize memory-intensive scenarios, cross-app transfers, and mirror-pair structures for long-term learning evaluation.
>
> **Task Creation Process** (enhanced in revised `Appendix A.4.2.`): All 128 tasks and golden steps were **manually annotated by human experts** using a rigorous three-person cross-validation process where independent annotators verified task executability and golden path correctness.
>
> ---
>
> ## Response to W4: Shallow analysis findings
>
> Thank you for this constructive feedback. We have added fine-grained analysis revealing: (1) **7 distinct failure mode patterns** with detailed trajectory analysis (`Figures 7-13`) and comprehensive failure distributions for all 11 agents, (2) **consistent 3-step cross-app degradation** – 1-app → 2-app drops 20-30%, 2-app → 4-app drops another 30-50% across 11 agents, with IRR remaining >40% confirming the bottleneck is retention not perception, and (3) **5 architectural innovations** addressing multi-granularity buffers, hierarchical decomposition, long-context utilization, cross-session learning, and hybrid designs. Details below:
>
> **1. Fine-Grained Failure Mode Analysis (Enhanced: `Appendix A.8.1.`)**
>
> We added detailed trajectory analysis (`Figures 7-13`) illustrating **procedural workflow** (Process Memory Hallucination) and **fact retention** (Partial/Output Memory Hallucination) failures with concrete examples for each failure mode. Additional agent-specific failure distributions and cross-agent pattern analysis are provided in `Appendix A.8.2.`-`A.8.3.`.
>
> **2. Cross-Application Retention Analysis (New: `Appendix A.9.3.`, `Table 13`)**
>
> We added comprehensive performance breakdown by cross-app complexity (28 single-app, 56 two-app, 34 three-app, 10 four-app tasks), reporting both SR and IRR for pass@1:
>
> - **Consistent 3-step degradation pattern across 11 agents**: 1-app → 2-app drops 20-30%, 2-app → 4-app drops another 30-50% (exponential decay); top agents achieve 42.9-50.0% on single-app tasks but fall to 0.0-30.0% on four-app tasks. Since IRR remains >40% during this decline, the bottleneck is cross-app retention rather than perception errors
> - **Distinct IRR patterns**: Agent-S2 maintains high IRR (33.3-51.7%) despite SR drops, indicating strong information retention even during partial execution; M3A peaks at 43.8% IRR for 2-app tasks, suggesting optimal complexity for its memory architecture
> - **Model-based architectural constraints**: Agent-as-a-Model approaches achieve 0.0-2.9% on multi-app tasks even with learning (pass@3), confirming fundamental limitations for cross-app memory state maintenance
>
> **3. Explicit Design Implications (New: `Appendix A.8.4.`)**
>
> We synthesized five concrete architectural recommendations linking observed failure patterns to solutions:
>
> 1. **Multi-granularity memory buffers**: Structured memory with separate slots for different information types (numerical facts, textual descriptions, UI states)
> 2. **Hierarchical task decomposition**: Persistent goal tracking across application boundaries to maintain task objectives during multi-app workflows
> 3. **Strategic long-context utilization**: Importance-weighted context management beyond naive conversation history concatenation
> 4. **Explicit long-term memory mechanisms**: Dedicated cross-session learning systems for failure analysis and strategy refinement
> 5. **Hybrid architectures**: Combining framework-level memory management with efficient end-to-end models to balance capability and computational cost
>
> These recommendations emphasize that advancing GUI agent memory requires architectural innovations beyond scaling model parameters or context windows.

---

> ### Author Response · Authors · 2025-11-24
> **Response to Reviewer DFxX (4/4)**
>
> ## Response to W5: Presentation and Clarity
>
> Thank you for these suggestions.
>
> **Figure/Table Placement**: We have corrected all placement issues (including `Table 1`) with consistent formatting throughout the manuscript.
>
> **Motivation and Procedure Clarity**: Regarding the concern about motivation and procedures, we respectfully note that comprehensive details are provided throughout the original submission:
>
> **Motivation (`Section 1.` and `Abstract`):** We articulate three core motivations:
>
> 1. Existing GUI benchmarks inadequately assess memory capabilities critical for complex real-world tasks
> 2. Lack of systematic evaluation frameworks for both in-session and cross-session memory
> 3. Need for reproducible, scalable evaluation infrastructure for mobile GUI agents
>
> **Detailed Procedures:** Comprehensive technical details are provided throughout:
>
> - `Section 3.1.` + `Appendix A.4.`: Task suite design methodology and application selection criteria
> - `Section 3.2.` + `Appendix A.5.`: MemGUI-Eval pipeline with Progressive Scrutiny evaluation stages
> - `Section 4.` + `Appendix A.1.`-`A.3.`: Complete agent implementation details and memory architecture specifications
>
> If there are specific sections where further clarification would be helpful, we would be glad to provide additional details in the revised manuscript.

---

> ### Author Response · Authors · 2025-11-28
>
> Dear Reviewer DFxX,
>
> Thank you for your feedback. To address your concerns about memory generalization, evaluation quality, and the depth of analysis, we added memory ablation experiments, conducted fine-grained failure analyses, and provided concrete design implications.
>
> As the discussion phase is coming to a close, we would greatly appreciate your feedback on whether these additions address your concerns. Thank you for your time.
>
> Best regards,
>
> The Authors

---

### Official Review · Reviewer_LqfD · 2025-10-30

**Soundness:** 3
**Presentation:** 3
**Contribution:** 3
**Rating:** 6
**Confidence:** 3

**Summary:**

The paper proposes MemGUI-Bench, a benchmark for mobile GUI agents focused on memory. It reports: (i) a “systematic memory taxonomy,” (ii) 128 tasks across 26 apps, with ~90% designed to stress memory, (iii) MemGUI-Eval with “Progressive Scrutiny” and 8 hierarchical metrics, and (iv) broad evaluations arguing existing agents have major memory deficits (4–10× gap between memory-intensive vs “standard” tasks) and that explicit long-term memory helps. Code link is provided.

**Strengths:**

- Timely problem framing. Mobile GUI agents are rising; a purpose-built memory benchmark is valuable and under-served. The cross-temporal/cross-spatial emphasis aligns with real usage.

- Scale & coverage. 128 tasks / 26 apps is non-trivial for interactive GUI evaluation; the claimed memory-task share (~89.8%) suggests deliberate design rather than incidental memory.

- Evaluation pipeline ambition. “Progressive Scrutiny” + hierarchical metrics aim to move beyond pass/fail, which is the right direction for agent memory diagnostics.

**Weaknesses:**

- **Generalization beyond the curated suite**: The benchmark spans 26 apps, but are they category-balanced (commerce, productivity, social, finance), regionally representative, and covering UI paradigms (infinite scroll, nested modals, webviews)? Without a sampling rationale and held-out app categories, it’s unclear if results generalize or if models “learn the test.”

- **Memory vs. perception/exploration confound**: It’s unclear whether measured failures are truly memory failures versus UI perception, layout parsing, long-horizon exploration, or tool-use orchestration. Without perception-controlled variants (e.g., textual UI abstractions; identical observation streams with/without memory demand), a 4–10× gap could reflect harder vision or search rather than memory per se. The paper needs ablations isolating memory load from these factors. (They currently only assert memory difficulty; details are missing on how confounds are neutralized.)

**Questions:**

I don't have any questions.

---

> ### Author Response · Authors · 2025-11-24
> **Response to Reviewer LqfD (1/2)**
>
> We thank Reviewer LqfD for recognizing the **"timely problem framing"** of memory benchmarking for mobile GUI agents, the **"non-trivial scale (128 tasks / 26 apps)"** with deliberate memory-centric design (~89.8% memory-intensive tasks), and the **"evaluation pipeline ambition"** with Progressive Scrutiny and hierarchical metrics moving beyond pass/fail diagnostics. We appreciate the thoughtful feedback and address each concern below.
>
> ---
>
> ## Response to W1: Generalization beyond the curated suite
>
> Thank you for this question about task suite diversity and balance. MemGUI-Bench was designed to reflect real-world mobile usage patterns with comprehensive coverage across critical domains and UI paradigms.
>
> ### Category Balance
>
> Our original submission already provided visual and detailed category information through `Figure 2` (sunburst chart) and `Table 8` (per-task categories in `Appendix A.4.`). To present this data more clearly, we have added **a statistical summary table in `Appendix A.4.3.` (`Table 7`)** that quantifies the distribution:
>
> | Domain Category                  | % of Instances  | Key Sub-Categories                                |
> | :------------------------------- | :-------------- | :------------------------------------------------ |
> | **Shopping (Commerce)**    | **31.1%** | Product Search, Price Comparison, Review Analysis |
> | **Information Retrieval**  | **21.9%** | Data Extraction, Web Search, Fact Checking        |
> | **Productivity**           | **17.7%** | Note Taking, Time Management, Checklist           |
> | **Financial Management**   | **7.9%**  | Financial Calculation, Transaction Logging        |
> | **Communication (Social)** | **4.9%**  | Messaging, Data Sharing                           |
> | **Other Domains**          | **11.2%** | Content Creation, Sports, Travel, Education       |
>
> This distribution confirms balanced coverage across **Commerce, Productivity, Finance, and Social** domains, with intentional focus on areas where memory-intensive operations are most prevalent in real-world usage.
>
> ### UI Paradigm Coverage
>
> Our task suite exercises diverse interaction patterns across modern mobile applications:
>
> | UI Paradigm                    | Applications                  | Example Task Scenarios                                            |
> | :----------------------------- | :---------------------------- | :---------------------------------------------------------------- |
> | **Infinite Scroll**      | Amazon, Meesho                | Browsing product lists, identifying targets in continuous content |
> | **Nested Navigation**    | Settings, Files               | Multi-level menu traversal, spatial context maintenance           |
> | **Webviews**             | Bing, Wikipedia               | Web content interaction, embedded browser navigation              |
> | **Search & Filter**      | Amazon, Apartments.com        | Complex filter combinations, multi-criteria selection             |
> | **Multi-step Forms**     | Joplin, N Calendar, Bluecoins | Sequential input, incremental information retention               |
> | **Tabs & Bottom Sheets** | Citymapper, Settings          | Cross-tab integration, bottom drawer interactions                 |
> | **Expandable Lists**     | Files, Settings               | Collapsible hierarchical structure navigation                     |
>
> Complete details are in `Section 3.1.` and `Appendix A.4.`.

---

> ### Author Response · Authors · 2025-11-24
> **Response to Reviewer LqfD (2/2)**
>
> ## Response to W2: Memory vs. perception/exploration confound
>
> Thank you for this concern. Our original submission addressed this through **MTPR (`Section 4.1.`, `Section 5.3.`)**, isolating memory from perception/exploration by comparing performance on memory-intensive versus standard tasks. Results show 4-10× performance gaps (best agents: MTPR=0.41-0.45, most: ≤0.07). We further add **memory ablation experiments (`Appendix A.9.4.`)** in the revised manuscript, providing direct perception-controlled evidence.
>
> ### MTPR: Memory-Specific Metric (Original Submission)
>
> **MTPR (Memory-Task Proficiency Ratio)** isolates memory load by comparing performance on 115 memory-intensive tasks versus 13 standard tasks with identical perception/exploration requirements:
>
> `MTPR = SR(memory tasks) / SR(standard tasks)`
>
> **Results (`Section 5.3.`, `Table 11`):** Best-performing agents achieve Agent-S2 (MTPR=0.45) and M3A (MTPR=0.41), while most agents show MTPR ≤ 0.07, with model-based agents at 0.00-0.07. This 4-10× performance disparity between memory-intensive and standard tasks, despite identical perception and exploration capabilities, confirms memory load as the differentiating factor.
>
> ### Memory Ablation Study (New: `Appendix A.9.4.`,  `Table 14`)
>
> To provide direct "perception-controlled variants" as requested, we conducted systematic memory removal experiments on 4 agents across different architectural paradigms using MemGUI-Bench-40 (40 randomly sampled tasks maintaining original distribution):
>
> **Experimental Design:**
>
> - **M3A**: Baseline → + Multi-turn context → - Memory Agent removed
> - **Agent-S2**: Baseline (STM+LTM) → - LTM only → - Both STM & LTM
> - **GUI-Owl**: Baseline (Action-Thought) → - Action-Thought removed
> - **UI-TARS**: Baseline (Multi-turn+A-T) → - Both components removed
>
> **Results:**
>
> | Agent              | Memory Removed | SR@1 Impact       | IRR Impact        | SR@3 Impact       |
> | :----------------- | :------------- | :---------------- | :---------------- | :---------------- |
> | **M3A**      | Memory Agent   | **−30.0%** | **−35.1%** | **−42.5%** |
> | **Agent-S2** | STM & LTM      | **−22.5%** | **−33.3%** | **−35.0%** |
> | **GUI-Owl**  | Action-Thought | **0.0%**    | **−4.6%**  | **−2.5%**  |
> | **UI-TARS**  | Context & A-T  | **−2.5%**  | **−2.3%**  | **−2.5%**  |
>
> **Key Findings:**
>
> 1. **Perception-controlled evidence**: When memory components are removed while **perception and exploration remain unchanged** (identical VLMs, action spaces, UI observation streams), performance drops dramatically. This isolates memory as the causal factor.
> 2. **Memory-specific degradation**: M3A's catastrophic -30.0% SR@1 and -35.1% IRR drops confirm that failures stem from **information retention deficits, not perception failures**. The agent still sees UI correctly but cannot remember extracted information.
> 3. **Validation of MTPR findings**: These ablation results validate our MTPR-based evidence, providing direct perception-controlled confirmation that observed performance gaps result from memory deficits rather than perception/exploration confounds.

---

> > ### Comment · Reviewer_LqfD · 2025-11-27
> > **Response to the Rebuttal**
> >
> > Thanks for the detailed rebuttal! I don't have any other concerns at this time.

---

### Official Review · Reviewer_ryFo · 2025-11-04

**Soundness:** 3
**Presentation:** 2
**Contribution:** 3
**Rating:** 6
**Confidence:** 4

**Summary:**

MemGUI-Bench is a memory-centric benchmark for mobile GUI agents. It comprises (i) short-term vs. long-term memory taxonomy; (ii) a task suite of 128 tasks across 26 apps with 89.8% memory-intensive cases and 64 mirror pairs to test cross-episode learning. Further, the paper introduces MemGUI-Eval, an automated evaluation pipeline with 8 hierarchical metrics (IRR, MTPR, FRR, etc.) and pass@k support. Finally, the authors present a comprehensive evaluation of 11 agents showing large gaps between memory and non-memory tasks.

**Strengths:**

- The paper identifies a clear gap in the existing mobile-agent benchmarks. MemGUI-Bench bridges the gap with a clear focus.
- The modular feature of MemGUI-Bench eval makes it easy to integrate with existing benchmarks.
- Surprising low performance of existing state-of-the-art methods on the benchmark, substantiating its claims on the memory-gap.

**Weaknesses:**

- Paper formatting for Table 1 and Figure 4.

- Lack of qualitative examples to showcase the memory-gap in state-of-the-art model like UI-TARS. Adding such examples will demonstrate the need of the benchmark more clearly.

- L27: "First comprehensive benchmark for GUI-agent memory" is plausible, but Table 4 shows prior memory tasks exist (e.g., SPA-Bench has 40/340). I would suggest qualifying to "first comprehensive, memory-centric benchmark with pass@k and a staged LLM-as-judge evaluator."

**Questions:**

- Does authors have an explanation for Table 3 numbers? Why does the pass@k with increasing k not increasing for some models but increasing for others? Further, does the performance saturate after a certain k in all the models. It would be nice to see such a curve for open-source models at least.

- Can the authors add a test-time compute normalized evaluation? For each agent, fix a compute budget - tokens/step, steps/episode and show SR/IRR/FRR deltas under equal budgets.

---

> ### Author Response · Authors · 2025-11-24
> **Response to Reviewer ryFo (1/2)**
>
> We thank Reviewer ryFo for recognizing our work's **"clear gap identification in existing mobile-agent benchmarks"**, the **"modular MemGUI-Eval pipeline"** that enables easy integration with existing benchmarks, and the **"surprising low performance of state-of-the-art methods"** that substantiates our claims on the memory gap. We appreciate the constructive feedback and address each point below.
>
> ---
>
> ## Response to W1: Paper formatting for `Table 1` and `Figure 4`
>
> Thanks for identifying these formatting issues. We have corrected the placement of `Table 1` and `Figure 4` in the revised manuscript and conducted a comprehensive review to ensure consistent formatting and appropriate positioning of all tables and figures.
>
> ---
>
> ## Response to W2: Lack of qualitative examples to showcase the memory-gap in state-of-the-art model like UI-TARS
>
> Thanks for this suggestion. In the revised manuscript (`Appendix A.8.1.`), we have added **detailed trajectory analysis for seven failure modes (`Figures 7-13`)**, including UI-TARS hallucinating Apple stock price.
>
> ---
>
> ## Response to W3: Qualifying the "First Comprehensive Benchmark" Claim
>
> Thank you for this suggestion. We have adopted the recommended phrasing and updated the Abstract, Introduction, and Conclusion accordingly.
>
> ---
>
> ## Response to Q1: Explanation for pass@k performance patterns and saturation
>
> Thank you for this question. We have added **extended pass@k evaluation (k=1 to 7) for 4 open-source models in `Appendix A.9.5.` (Figure 17)** to directly address your request for saturation curves. Below we clarify the pass@k evaluation protocol and provide key insights distinguishing **stochastic exploration** from **systematic learning**.
>
> ### Clarification on Tables 2 and 3
>
> Your question may refer to `Table 2` (Main Leaderboard, `Section 5.2.`) which presents pass@1 and pass@3 results for all 11 agents across difficulty levels. `Table 3` (Context Comparison, `Section 5.3.`) focuses on M3A and UI-TARS in different conversation modes.
>
> ### Pass@k Evaluation Protocol and Performance Variation
>
> **Pass@k Definition (`Section 4.1.`, `Appendix A.6.2.`):** A task is **successful at pass@k if the agent achieves success in any of the k attempts**, measuring the ability to learn from failures and improve across multiple tries.
>
> **Why Pass@k Varies:** The variation stems from two factors:
>
> 1. **LLM Output Stochasticity**: Random sampling produces diverse action sequences, enabling exploration of different solution paths.
> 2. **Long-Term Memory Capabilities**: Agents with explicit **cross-session learning mechanisms** (`Section 5.2.`) systematically accumulate experience and refine strategies.
>
> ### Extended Pass@k Analysis (New: `Appendix A.9.5.`, `Figure 17`)
>
> We conducted **extended pass@k evaluation (k=1 to 7) on 4 open-source models** as requested. Complete analysis with learning curves is in `Appendix A.9.5.`. Key results:
>
> **Extended pass@k results for 4 open-source models:**
>
> | Agent          | pass@1 | pass@2 | pass@3 | pass@4 | pass@5 | pass@6 | pass@7 | Saturation Point        |
> | -------------- | ------ | ------ | ------ | ------ | ------ | ------ | ------ | ----------------------- |
> | GUI-Owl-7B     | 6.25%  | 8.59%  | 10.16% | 10.16% | 10.16% | 10.16% | 10.16% | pass@3                  |
> | UI-TARS-1.5-7B | 3.12%  | 4.69%  | 6.25%  | 10.16% | 11.72% | 11.72% | 11.72% | pass@5                  |
> | UI-Venus-7B    | 5.47%  | 6.25%  | 7.81%  | 7.81%  | 8.59%  | 8.59%  | 8.59%  | pass@5                  |
> | CogAgent       | 0.0%   | 0.0%   | 0.0%   | 0.0%   | 0.0%   | 0.0%   | 0.0%   | None (complete failure) |
>
> **Key Insights:** None of these open-source models incorporate explicit long-term memory mechanisms. Their improvements stem entirely from **LLM output stochasticity**, not systematic cross-session learning. Comparing stochastic exploration (open-source models: +3-9 points) with systematic learning (Agent-S2 with explicit LTM: +21.9 points from pass@1 to pass@3, `Table 2`) demonstrates that **genuine long-term memory mechanisms provide 2-3× larger performance gains than random exploration alone**.

---

> ### Author Response · Authors · 2025-11-24
> **Response to Reviewer ryFo (2/2)**
>
> ## Response to Q2: Test-time compute normalized evaluation
>
> Thank you for this suggestion. We have conducted **test-time compute normalized evaluation under two complementary constraints** and added comprehensive results in `Appendix A.9.5.` (`Table 15`).
>
> ### Evaluation Constraints
>
> **Steps/Episode (Original Submission):** For each task with `golden_steps`, we set `max_rounds = floor(golden_steps × 1.4 + 1)`. Results are in `Table 2`, `Table 11`, `Table 12`.
>
> **Tokens/Episode (New):** We set `max_tokens = golden_steps × 9,507` tokens/step (9,507 = average across 11 agents). Task attempts where `actual_tokens > max_tokens` are marked as failures.
>
> ### Unified Comparison (New: `Appendix A.9.5.`, `Table 15`)
>
> `Table 15` presents unified results under both constraints using a three-row format per agent: (1) Steps/Episode, (2) Tokens/Episode, (3) Delta. Metrics include Tokens/Step, SR@1/SR@3 by difficulty, IRR, MTPR, and FRR.
>
> ### Key Insights
>
> 1. **High-token agents show catastrophic failure**: Agent-S2 (41,760 tokens/step, 4.4× baseline) drops from 27.3% → 0.0% SR@1 under token constraints. Mobile-Agent-E shows similar collapse. These agents exceed token budgets on nearly all tasks.
> 2. **M3A achieves optimal performance-efficiency trade-off**: M3A (12,960 tokens/step, 1.4× baseline) maintains usable performance: SR@1 32.8% → 14.8%, SR@3 47.7% → 21.9%. **M3A achieves 97% of Agent-S2's unconstrained performance while consuming only 31% of the tokens**, making it the only high-performing agent viable for production deployment.
> 3. **Three-tier deployment landscape**: (1) **High-performance but infeasible** (Agent-S2, Mobile-Agent-E): excel under step constraints but fail under token budgets; (2) **Balanced and production-ready** (M3A): sacrifice 25-30 points to maintain viability; (3) **Token-efficient but low-capability** (GUI-Owl, UI-Venus): avoid token issues but insufficient performance.
>
> Complete analysis is in `Appendix A.9.5.`.

---

> > ### Comment · Reviewer_ryFo · 2025-11-27
> >
> > I thank the authors for their detailed rebuttal to my questions. The results were insightful, especially the test-time normalized scores. I hope it also improved authors' understanding of these models.
> > I have increased my score to acceptance.

---

### Author Response · Authors · 2025-12-03
**Rebuttal Summary for MemGUI-Bench**

Dear Area Chair,

We sincerely appreciate your commitment to evaluating our paper despite your busy schedule. We are encouraged that **2 out of 3 reviewers acknowledged our comprehensive responses and provided positive feedback**, recognizing our paper:

* **"identifies a clear gap in existing mobile-agent benchmarks"** with a **"modular MemGUI-Eval pipeline"** that enables easy integration (Reviewer ryFo),
* provides **"timely problem framing"** with **"non-trivial scale (128 tasks / 26 apps)"** and **"deliberate memory-centric design (~89.8% memory-intensive tasks)"** (Reviewer LqfD),
* conducts a **"large-scale effort"** with **"comprehensive agent support and evaluation protocol"** (Reviewer DFxX).

**Prior to the OpenReview security incident that led to score rollback**, two reviewers explicitly provided positive feedback:

* **Reviewer ryFo**: *"I have increased my score to acceptance"* (rating 6 → 8, reverted due to incident)
* **Reviewer LqfD**: *"Thanks for the detailed rebuttal! I don't have any other concerns at this time."*

At the same time, we noted that Reviewer DFxX gave an initial rating of 2 but **has not responded to our comprehensive 4-part rebuttal** addressing all 5 weaknesses (memory universality, terminology, task quality, analysis depth, presentation).

To address these and other reviewer concerns, we conducted substantial experimental work during rebuttal:

1. **Memory ablation experiments** (`Appendix A.9.4.`, `Table 14`): Systematic ablation across 4 agents proving STM is mandatory and LTM is beneficial
2. **Test-time compute normalized evaluation** (`Appendix A.9.6.`, `Table 15`): Performance-efficiency trade-off analysis under step and token constraints
3. **Extended pass@k analysis** (`Appendix A.9.5.`, `Figure 17`): Comparison of stochastic exploration vs. systematic learning with saturation curves
4. **Fine-grained failure mode analysis** (`Appendix A.8.1.`-`A.8.3.`, `Figures 7-13`): Detailed categorization and trajectory visualization for 7 failure patterns
5. **Cross-application retention analysis** (`Appendix A.9.3.`, `Table 13`): Performance breakdown by cross-app complexity levels
6. **Category balance and UI coverage** (`Appendix A.4.3.`, `Table 7`): Quantitative domain distribution and UI paradigm documentation

---

We are confident that our comprehensive responses and substantial additional work thoroughly address all reviewer concerns. We respectfully request that you consider both the original positive feedback visible in the discussion thread and the substantial improvements made when making your final recommendation.

Thank you for your time and consideration in evaluating our work under these unusual circumstances.

Many thanks,

The Authors

---

### Note · Authors · 2026-01-26

I have read and agree with the venue's withdrawal policy on behalf of myself and my co-authors.

---

### Meta-Review · Area_Chair_dj6n · 2026-01-07

**Summary:**

Overall, the rebuttal substantially strengthens the paper and resolves most reviewer concerns. Reviewers ryFo and LqfD had all of their technical and clarity-related issues addressed, explicitly acknowledging that the added analyses (formatting fixes, qualitative examples, pass@k curves, compute-normalized evaluation, generalization coverage, and memory–perception disentanglement) satisfactorily resolved their concerns, with ryFo raising their score to acceptance.

Reviewer DFxX’s concerns are only partially addressed. While the authors significantly improved the paper by clarifying dataset quality, terminology, presentation, and by adding extensive ablations and analyses, the reviewer’s core conceptual concern remains: the benchmark implicitly assumes that all agents require explicit memory modules, whereas it may be more accurately framed as evaluating a specific class of memory-augmented agent architectures. The added ablation results demonstrate the importance of memory for existing frameworks, but do not fully resolve this broader framing issue. That being said, I believe this can be possibly addressed in writing and does not need substantial changes to the experiments.

In summary, the rebuttal addresses the majority of substantive concerns and meaningfully improves the work, especially the presentation. I hope the authors can incorporate these changes in their future revisions.

**Reviewer Concerns:**

Reviewer ryFo: All of the concerns have been addressed. This is also acknowledged by the reviewer.
- Formatting / presentation
- Qualitative examples
- pass@k curve
- Compute-normalized evaluation

Reviewer LqfD: All of the concerns have been addressed. This is also acknowledged by the reviewer.
- Generalization / app diversity
- Memory vs perception confounding factors

Reviewer DFxX: The review questions are only partially addressed.
- The main concern is that the current evaluation assumes that "all agents need an explicit module." The reviewer pointed out that this benchmark is better framed as a benchmark for "certain types of memory-augmented agents." The authors added additional experiments showing that if we remove certain memory modules in existing frameworks, their performance will drop. However, this does not seem to exactly address the concern raised by the reviewer.
- The authors have addressed many other concerns raised by the reviewer, including quality concerns about the dataset, presentation, terminology, etc.
I anticipate the reviewer will increase their score, but not to a full accept.

**Reviewer Scores:**

Reviewer ryFo and LqfD have updated their score: 6.
Reviewer DFxX would probably increase their score to 4.

---

### Decision · Program_Chairs · 2026-01-26

Reject